# Wasserstein Convergence of Critically Damped Langevin Diffusions

**Stanislas Strasman**[1*]    **Sobihan Surendran**[1,2*]    **Claire Boyer**[3]    **Sylvain Le Corff**[1]

**Vincent Lemaire**[1]    **Antonio Ocello**[4]

[1]Sorbonne Université and Université Paris Cité, CNRS, LPSM, F-75005 Paris, France
[2]LOPF, Califrais' Machine Learning Lab, Paris, France
[3]LMO, Université Paris-Saclay, UMR CNRS 8628, Institut Universitaire de France, Orsay, France
[4]CREST, ENSAE, Institut Polytechnique de Paris, Palaiseau, France

## Abstract

Score-based Generative Models (SGMs) have achieved impressive performance in data generation across a wide range of applications and benefit from strong theoretical guarantees. Recently, methods inspired by statistical mechanics, in particular, Hamiltonian dynamics, have introduced Critically-damped Langevin Diffusions (CLDs), which define diffusion processes on extended spaces by coupling the data with auxiliary variables. These approaches, along with their associated score-matching and sampling procedures, have been shown to outperform standard diffusion-based samplers numerically. In this paper, we analyze a generalized dynamic that extends classical CLDs by introducing an additional hyperparameter controlling the noise applied to the data coordinate, thereby better exploiting the extended space. We further derive a novel upper bound on the sampling error of CLD-based generative models in the Wasserstein metric. This additional hyperparameter influences the smoothness of sample paths, and our discretization error analysis provides practical guidance for its tuning, leading to improved sampling performance.

## 1   Introduction

Recent surge in machine learning and artificial intelligence has driven substantial progress in generative modeling, particularly with the development of Score-based Generative Models (SGMs). These models build on the foundational works in Denoising Diffusion Probabilistic Models (DDPMs) by Sohl-Dickstein et al. (2015); Song and Ermon (2019); Ho et al. (2020) and the advances in score-matching techniques introduced by Hyvärinen and Dayan (2005); Vincent (2011).

**Score-based Generative Models (SGMs).**   SGMs are probabilistic models designed to create synthetic instances of a target distribution when only a genuine sample (*e.g.*, a dataset of real-life images) is accessible. First, the forward process involves progressively perturbing the training distribution by adding noise to the data until its distribution approximately reaches an easy-to-sample distribution $\pi_\infty$. Then, the backward process involves learning to reverse this noising dynamics by sequentially removing the noise. SGMs have quickly gained recognition for their ability to generate high-quality synthetic data. Their applications span diverse areas, including computer vision (Li et al., 2022; Lugmayr et al., 2022), natural language processing (Gong et al., 2023), and other domains where realistic data generation is crucial. This growing body of work has been comprehensively surveyed by Yang et al. (2023), highlighting the versatility and potential of diffusion models. In

addition, SGMs provide a particularly interesting class of prior distributions to solve Bayesian inverse problems. Although they lack an explicit and tractable probability density function, a very active research area focuses on combining Monte Carlo guidance and SGMs to solve posterior sampling problems, Wu et al. (2023); Moufad et al. (2025); Victorino Cardoso et al. (2024).

**Critically-damped Langevin Diffusion (CLD).**   In Dockhorn et al. (2022), the authors proposed Critically-damped Langevin Diffusion as a second-order extension of conventional diffusion models. By introducing velocity variables alongside the usual state variables —much like in Hamiltonian Monte Carlo— CLD accelerates exploration of high-dimensional spaces and often yields better sample quality in practice. Although empirical work demonstrates the benefit of CLD over standard score-based models (Dockhorn et al., 2022), its theoretical underpinnings remain incomplete. Existing convergence guarantees are only expressed in terms of Kullback–Leibler divergence (Conforti et al., 2025; Chen et al., 2023) and fail to capture any computational advantage for kinetic dynamics, leaving a gap between observed performance and formal analysis.

**Contributions.**   We first discuss the challenges of establishing Wasserstein convergence under the standard assumptions used for Variance-Preserving (VP) or Variance-Exploding (VE) SGMs, where the forward process is elliptic (Gao et al., 2025; Strasman et al., 2025; Gentiloni-Silveri and Ocello, 2025; Bruno et al., 2025). We then provide, to the best of our knowledge, the first upper bound for CLD in the Wasserstein metric through coupling techniques under weaker assumptions, achieving convergence rates comparable to those of other SGMs. Crucially, this result is not implied by previous Kullback–Leibler divergence bounds (Conforti et al., 2025; Chen et al., 2023), and our proof technique differs significantly from existing Wasserstein analyses of diffusion models.

However, it is possible to introduce a modified dynamics that includes an additional hyperparameter controlling the noise on the data coordinate of CLD, thereby restoring ellipticity and enabling an analysis closely aligned with that of VP and VE models, but formulated on an extended phase space with matrix-valued drifts and diffusions. This hyperparameter governs the smoothness of sample paths, allowing a detailed analysis of the generative error as a function of this smoothness parameter. Such analysis offers practical guidance for tuning this hyperparameter and potentially improves sampling performance compared to standard SGMs and CLD methods. The benefits of this additional parameterization are demonstrated numerically on challenging synthetic datasets.

## 2   Notation and Background

**Notation.**   We use $\pi$ to denote probability distributions and $p$ to denote their corresponding densities with respect to the Lebesgue measure or another reference measure. The identity matrix of size $d$ is written $\mathbf{I}_d$. For $x, y \in \mathbb{R}^d$, we denote by $\langle x, y \rangle$ the standard inner product of $\mathbb{R}^d$, by $\|\cdot\|$ the Euclidean norm for vectors and its induced operator norm for matrices. Let $\|\cdot\|_F$ be the Frobenius norm defined for $A \in \mathbb{R}^{d \times d}$ as $\|A\|_F := \sqrt{\mathrm{Tr}(A^\top A)}$. For symmetric matrices $A, B \in \mathbb{R}^{d \times d}$, we write $A \preccurlyeq B$ to mean that $B - A$ is positive semidefinite. We denote the time derivative of a function by $\dot{f}(t) := \frac{\mathrm{d}}{\mathrm{d}t} f(t)$. We use the symbol $\otimes$ either for the Kronecker product when applied to matrices and for the product of probability measures when applied to distributions. The intended meaning will be clear from context. For any matrix $A \in \mathbb{R}^{d \times d}$ we denote its largest eigenvalue (resp. singular value) by $\lambda_{\max}(A)$ (resp. $\sigma_{\max}(A)$) and smallest eigenvalue by $\lambda_{\min}(A)$ (resp. $\sigma_{\min}(A)$). For random vectors $X, Y \in \mathbb{R}^d$, define $\|X\|_{L_2} := \left( \mathbb{E}\left[ \|X\|^2 \right] \right)^{1/2}$ and we write $X \perp Y$ to mean that $X$ is independent of $Y$. The notation $\mathcal{L}(X)$ denotes the law (distribution) of a random vector $X$. For $a, b \in \mathbb{R}$, we write $a \wedge b := \min\{a, b\}$ and $a \vee b := \max\{a, b\}$.

**Score-based Generative Models.**   SGMs employ a Gaussian Markovian diffusion process that smoothly transports the target data distribution $\pi_{\mathrm{data}} \in \mathcal{P}(\mathbb{R}^d)$ towards an easy-to-sample Gaussian distribution $p_\infty \in \mathcal{P}(\mathbb{R}^d)$. This process, known as forward diffusion, is the solution to the following stochastic differential equation (SDE) on a fixed time horizon $t \in [0, T]$,

$$\mathrm{d}\overrightarrow{X}_t = -\alpha\beta(t)\overrightarrow{X}_t \mathrm{d}t + \sqrt{2\beta(t)}\mathrm{d}B_t, \quad \overrightarrow{X}_0 \sim \pi_{\mathrm{data}}, \tag{1}$$

with $(B_t)_{t \in [0,T]}$ a $d$-dimensional Brownian motion and $\beta(t) : [0, T] \to \mathbb{R}_+$. In particular, when $\alpha = 0$ and $\beta(t)$ is of the form $\beta^{\mathrm{VE}}(t)\dot{\beta}^{\mathrm{VE}}(t)$ the process is known as Variance Exploding (Song

and Ermon, 2019) and when $\alpha = 1$, the process is known as Variance Preserving (Sohl-Dickstein et al., 2015; Ho et al., 2020). This transformation can be reversed (Anderson, 1982; Haussmann and Pardoux, 1986; Cattiaux et al., 2023) and is also governed by an SDE, known as the backward process

$$\mathrm{d}\overleftarrow{X}_t = \left( \alpha\beta(T-t)\overleftarrow{X}_t + 2\beta(T-t)\nabla \log p_{T-t}(\overleftarrow{X}_t) \right) \mathrm{d}t + \sqrt{2\beta(T-t)}\mathrm{d}B_t, \quad \overleftarrow{X}_0 \sim p_T, \quad (2)$$

where $p_t$ is the time marginal p.d.f. of the forward process for $0 \leq t \leq T$. As a consequence, $\overleftarrow{X}_T$ has the same distribution as $\pi_{\mathrm{data}}$. In practice, however, one cannot draw exact i.i.d. samples from this continuous-time process, and implementations of SGMs rely on three key approximations.

- *Mixing error.* The distribution of $\overrightarrow{X}_T$ is not analytically available in most cases, $\overleftarrow{X}_0$ is initialized at a known distribution $\pi_\infty$, close to $p_T$.

- *Discretization error.* In most cases, the backward dynamic is non-linear, the backward process is discretized to sample from $\overleftarrow{X}_T$, which introduces an error due to evaluating the (time-continuous) score function only at discrete time steps.

- *Approximation error.* The score function depends on the unknown data distribution and thus cannot be computed in closed form. To approximate it, we use a neural network architecture $s_\theta : [0, T] \times \mathbb{R}^d \mapsto \mathbb{R}^d$ parameterized by $\theta \in \Theta$, and trained, for example, via Denoising Score Matching (see, *e.g.*, Vincent, 2011):

$$\mathcal{L}_{\mathrm{DSM}}(\theta) = \mathbb{E}\left[ \lambda(t) \left\| s_\theta\left(\tau, \overrightarrow{X}_\tau\right) - \nabla \log p_\tau\left(\overrightarrow{X}_\tau | \overrightarrow{X}_0\right) \right\|^2 \right], \quad (3)$$

where $\tau$ is uniformly distributed on $[0, T]$, $\tau$ is independent of $\overrightarrow{X}_0$, $\overrightarrow{X}_\tau \sim p_\tau(\cdot | \overrightarrow{X}_0)$ and $\lambda : [0, T] \to \mathbb{R}_{>0}$ is a positive weighting function.

Theoretical studies of SGMs focus on those sources of errors to derive results for the total variation distance (De Bortoli et al., 2021), the Kullback–Leibler divergence (Conforti et al., 2025; De Bortoli et al., 2021; Chen et al., 2023; Benton et al., 2024) or the Wasserstein-2 distance (Lee et al., 2022, 2023; Bruno et al., 2025; Gao et al., 2025; Strasman et al., 2025; Gentiloni-Silveri and Ocello, 2025).

**Kinetic Ornstein–Uhlenbeck.** Inspired by Hamiltonian mechanics, kinetic SGMs operate in an extended position-velocity phase space, defined as $\overrightarrow{\mathbf{U}}_t = (\overrightarrow{X}_t, \overrightarrow{V}_t)^\top \in \mathbb{R}^{2d}$ which satisfies the following stochastic differential equation

$$\mathrm{d}\overrightarrow{\mathbf{U}}_t = A\overrightarrow{\mathbf{U}}_t\mathrm{d}t + \Sigma\mathrm{d}B_t, \qquad \overrightarrow{\mathbf{U}}_0 \sim \pi_{\mathrm{data}} \otimes \pi_v, \quad (4)$$

where $\pi_v \sim \mathcal{N}(0, v^2\mathbf{I}_d)$, $(B_t)_{t\in[0,T]}$ denotes a $2d$-dimensional standard Brownian motion,

$$A = \begin{pmatrix} 0 & a^2 \\ -1 & -2a \end{pmatrix} \otimes \mathbf{I}_d, \quad \text{and} \quad \Sigma = \begin{pmatrix} 0 & 0 \\ 0 & \sigma \end{pmatrix} \otimes \mathbf{I}_d. \quad (5)$$

Similar to (1), this process is Gaussian conditional on the distribution at time 0 (see Proposition A.2). The associated linear system corresponds to the stochastic analogue of a damped harmonic oscillator in the critically damped regime, with $a = 1/\sqrt{M}$ and $\sigma = 2/\sqrt{a}$, following the parameterization of Dockhorn et al. (2022). Note that (4) can also be expressed using a time-change or noise-schedule function $\beta : [0, T] \to \mathbb{R}_+$ (see Section E.2). This will not play a key role in our theoretical analysis but is an important feature of practical numerical implementation.

Applying time-reversal results for diffusion processes (see, *e.g*, Haussmann and Pardoux, 1986; Cattiaux et al., 2023), the backward process $(\overleftarrow{\mathbf{U}}_t)_{t\geq 0}$ is solution to the following SDE:

$$\mathrm{d}\overleftarrow{\mathbf{U}}_t = -A\overleftarrow{\mathbf{U}}_t\mathrm{d}t + \Sigma^2\nabla \log p_{T-t}\left(\overleftarrow{\mathbf{U}}_t\right)\mathrm{d}t + \Sigma\mathrm{d}B_t, \quad (6)$$

with initial condition $\overleftarrow{\mathbf{U}}_0 \sim p_T$, where $p_t : \mathbb{R}^{2d} \to \mathbb{R}_+$ is the probability density function of $\overrightarrow{\mathbf{U}}_t$.

**CLD-based SGMs.** To sample from $\overleftarrow{\mathbf{U}}_t$ (and, in particular, from $\overleftarrow{X}_T \sim \pi_{\mathrm{data}}$), one must rely on the three SGM approximations discussed earlier. The *mixing error* is analogous to that of standard SGMs and leverages the ergodicity of the forward process—converging to a known Gaussian

distribution—to initialize the backward process. The *discretization* of the nonlinear backward SDE can be performed using classical numerical integrators commonly employed in SGMs, such as Euler–Maruyama (Song et al., 2021) or exponential integrators (Conforti et al., 2025). Additionally, due to the Hamiltonian structure of the kinetic process, symplectic integrators (Neal, 2011) may also be appropriate (Dockhorn et al., 2022). Finally, the *score approximation* can be implemented by applying Denoising Score Matching–similar to (3)–on the extended phase space $\overrightarrow{\mathbf{U}}_t = (\overrightarrow{X}_t, \overrightarrow{V}_t)^\top$, that is, using the conditional score function $\nabla \log p_t(\overrightarrow{\mathbf{U}}_t | \overrightarrow{\mathbf{U}}_0)$. However, since the distribution of $\overrightarrow{V}_0$ is known and Gaussian, it can be analytically marginalized, yielding the following objective function known as Hybrid Score Matching:

$$\mathcal{L}_{\text{HSM}}(\theta) = \mathbb{E}\left[\lambda(t)\left\|s_\theta(\tau, \overrightarrow{\mathbf{U}}_\tau) - \nabla \log p_\tau(\overrightarrow{\mathbf{U}}_\tau \mid \overrightarrow{X}_0)\right\|^2\right],$$

where $\tau$ is uniformly distributed on $[0, T]$, $\tau \perp \overrightarrow{X}_0$, $\overrightarrow{\mathbf{U}}_\tau \sim p_\tau(\cdot | \overrightarrow{X}_0)$ and $\lambda : [0, T] \to \mathbb{R}_{>0}$ is a positive weighting function. Empirically, Hybrid Score Matching tends to yield more stable training dynamics by reducing the variance of the training objective (Dockhorn et al., 2022).

## 3 Wasserstein Convergence of CLDs

In this section, we analyze the convergence of CLDs with respect to the 2-Wasserstein distance under the Euler–Maruyama discretization scheme. We first discuss the motivation for this analysis before introducing the setting, assumptions, and main results.

### 3.1 Motivation

While convergence results have been established in terms of the Kullback–Leibler divergence (Conforti et al., 2025; Chen et al., 2023), no analogous results currently exist for the Wasserstein-2 metric. Proving convergence in $\mathcal{W}_2$ requires establishing a contraction property of the backward dynamics in this metric—a challenging task for hypo-coercive SDEs (Villani, 2009; Eberle et al., 2019; Monmarché, 2023). The main difficulty arises from the degeneracy of the diffusion term, since the Brownian motion in CLDs acts only on the velocity component. To illustrate this point, consider the following example.

Introduce the change of variables $\overrightarrow{Y}_t = \overrightarrow{X}_t + a\overrightarrow{V}_t$, under which one component of the system evolves as an Ornstein–Uhlenbeck process. Writing $\overrightarrow{Z}_t = (\overrightarrow{X}_t, \overrightarrow{Y}_t)^\top$, the forward SDE in (4) can be rewritten as

$$d\overrightarrow{Z}_t = a\begin{pmatrix} -1 & 1 \\ 0 & -1 \end{pmatrix}\overrightarrow{Z}_t dt + \begin{pmatrix} 0 & 0 \\ 0 & \sigma \end{pmatrix}dB_t.$$

Notably, the transformed process $(\overrightarrow{Y}_t)_{t\in[0,T]}$ corresponds to an Ornstein–Uhlenbeck process. By the time-reversal property, the corresponding backward process satisfies

$$\overleftarrow{Y}_t = \overleftarrow{X}_t + a\overleftarrow{V}_t,$$

which leads to the following backward SDE:

$$d\overleftarrow{Z}_t = a\begin{pmatrix} 1 & -1 \\ 0 & 1 \end{pmatrix}\overleftarrow{Z}_t dt + \sigma^2\begin{pmatrix} 0 \\ \nabla_y \log p_{T-t}\left(\overleftarrow{Z}_t\right) \end{pmatrix}dt + \begin{pmatrix} 0 & 0 \\ 0 & \sigma \end{pmatrix}dB_t, \tag{7}$$

where $p_t$ denotes the probability density function of $\overrightarrow{Z}_t$. A standard approach to establishing contraction consists in studying the difference process associated with the dynamics in (7), starting from two deterministic initial conditions $(x_0, y_0), (x_0', y_0') \in \mathbb{R}^{2d}$ and denoting by $(X_t, Y_t)_{t\in[0,T]}$ and $(X_t', Y_t')_{t\in[0,T]}$ the corresponding solutions. Under a synchronous coupling–*i.e.* using the same Brownian motion to drive the evolution of both processes–the difference process becomes a deterministic ODE, whose stability properties determine the contraction properties of the system. In particular, using the mean value theorem applied to the gradients of the log-density, the following holds for $t \in [0; T]$:

$$d\begin{pmatrix} X_t - X_t' \\ Y_t - Y_t' \end{pmatrix} = \begin{pmatrix} a + \sigma^2 G_t & -a \\ 0 & a + \sigma^2 H_t \end{pmatrix}\begin{pmatrix} X_t - X_t' \\ Y_t - Y_t' \end{pmatrix}dt. \tag{8}$$

where

$$H_t = \int_0^1 \nabla_y^2 \log p_{T-t}\big(X_t' + \gamma(X_t - X_t'),\ Y_t' + \gamma(Y_t - Y_t')\big)\,\mathrm{d}\gamma\,,$$

$$G_t = \int_0^1 \nabla_y \nabla_x^\top \log p_{T-t}\big(X_t' + \gamma(X_t - X_t'),\ Y_t' + \gamma(Y_t - Y_t')\big)\,\mathrm{d}\gamma\,.$$

To ensure contraction of the system, all eigenvalues of the matrix in (8) must be negative. However, the main difficulty lies in controlling the term $G_t$, which involves the mixed second-order derivative $\nabla_y \nabla_x^\top \log p_{T-t}$. For contraction to occur, this term must also be sufficiently negative. This is a strong and challenging requirement, as it demands a form of joint concavity of cross-derivatives, which is not generally ensured even when $p_{T-t}$ is strongly log-concave in each variable separately.

## 3.2 Settings: Dynamics and Backward Discretization

**Position-noise regularization in the extended phase space.** As detailed in Dalalyan and Riou-Durand (2020), kinetic Langevin-based samplers depend on the mixing rate and on the regularity of the underlying dynamics. To better exploit the extended phase space, we introduce a modified dynamics that adds a small noise term on the position coordinate $\varepsilon \geq 0$. Crucially, when $\varepsilon$ is strictly positive, this modification restores ellipticity of the forward and backward processes, which facilitates greatly the theoretical analysis. This hyperparameter controls the smoothness of the sample paths and the analysis of the discretization error allows a practical tuning to improve sampling performance in comparison with standard SGM models and kinetic-based diffusion samplers. The diffusion coefficient of the forward SDE is then given by

$$\Sigma_\varepsilon := \begin{pmatrix} \varepsilon & 0 \\ 0 & \sigma \end{pmatrix} \otimes \mathbf{I}_d\,,$$

giving a process $(\overrightarrow{\mathbf{U}}_t)_{t \in [0,T]} \in \mathbb{R}^{2d}$ which satisfies the following SDE

$$\mathrm{d}\overrightarrow{\mathbf{U}}_t = A\overrightarrow{\mathbf{U}}_t\mathrm{d}t + \Sigma_\varepsilon \mathrm{d}B_t\,, \qquad \overrightarrow{\mathbf{U}}_0 \sim \pi_{\mathrm{data}} \otimes \pi_v \tag{9}$$

with $\varepsilon \geq 0$. Note that the case $\varepsilon = 0$ recovers the classical CLD framework. In the following, we write

$$\mathsf{s}_t(u) = \nabla \log p_t(u)\,, \quad \text{for } t \geq 0,\ u \in \mathbb{R}^{2d}\,. \tag{10}$$

**Modified score function.** Following Conforti et al. (2025), we adopt a modified score formulation based on the rescaled density $\tilde{p}_t := p_t/p_\infty$, where $p_\infty$ is the density of the stationary distribution associated with (4). This perspective, also emphasized in Cattiaux et al. (2023); Conforti and Léonard (2022); Strasman et al. (2025); Conforti et al. (2025); Gentiloni-Silveri and Ocello (2025); Pham et al. (2025), reveals deep connections with stochastic control theory. In particular, the modified score satisfies a Hamilton–Jacobi–Bellman (HJB) equation, which we highlight and exploit in the sequel. With this notation, the backward process $\overleftarrow{\mathbf{U}}$ can be written equivalently as

$$\mathrm{d}\overleftarrow{\mathbf{U}}_t = \tilde{A}_\epsilon \overleftarrow{\mathbf{U}}_t \mathrm{d}t + \Sigma_\varepsilon^2 \nabla \log \tilde{p}_{T-t}\left(\overleftarrow{\mathbf{U}}_t\right)\mathrm{d}t + \Sigma_\varepsilon \mathrm{d}B_t\,, \tag{11}$$

with $\tilde{A}_\epsilon = -A - \Sigma_\epsilon^2 \Sigma_\infty^{-1}$. In the following, we write $\tilde{\mathsf{s}}_t(u) := \nabla \log \tilde{p}_t(u)$, for $t \geq 0$, $u \in \mathbb{R}^{2d}$.

**Backward process discretization.** Let $N \in \mathbb{N}$ denote the number of discretization steps, so that $0 = t_0 < t_1 < \ldots < t_N = T$. To analyze the convergence of the discretized backward process, we introduce the continuous-time interpolation $(\bar{\mathbf{U}}_t)_{t \in [0,T]}$ of the Euler scheme for the time-reversed process $(\overleftarrow{\mathbf{U}}_t)_{t \in [0,T]}$. This is defined as the Itô process such that, for $t \in [t_k, t_{k+1}]$,

$$\bar{\mathbf{U}}_t = \bar{\mathbf{U}}_{t_k} + \left(\tilde{A}_\epsilon \bar{\mathbf{U}}_{t_k} + \Sigma_\varepsilon^2 \tilde{\mathsf{s}}_{T-t_k}\big(\bar{\mathbf{U}}_{t_k}\big)\right)(t - t_k) + \Sigma_\varepsilon\big(B_t - B_{t_k}\big)\,, \tag{12}$$

where the process is initialized at $p_T$ (i.e., $\bar{\mathbf{U}}_0 \sim p_T$). When initialized at $\pi_\infty$, we denote by $(\bar{\mathbf{U}}_t^\infty)_{t \in [0,T]}$ the corresponding Itô process. For simplicity, the discretization is performed on a uniform grid, with step size $h = T/N$, so that $t_{k+1} - t_k = h$ for all $k$.

**Generative model.** The *generative model* is defined as the continuous-time interpolation of the discretized backward process, in which the true (unknown) modified score function is replaced by its parametric approximation $\tilde{s}_\theta : [0,T] \times \mathbb{R}^{2d} \mapsto \mathbb{R}^d$. The resulting process, denoted by $(\bar{\mathbf{U}}_t^\theta)_{t \in [0,T]}$, satisfies for $t \in [t_k, t_{k+1}]$

$$\bar{\mathbf{U}}_t^\theta = \bar{\mathbf{U}}_{t_k}^\theta + \left( \tilde{A}_\epsilon \bar{\mathbf{U}}_{t_k}^\theta + \Sigma_\varepsilon^2 \tilde{s}_\theta\big(t_k, \bar{\mathbf{U}}_{t_k}^\theta\big) \right)(t - t_k) + \Sigma_\varepsilon \big(B_t - B_{t_k}\big), \tag{13}$$

with initialization $\bar{\mathbf{U}}_0^\theta \sim \pi_\infty$. Learning the modified score function $\nabla \log \tilde{p}_t$ is theoretically equivalent to learning the standard score function $\nabla \log p_t$, since the two functions differ only by a known linear term. As a consequence, the modified score approximation can be written, for any $t \geq 0$ and any $u \in \mathbb{R}^{2d}$, as

$$\tilde{s}_\theta(t, u) := s_\theta(t, u) + \Sigma_\infty^{-1} u\,.$$

Ultimately, the objective is to control the $\mathcal{W}_2$–distance between $\mathcal{L}(\bar{X}_T^\theta)$ the generated data marginal distribution at time $T$ and $\pi_{\mathrm{data}}$ the true data distribution (recall that $\bar{\mathbf{U}}_T^\theta = (\bar{X}_T^\theta, \bar{V}_T^\theta)^\top$).

### 3.3 Assumptions

**Regularity assumptions.**

**H1** The data distribution $\pi_{\mathrm{data}}$ is absolutely continuous w.r.t. the Lebesgue measure, with density $p_{\mathrm{data}}$ and the relative Fisher information between $\pi_0 = \pi_{\mathrm{data}} \otimes \pi_v$ (*i.e.* the initialization of the stochastic process defined in (4)) and $\pi_\infty$ is finite, *i.e.*

$$\mathcal{I}(\pi_0 | \pi_\infty) := \int \left\| \nabla \log \left( \frac{\mathrm{d}\pi_0}{\mathrm{d}\pi_\infty}(u) \right) \right\|^2 \pi_0(\mathrm{d}u) < \infty\,.$$

Assumption H1, particularly the requirement of finite Fisher information, is standard in most works establishing convergence bounds for SGMs. This condition is either explicitly assumed or implied by stronger regularity assumptions used in the literature (Conforti et al., 2025; Strasman et al., 2025).

**H2** $(i)$ The data distribution is of the form $p_{\mathrm{data}}(x) \propto \exp\left(-(V(x) + H(x))\right)$ and satisfies:
   * There exists $L > 0$ such that $|H(x) - H(y)| \leq L\|x - y\|$ for all $x, y \in \mathbb{R}^d$ .
   * There exists $\alpha > 0$ such that $\alpha \mathbf{I}_d \preceq \nabla^2 V(x)$ for all $x \in \mathbb{R}^d$.
$(ii)$ $(-\log p_{\mathrm{data}})$ is $L_0$-one-sided Lipschitz, *i.e.*, for all $x, y \in \mathbb{R}^d$,

$$-\left(\nabla \log p_{\mathrm{data}}(x) - \nabla \log p_{\mathrm{data}}(y)\right)^\top (x - y) \leq L_0 \|x - y\|^2\,. \tag{14}$$

The first point of Assumption H2 models the data distribution as a strongly log-concave component $V$ perturbed by a term $H$, similar to the settings considered in Brigati and Pedrotti (2025); Stéphanovitch (2025). Intuitively, this assumption allows the distribution to deviate from strong log-concavity via the perturbation $H$, while still maintaining sufficient regularity for the analysis. When $H = 0$, the distribution reduces to the strongly log-concave case, which is commonly used to establish contraction in the Wasserstein metric (Bruno et al., 2025; Gao et al., 2025; Strasman et al., 2025). The second point of Assumption H2 assumes a one-sided Lipschitz condition, which is weaker than requiring full Lipschitz continuity of the score function (Gentiloni-Silveri and Ocello, 2025). Notably, H2 implies the Lipschitz continuity of the score function. This means that for all $t \in (0, T]$, there exists $L_t > 0$ such that for all $u, \bar{u} \in \mathbb{R}^{2d}$,

$$\|\mathsf{s}_t(u) - \mathsf{s}_t(\bar{u})\| \leq L_t \|u - \bar{u}\|\ . \tag{15}$$

This condition can be verified under standard assumptions. In particular, if $\nabla \log p_{\mathrm{data}}$ is Lipschitz, the assumption holds. Since $\pi_{\mathrm{data}}$ and $\pi_v$ are independent and $\pi_v$ is often Gaussian, it suffices to assume that $p_{\mathrm{data}}$ is log-smooth, a common condition in the analysis of SGMs to ensure convergence (Gao et al., 2025; Chen et al., 2023).

Furthermore, Assumption H2 ensures that $\pi_{\mathrm{data}}$ has sub-Gaussian tails (Lemma D.1). Consequently, all its polynomial moments are finite. In particular, $\pi_{\mathrm{data}}$ admits a finite second moment, a standard condition—either explicit or implied by stronger regularity assumptions—in convergence analyses of SGMs. Importantly, Assumption H2, together with the polynomial growth condition $\|\nabla V(x)\| \leq C(1 + \|x\|^m)$ for all $x \in \mathbb{R}^d$, with some $C > 0$ and $m \in \mathbb{N}$, implies Assumption H1 (Lemma D.2).

These assumptions are satisfied by standard distributions such as Gaussian and mixtures of Gaussians. They are strictly weaker than the conditions typically required in the literature to establish Wasserstein convergence guarantees—such as strong log-concavity combined with the Lipschitz continuity of the score function (Gao et al., 2025; Strasman et al., 2025)—which hold only for non-degenerate Gaussian distributions and therefore exclude many practically relevant settings, even though they remain common in the literature.

**Score approximation.**

**H3** There exists $M \geq 0$ such that,

$$\sup_{k \in \{0,..,N-1\}} \left\| \tilde{\mathsf{s}}_{T-t_k} \left( \bar{\mathbf{U}}_{t_k}^\theta \right) - \tilde{s}_\theta \left( T - t_k, \bar{\mathbf{U}}_{t_k}^\theta \right) \right\|_{L_2} \leq M .$$

Assumption H3 is standard in the literature (De Bortoli et al., 2021; Conforti and Léonard, 2022; Gao et al., 2025; Bruno et al., 2025; Strasman et al., 2025; Gentiloni-Silveri and Ocello, 2025; Cordero-Encinar et al., 2025) as essentially all convergence proofs for diffusion-based score models require that the neural network has learned the score within some uniform error. This condition quantifies the ability of the neural network architecture to approximate the true score function and serves to control the score approximation error.

## 3.4 Main Results

We establish here the Wasserstein-2 convergence of CLD-based SGMs under these weak assumptions. A key step is to show that, under Assumption H2, the scaled score function $\Sigma_\varepsilon^2 \nabla \log p_t$ (resp. $\Sigma_\varepsilon^2 \nabla \log \tilde{p}_t$) is $L_t$-Lipschitz (resp. $\tilde{L}_t$-Lipschitz), for $t > 0$ (Proposition B.1). This, in particular, yields an exponential decay of the operator norm $\|\Sigma_\varepsilon^2 \nabla^2 \log \tilde{p}_t\|$ as $t \to \infty$. The following theorem provides, to the best of our knowledge, the first convergence rates in Wasserstein distance for CLD-based approaches and aligns with recent developments in the literature of Variance-Preserving and Variance-Exploding SGMs.

**Theorem 3.1.** *Assume that Assumptions H1- H3 hold. Then, there exist $c_1, c_2 > 0$ such that, for all $h > 0$,*

$$\mathcal{W}_2 \left( \pi_{\text{data}}, \mathcal{L} \left( \bar{X}_T^\theta \right) \right) \leq c_1 \mathrm{e}^{-c_2 T} \mathcal{W}_2 \left( \pi_{\text{data}} \otimes \pi_v, \pi_\infty \right) + c_1 \sigma^2 M + c_1 \sqrt{h} .$$

*Proof.* Let $P_X : \mathbb{R}^{2d} \to \mathbb{R}^d$ denote the projection $P_X(x, v) = x$. Using that $P_X$ is 1–Lipschitz for the Euclidean norm, yields,

$$\mathcal{W}_2 \left( \pi_{\text{data}}, \mathcal{L} \left( \bar{X}_T^\theta \right) \right) \leq \mathcal{W}_2 \left( \pi_{\text{data}} \otimes \pi_v, \mathcal{L} \left( \bar{\mathbf{U}}_T^\theta \right) \right) . \tag{16}$$

The right-hand side of (16) is then bounded by decomposing the total generation error, using the triangle inequality, into the three sources of error for SGMs discussed in Section 2:

$$\mathcal{W}_2 \left( \pi_{\text{data}} \otimes \pi_v, \mathcal{L} \left( \bar{\mathbf{U}}_T^\theta \right) \right) \leq \mathcal{W}_2 \left( \mathcal{L} \left( \overleftarrow{\mathbf{U}}_T \right), \mathcal{L} \left( \bar{\mathbf{U}}_T \right) \right) + \mathcal{W}_2 \left( \mathcal{L} \left( \bar{\mathbf{U}}_T^\infty \right), \mathcal{L} \left( \bar{\mathbf{U}}_T^\theta \right) \right)$$
$$+ \mathcal{W}_2 \left( \mathcal{L} \left( \bar{\mathbf{U}}_T \right), \mathcal{L} \left( \bar{\mathbf{U}}_T^\infty \right) \right) ,$$

where $\bar{\mathbf{U}}_T$ and $\bar{\mathbf{U}}_T^\infty$ are defined in Equation (12) and $\bar{\mathbf{U}}_T^\theta$ in Equation (13). The first term (discretization error) is controlled by Lemma B.2, which ensures that there exists $c_1 > 0$ such that, for all $h > 0$,

$$\mathcal{W}_2(\mathcal{L}(\overleftarrow{\mathbf{U}}_T), \mathcal{L}(\bar{\mathbf{U}}_T)) \leq c_1 \sqrt{h} .$$

The second term (score approximation error) is bounded by Lemma B.3,

$$\mathcal{W}_2 \left( \mathcal{L} \left( \bar{\mathbf{U}}_T^\infty \right), \mathcal{L} \left( \bar{\mathbf{U}}_T^\theta \right) \right) \leq c_1 \sigma^2 M .$$

Finally, the third term (mixing error) is controlled by Lemma B.4, which guarantees the existence of $c_2 > 0$ such that,

$$\mathcal{W}_2 \left( \mathcal{L} \left( \bar{\mathbf{U}}_T \right), \mathcal{L} \left( \bar{\mathbf{U}}_T^\infty \right) \right) \leq c_1 \mathrm{e}^{-c_2 T} \mathcal{W}_2 \left( \pi_{\text{data}} \otimes \pi_v, \pi_\infty \right) .$$

Combining these three bounds together with (16) concludes the proof. $\qquad \square$

Theorem 3.1 establishes convergence rates in the Wasserstein distance for CLD-based approaches for all $\epsilon \geq 0$, recovering the vanilla CLD when $\epsilon = 0$. In this case, our result aligns with the KL convergence analyses of kinetic Langevin dynamics by Chen et al. (2023) and Conforti et al. (2025) for the specific choice $a = 1$ and $\sigma = 2$. It is worth emphasizing that, under our weaker assumptions, no equivalence holds between KL and Wasserstein convergence, so our results are not implied by existing KL-based analyses. Beyond this theoretical bound, our analysis indicates that smaller values of $v$ yield better log-concavity constants; however, $v$ is typically chosen to be small but not too small, to avoid an explosion in the Lipschitz constant. This remark is consistent with the empirical evidence brought forward by Dockhorn et al. (2022), which suggests that small values of $v$ may improve training stability and sampling performance.

## 3.5 Strongly Log-Concave Case

This subsection focuses on the elliptic case, *i.e.*, when $\varepsilon > 0$. In this setting, the forward process associated with CLD becomes a *multidimensional Ornstein–Uhlenbeck process* with matrix-valued drift and diffusion coefficients. The presence of the additional noise term on the position coordinate restores ellipticity, which allows us to extend classical convergence analyses developed for VP and VE diffusions to this kinetic framework.

Crucially, in the strongly log-concave case, i.e., when $H = 0$ in Assumption H2, the upper bound can be expressed with more explicit constants that depend on the regularity of the data. Moreover, in this case, the one-sided Lipschitz condition becomes equivalent to the Lipschitz continuity of the score function. Assumption H2 then reduces to the following assumption.

**H2′** The data distribution is absolutely continuous w.r.t. the Lebesgue measure, is of the form $p_{\mathrm{data}}(x) \propto \mathrm{e}^{-V(x)}$ and is $\alpha_0$–strongly log-concave and $L_0$-log-smooth, *i.e.*, there exists $\alpha_0 > 0$ and $L_0 > 0$ such that,
$$\alpha_0 \mathbf{I}_d \preceq \nabla^2 V(x) \preceq L_0 \mathbf{I}_d, \qquad \text{for all } x \in \mathbb{R}^d .$$

Under this assumption, the forward flow preserves both strong log-concavity and smoothness. Indeed, Propositions C.1 and C.2 guarantee that $p_t$ remains $\alpha_t$–log-concave and $L_t$–log-smooth for all $t \in [0, T]$, with $\alpha_t$ and $L_t$ explicitly defined as functions of $\alpha_0$ and $L_0$ in the respective propositions. Such regularity properties are fundamental for proving exponential contraction in the Wasserstein metric, and are consistent with the analysis of classical (VP) diffusion models (Bruno et al., 2025; Gao et al., 2025; Strasman et al., 2025). In contrast, Chen et al. (2023) obtain Wasserstein convergence guarantees without requiring strong log-concavity, instead relying on the compactness of the domain and (15), a setting where convergence in KL divergence is effectively equivalent. The following theorem presents Wasserstein convergence results under assumptions for which no such equivalence with the KL divergence holds. In particular, our result is not implied by existing analyses based on KL convergence.

**Theorem 3.2.** *Assume that H2′ and H3 hold, and let $\varepsilon > 0$. If the step size $h$ satisfies*
$$0 < h < \frac{2\min_k \alpha_{t_k}\left(\sigma^2 \wedge \varepsilon^2\right) - (\sigma - \varepsilon)^2 \max_k L_{t_k} - (a+1)^2}{\|A\|^2 + (\varepsilon^4 + \sigma^4)\max_k L_{t_k}^2 + 2\left(\sigma^2 \vee \varepsilon^2\right)\|A\|\max_k L_{t_k}}, \tag{17}$$
*then,*
$$\mathcal{W}_2\left(\pi_{\mathrm{data}}, \mathcal{L}\left(\bar{X}_T^\theta\right)\right) \lesssim K_T \mathrm{e}^{-aT}\mathcal{W}_2\left(\pi_{\mathrm{data}} \otimes \pi_v, \pi_\infty\right) + \sigma^2 M + \sqrt{h}\, C_a(\varepsilon) .$$
*where $K_T = (1 + \max\{a+1; a(a+1)\}T)$ and*
$$C_a(\varepsilon) = \left(2\|A\|^4 B_\varepsilon + 4d(a^2\sigma^2 + \varepsilon)^2 \Lambda_\varepsilon^*(T)\right)h + 4d\left(\|A\|^2 + \sigma^4 \sup_{t \in [0,T]} L_{T-t}^2\right),$$

*with $B_\varepsilon$ and $\Lambda_\varepsilon^*(T)$ as in Lemma C.3.*

*Proof.* The error decomposition is the same as in Theorem 3.1. The full statement and proof for each error term is provided in Appendix C. □

This bound highlights the stabilizing role of the parameter $\varepsilon > 0$, which restores ellipticity in the dynamics. A key observation is that
$$\Sigma_\varepsilon \nabla^2 \log p_t \Sigma_\varepsilon \preccurlyeq -(\varepsilon^2 \wedge \sigma^2)\alpha_t \mathbf{I}_{2d} ,$$

which can be negative only for positive values of $\varepsilon$. In this sense, increasing $\varepsilon$ tends to enhance the contractive behavior of the dynamics, as also reflected by the admissible step-size condition (17). However, this effect is not purely beneficial: several terms in the discretization error scale with $\varepsilon^2$, illustrating that excessive noise injection may deteriorate the regularity of the process. Consequently, there is a trade-off in the choice of $\varepsilon$ to balance these competing effects. This trade-off is numerically illustrated in Section 4.

*Remark* 3.3. Finite second order moment is also necessary in this approach and is deduced from H2′ (see, *e.g.*, Lemma B.1, Gentiloni-Silveri and Ocello, 2025). Regarding H3, it is implied that the score approximation is now made for the true score function, not the modified one.

# 4   Experiments

We illustrate the effect of the regularization parameter $\varepsilon$ on the generation quality of CLDs on a simple yet challenging toy dataset. The regularization parameter $\varepsilon$ is chosen to be in $\{0, 0.1, 0.25, 0.5, 1\}$. Notably, $\varepsilon = 0$ corresponds to the vanilla CLD setting. Our source code is publicly available here[1].

**Evaluation metric.**   To assess the quality of the generated samples, directly computing the Wasserstein-2 distance is infeasible, as it requires solving a computationally expensive optimal transport problem. Instead, we approximate the $\mathcal{W}_2$-distance between the generated samples (with distribution $\hat{\pi}$) and the training samples (with distribution $\pi$data) using the sliced Wasserstein distance (Flamary et al., 2021). It is defined as $SW_2^2(\pi_{\text{data}}, \hat{\pi}) = \mathbb{E}_{\mathbf{u} \sim \mathcal{U}(\mathbb{S}^{d-1})}[\mathcal{W}_2^2(\mathbf{u}_{\#}\pi_{\text{data}}, \mathbf{u}_{\#}\hat{\pi})]$ where $\mathcal{U}(\mathbb{S}^{d-1})$ denotes the uniform distribution over the unit sphere and $\mathbf{u}_{\#}$ is the push-forward operator associated with $\mathbf{u}$. The expectation is approximated using the standard Monte Carlo method with 2000 projections and $\pi_{\text{data}}$ and $\hat{\pi}$ are replaced by their empirical distributions.

**Dataset.**   We evaluate the generation quality on the Funnel distribution, which is characterized by a strong imbalance in variance across dimensions and was previously used in Thin et al. (2021). To further illustrate our results, we extend the evaluation to two additional challenging toy datasets (Appendix E.5): MG-25 (a 25-mode, 100-dimensional Gaussian mixture) and Diamonds (a 2-dimensional Gaussian mixture with a diamond-shaped geometry).

**Hybrid Score Matching.**   Following the insights of Ho et al. (2020), the networks are trained to predict the noise (or rescaled noise) added during the forward process. When $\varepsilon = 0$, we use the positive weighting function proposed by Dockhorn et al. (2022) (see page 5, $\lambda(t) = \ell_t^{-2}$, in Dockhorn et al. (2022)). A similar reweighting, however, is not feasible for $\varepsilon \neq 0$ due to the matrix-valued nature of the objective function. Empirically, we observe that much of the training variance arises from the determinant computation involved in the $2 \times 2$ matrix inversions. To mitigate this, we set $\lambda(t) = \det(\Sigma_{0,t})^2$, which effectively stabilizes training. We parameterize the score network as $s_\theta(\overrightarrow{\mathbf{U}}_t, t) := -\Sigma_{0,t}^{-1} \alpha_\theta(\overrightarrow{\mathbf{U}}_t, t)$ so that the hybrid score matching objective for $\varepsilon > 0$ is given by

$$\mathcal{L}_{(\text{HSM})^\varepsilon}(\theta) = \mathbb{E}\left[\det(\Sigma_{0,t})^2 \left\| \Sigma_{0,t}^{-1}\left(s_\theta\left(\tau, \overrightarrow{X}_\tau\right) - \Sigma_{0,t}^{1/2}G_{2d}\right)\right\|^2\right], \tag{18}$$

where $G_{2d}$ denotes a $2d$-dimensional standard Gaussian noise.

**Model, Training and Generation.**   All score networks share the same architecture: a fully connected neural network with three hidden layers of width 512 (see Figure 3). Training is performed using the Adam optimizer to minimize the hybrid score matching objective in (18), with a learning rate of $10^{-4}$ over 2000 epochs. The training set consists of 50 000 samples. For evaluation, we generate 50 000 samples using the Euler–Maruyama discretization scheme with $N = 1000$ steps and compare them against a test set of 50 000 samples. Both training and generation are independently repeated five times. The training (Algorithm 1) and sampling (Algorithm 2) procedures are provided in Appendix E.1.

**Effect of the regularization parameter.**   Figure 1 illustrates the Wasserstein error for different values of the regularization parameter $\varepsilon \in \{0, 0.1, 0.25, 0.5, 1\}$ and drift coefficient $a \in \{0.1, 0.25, 0.5, 1, 2\}$. Across all values of $a$, introducing a small regularization parameter $\varepsilon$ notably improves generation quality, even though the score network in the regularized case must predict a vector twice as long as in the non-regularized one. Moreover, regularization consistently reduces the

---

[1] https://github.com/SobihanSurendran/CLD

variance across runs. For smaller values of $a$, the error increases sharply when $\varepsilon = 0$ and also for large $\varepsilon$ values. In contrast, for moderate values of $a$, the error becomes negligible, with $\varepsilon \in [0.1, 0.5]$ yielding slightly better performance than the other settings. It is worth noting that our experimental configuration closely follows that of Dockhorn et al. (2022), using $\sigma = \sqrt{2}$, $a = 2$, and in particular $\varepsilon = 0$. This observation justifies their choice of $a = 2$ for the vanilla CLD.

**Effect of $\varepsilon$ in controlled settings.**

Varying $\varepsilon$ modifies both the stationary distribution and the noise schedule—two factors known to strongly influence performance (Guo et al., 2023; Chen et al., 2023; Strasman et al., 2025)—it is important to control for these effects. To mitigate this confounding factor, one can fix the stationary distribution of the base case to $\mathcal{N}(0_{2d}, \mathbf{I}_{2d})$ and maintain comparable noise levels in the position and velocity spaces by setting $a(\varepsilon) = 1 - \varepsilon^2/2$ and $\sigma(\varepsilon) = \sqrt{4 + \varepsilon^2}$. This choice ensures that the stationary distribution remains close to $\mathcal{N}(0_{2d}, \mathbf{I}_{2d})$ for small values of $\varepsilon$. Although this adjustment becomes less accurate for larger $\varepsilon$, there is no practical limitation preventing the use of higher regularization values.

Figure 2 still shows an improvement in generation quality for small regularization parameters $\varepsilon$. To confirm that this effect is not tied to the discretization method, we reproduce the experiments using a Leapfrog integrator. As expected,

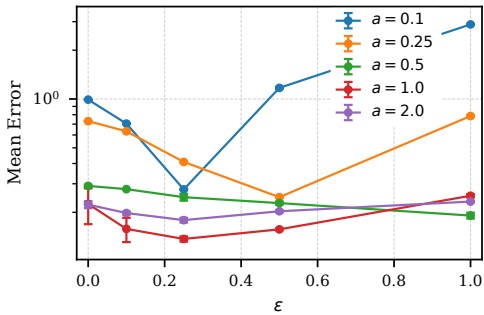

Figure 1: Mean $\mathcal{W}_2$ distance over 5 repetitions between the test set and generated samples on Funnel distribution in dimension 100. Error bars represent $\pm$ one standard deviation.

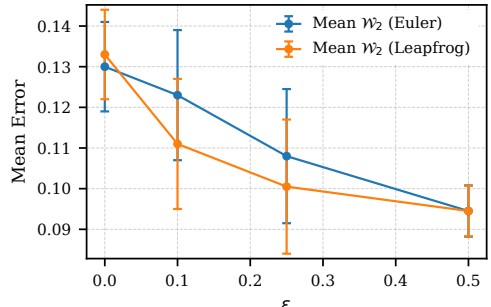

Figure 2: Mean $\mathcal{W}_2$ distance over 5 repetitions between the test set and generated samples on Funnel distribution with $a(\varepsilon) = 1 - \varepsilon^2/2$ and $\sigma(\varepsilon) = \sqrt{4 + \varepsilon^2}$.

the Leapfrog scheme outperforms Euler–Maruyama, yet the relative benefit of regularization persists. Finally, we emphasize that our objective is not to conduct an extensive numerical comparison of integrators or training strategies, but rather to highlight the potential of introducing controlled regularization within the CLD framework—a direction theoretically supported by Theorem 3.2.

## 5 Discussion

In this paper, we present the first theoretical analysis of the sampling error of CLDs in the Wasserstein metric under weaker assumptions than those previously used in the literature. Our results show that CLD-based samplers can achieve comparable convergence rates while effectively leveraging the structure of the extended space. We further analyze a generalized dynamic that extends classical CLDs by introducing a smoothness-controlling hyperparameter that regulates the noise on the data coordinate. This parameter provides more precise control over the regularity of sample paths and plays a central role in the discretization error analysis. Both theoretical and empirical results suggest that appropriately tuning this parameter leads to improved sampling quality and stability. Overall, our work offers both theoretical insights and practical guidance for CLD methods in generative modeling, particularly in scenarios where standard assumptions may not hold. Several promising directions remain for future research. Replacing the Euler discretization scheme with a higher-order method—such as the Leapfrog integrator, which is specifically designed for CLD-based dynamics—could further enhance sampling performance. Analyzing such schemes would likely yield sharper convergence rates consistent with the numerical results. Moreover, developing denoiser architectures specifically tailored to the extended space represents another promising avenue for applied research, potentially leading to tighter bounds on the approximation error.

## Acknowledgements

The PhD of Sobihan Surendran was funded by the Paris Region PhD Fellowship Program of Région Ile-de-France. The work of Antonio Ocello was funded by the European Union (ERC-2022-SYG-OCEAN-101071601). Views and opinions expressed are however those of the author only and do not necessarily reflect those of the European Union or the European Research Council Executive Agency. Neither the European Union nor the granting authority can be held responsible for them. We would also like to thank SCAI (Sorbonne Center for Artificial Intelligence) for providing the computing clusters.

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

# Supplementary Material for "Wasserstein Convergence of Critically Damped Langevin Diffusions"

## Table of Contents

# A  Forward process of Critically-Damped dynamics

In this section, we establish several mathematical properties of the forward processes:

$$\mathrm{d}\overrightarrow{\mathbf{U}}_t = A\overrightarrow{\mathbf{U}}_t\mathrm{d}t + \Sigma_\varepsilon \mathrm{d}B_t \,, \qquad \overrightarrow{\mathbf{U}}_0 \sim \pi_{\mathrm{data}} \otimes \pi_v \,,$$

as defined in (4) with $\epsilon = 0$ or in (9) with $\epsilon \geq 0$. These results will be used throughout our subsequent analysis.

**Lemma A.1.** *Let $A$ be the matrix defined in* (5)*, then*

$$A = \left( \begin{pmatrix} -a & -1 \\ 1 & 0 \end{pmatrix} \times \begin{pmatrix} -a & 1 \\ 0 & -a \end{pmatrix} \times \begin{pmatrix} 0 & 1 \\ -1 & -a \end{pmatrix} \right) \otimes \mathbf{I}_d$$

*so that*

$$\mathrm{e}^{tA} = \mathrm{e}^{-ta} \begin{pmatrix} 1 + at & a^2 t \\ -t & 1 - at \end{pmatrix} \otimes \mathbf{I}_d \,, \tag{19}$$

*and*

$$\|\mathrm{e}^{tA}\| \leq \|\mathrm{e}^{tA}\|_1^{1/2}\|\mathrm{e}^{tA}\|_\infty^{1/2} \leq (1 + \max\{a + 1; a(a + 1)\}t)\,\mathrm{e}^{-ta}$$
$$\leq \left(1 + (a + 1)^2 t\right)\mathrm{e}^{-ta} \,.$$

*Proof.* The Jordan matrix decomposition of $A$ when $d = 1$ is given by,

$$A_1 = \begin{pmatrix} 0 & a^2 \\ -1 & -2a \end{pmatrix} = \begin{pmatrix} -a & -1 \\ 1 & 0 \end{pmatrix} \times \begin{pmatrix} -a & 1 \\ 0 & -a \end{pmatrix} \times \begin{pmatrix} 0 & 1 \\ -1 & -a \end{pmatrix} \,.$$

We can use this decomposition to obtain a matrix factorization in any dimension. As for all $k \in \mathbb{N}$, $A^k = (A_1^k \otimes \mathbf{I}_d)$,

$$\mathrm{e}^{tA} = \sum_{k=0}^{\infty} \frac{t^k}{k!} \left(A_1^k \otimes \mathbf{I}_d\right) = \left(\sum_{k=0}^{\infty} \frac{t^k}{k!} A_1^k\right) \otimes \mathbf{I}_d = \mathrm{e}^{tA_1} \otimes \mathbf{I}_d \,.$$

Finally, we deduce an upper bound to the spectral norm of $\mathrm{e}^{tA}$, as

$$\|\mathrm{e}^{tA}\|_1 \leq \mathrm{e}^{-ta}\max\{(1 + (a + 1)t; 1 + a(a + 1)t\} \,,$$

and

$$\|\mathrm{e}^{tA}\|_\infty \leq \mathrm{e}^{-ta}\max\{1 + a(a + 1)t; 1 + (a + 1)t\} \,.$$

Then,

$$\|\mathrm{e}^{tA}\| \leq \|\mathrm{e}^{tA}\|_1^{1/2}\|\mathrm{e}^{tA}\|_\infty^{1/2} \leq \mathrm{e}^{-ta}\left(1 + \max\{a + 1; a(a + 1)\}t\right) \,,$$

which concludes the proof.

$\square$

**Lemma A.2.** *Let $(\overrightarrow{\mathbf{U}}_t)_{t\in[0,T]}$ be a solution to the forward process* (9) *with initial condition*

$$\overrightarrow{\mathbf{U}}_0 \sim \pi_{\mathrm{data}} \otimes \pi_v \,,$$

*where $\pi_v$ is a probability distribution on $(\mathbb{R}^d, \mathcal{B}(\mathbb{R}^d))$. Then, the conditional law of $\overrightarrow{\mathbf{U}}_t$ given $\overrightarrow{\mathbf{U}}_0$, is Gaussian with mean $\mu_{t|0}$ and covariance $\Sigma_{0,t}$ defined by*

$$\mu_{t|0} := \mathrm{e}^{tA}\overrightarrow{\mathbf{U}}_0 \,, \qquad \Sigma_{0,t} := \Sigma_\infty - \mathrm{e}^{tA}\Sigma_\infty (\mathrm{e}^{tA})^\top, \tag{20}$$

*with*

$$\Sigma_\infty := \frac{1}{4} \begin{pmatrix} 5\varepsilon^2 a^{-1} + a\sigma^2 & -2\varepsilon^2 a^{-2} \\ -2\varepsilon^2 a^{-2} & (\varepsilon^2 + a^2\sigma^2)a^{-3} \end{pmatrix} \otimes \mathbf{I}_d \,. \tag{21}$$

*The result still holds when the forward process is defined as in* (4) *by setting $\varepsilon = 0$.*

*Proof.* Recall that the forward process $(\overrightarrow{\mathbf{U}}_t)_{t\in[0,T]}$ is solution to,

$$\mathrm{d}\overrightarrow{\mathbf{U}}_t = A\overrightarrow{\mathbf{U}}_t\mathrm{d}t + \Sigma_\varepsilon\mathrm{d}B_t. \tag{22}$$

With initial condition $\overrightarrow{\mathbf{U}}_0 \sim \pi_{\mathrm{data}} \otimes \pi_v$, we have

$$\overrightarrow{\mathbf{U}}_t = \mathrm{e}^{tA}\overrightarrow{\mathbf{U}}_0 + \int_0^t \mathrm{e}^{(t-s)A}\Sigma_\varepsilon\mathrm{d}B_s.$$

This means that the law of $\overrightarrow{\mathbf{U}}_t$, conditional to the initial condition $\overrightarrow{\mathbf{U}}_0$ is Gaussian with mean

$$\mu_{t|0} := \mathbb{E}\left[\overrightarrow{\mathbf{U}}_t\right] = \mathrm{e}^{tA}\overrightarrow{\mathbf{U}}_0,$$

and covariance

$$\begin{aligned}
\Sigma_{0,t} := \mathrm{Cov}\left(\overrightarrow{\mathbf{U}}_t\right) &= \int_0^t \mathrm{e}^{(t-s)A}\Sigma_\varepsilon^2(\mathrm{e}^{(t-s)A})^\top\mathrm{d}s \\
&= \int_0^t \mathrm{e}^{(t-s)A}\Sigma_\varepsilon^2(\mathrm{e}^{(t-s)A})^\top\mathrm{d}s \\
&= \left(\int_0^t \mathrm{e}^{(t-s)A_1}\Sigma_\varepsilon^2(\mathrm{e}^{(t-s)A_1})^\top\mathrm{d}s\right)\otimes\mathbf{I}_d.
\end{aligned}$$

Using Lemma A.1, for $\delta \geq 0$,

$$\mathrm{e}^{\delta A_1}\Sigma_\varepsilon^2\mathrm{e}^{\delta A_1^\top} = \mathrm{e}^{-2a\delta}\begin{pmatrix} a^4\sigma^2\delta^2 + \varepsilon^2(1+a\delta)^2 & \delta\left(a^2\sigma^2(1-a\delta) - \varepsilon^2(1+a\delta)\right) \\ \delta\left(a^2\sigma^2(1-a\delta) - \varepsilon^2(1+a\delta)\right) & \sigma^2(1-a\delta)^2 + \delta^2\varepsilon^2 \end{pmatrix}.$$

Hence, a straightforward computation provides with $\alpha_t = (-(5 + 2at(3 + at))\varepsilon^2 - a^2(1 + 2at(1 + at))\sigma^2)a^{-1}$ and $\gamma_t = 2((\varepsilon + at\varepsilon)^2 + a^4t^2\sigma^2)a^{-2}$,

$$\begin{aligned}
\Sigma_{0,t} =& \frac{1}{4}\begin{pmatrix} 5\varepsilon^2 a^{-1} + a\sigma^2 & -2\varepsilon^2 a^{-2} \\ -2\varepsilon^2 a^{-2} & (\varepsilon^2 + a^2\sigma^2)a^{-3} \end{pmatrix} \\
&+ \frac{\mathrm{e}^{-2at}}{4}\begin{pmatrix} \alpha_t & \gamma_t \\ \gamma_t & (-(1 + 2at(1 + at))\varepsilon^2 - a^2(1 + 2at(-1 + at))\sigma^2)a^{-3} \end{pmatrix} \\
=& \Sigma_\infty - \mathrm{e}^{tA}\Sigma_\infty(\mathrm{e}^{tA})^\top,
\end{aligned} \tag{23}$$

where we used that,

$$\mathrm{e}^{tA}\Sigma_\infty(\mathrm{e}^{tA})^\top = \int_0^\infty \mathrm{e}^{(t+s)A}\Sigma^2\left(\mathrm{e}^{(t+s)A}\right)^\top\mathrm{d}s = \int_t^\infty \mathrm{e}^{\delta A}\Sigma^2\left(\mathrm{e}^{\delta A}\right)^\top\mathrm{d}\delta = \Sigma_\infty - \Sigma_{0,t}.$$

$\square$

**Lemma A.3.** *The covariance matrix $\Sigma_{0,t}$ defined in (20) satisfies, for all $\varepsilon > 0$,*

$$\lambda_{\min}(\Sigma_{0,t}) \geq \max\left\{\frac{\sigma^2}{4}\min\{a, 1/a\} - \left(\frac{\sigma^2}{4}\max\{a, 1/a\} + \frac{5\varepsilon^2}{4a}\right)\mathrm{e}^{-2at},\right.$$

$$\left.\min\{\varepsilon^2, \sigma^2\}\frac{1 - \mathrm{e}^{-2at}}{2a\left(1 + (a+1)^2t\right)^2}\right\},$$

$$\lambda_{\max}(\Sigma_{0,t}) \leq \frac{\sigma^2}{4}\max\{a, 1/a\} + \frac{5\varepsilon^2}{4a}.$$

*Proof.* First, consider the following decomposition of $\Sigma_\infty$ defined in (21)

$$\Sigma_\infty = \frac{1}{4}\begin{pmatrix} a\sigma^2 & 0 \\ 0 & \sigma^2 a^{-1} \end{pmatrix} + \frac{\varepsilon^2}{4a^3}\begin{pmatrix} 5a^2 & -2a \\ -2a & 1 \end{pmatrix} =: \frac{1}{4}\begin{pmatrix} a\sigma^2 & 0 \\ 0 & \sigma^2 a^{-1} \end{pmatrix} + E_\varepsilon.$$

Since $E_\varepsilon$ is positive definite, its trace and determinant are positive, then

$$\lambda_{\min}(\Sigma_\infty) \geq \frac{1}{4}\lambda_{\min}\left(\begin{pmatrix} a\sigma^2 & 0 \\ 0 & \sigma^2 a^{-1} \end{pmatrix}\right) = \frac{\sigma^2}{4}\min\{a, 1/a\}\,,$$

$$\lambda_{\max}(\Sigma_\infty) \leq \frac{1}{4}\lambda_{\max}\left(\begin{pmatrix} a\sigma^2 & 0 \\ 0 & \sigma^2 a^{-1} \end{pmatrix}\right) + \lambda_{\max}(E_\varepsilon) \leq \frac{\sigma^2}{4}\max\{a, 1/a\} + \frac{5\varepsilon^2}{4a}\,. \qquad (24)$$

Using that $\Sigma_{0,t} = \Sigma_\infty - \mathrm{e}^{tA}\Sigma_\infty \mathrm{e}^{tA^\top}$ together with Weyl's inequality we have that

$$\lambda_{\min}(\Sigma_{0,t}) \geq \lambda_{\min}(\Sigma_\infty) - \lambda_{\max}\left(\mathrm{e}^{tA}\Sigma_\infty \mathrm{e}^{tA^\top}\right)\,.$$

Note that, as $\Sigma_\infty$ is positive semidefinite,

$$\lambda_{\max}\left(\mathrm{e}^{tA}\Sigma_\infty \mathrm{e}^{tA^\top}\right) = \lambda_{\max}\left(\mathrm{e}^{tA}\Sigma_\infty^{1/2}\right)^2 \leq \lambda_{\max}\left(\mathrm{e}^{tA}\right)^2 \lambda_{\max}(\Sigma_\infty) \leq \mathrm{e}^{-2at}\lambda_{\max}(\Sigma_\infty)\,.$$

On the other hand, using that $\Sigma_{0,t} = \int_0^t \mathrm{e}^{sA}\Sigma_\varepsilon^2 \mathrm{e}^{sA^\top}\mathrm{d}s$, yields

$$\Sigma_{0,t} \succcurlyeq \min\{\varepsilon^2, \sigma^2\}\int_0^t \mathrm{e}^{sA}\mathrm{e}^{sA^\top}\mathrm{d}s\,,$$

therefore,

$$\lambda_{\min}(\Sigma_{0,t}) \geq \min\{\varepsilon^2, \sigma^2\}\int_0^t \lambda_{\min}\left(\mathrm{e}^{sA}\mathrm{e}^{sA^\top}\right)\mathrm{d}s$$

$$\geq \min\{\varepsilon^2, \sigma^2\}\int_0^t \frac{\mathrm{e}^{-2as}}{(1 + (a+1)^2 s)^2}\mathrm{d}s,$$

$$\geq \min\{\varepsilon^2, \sigma^2\}\frac{1 - \mathrm{e}^{-2at}}{2a(1 + (a+1)^2 t)^2}\,,$$

which gives the other lower bound of $\lambda_{\min}.(\Sigma_{0,t})$

To obtain the bound on $\lambda_{\max}(\Sigma_{0,t})$, it is enough to note that $\Sigma_{0,t} \preccurlyeq \Sigma_\infty$. $\qquad\square$

**Lemma A.4** (Forward process $\mathcal{W}_2$-contraction). *The forward process, defined as in* (9)*, is contractive for the $\mathcal{W}_2$ distance. In particular, it holds that*

$$\mathcal{W}_2\left(\mathcal{L}(\overrightarrow{\mathbf{U}}_T), \pi_\infty\right) \leq K_T e^{-aT}\mathcal{W}_2\left(\pi_{\mathrm{data}} \otimes \pi_v, \pi_\infty\right)\,,$$

*where $\pi_\infty$ is the stationary distribution of* (9) *as defined in Lemma A.2 and*

$$K_T := (1 + \max\{a + 1; a(a+1)\}T)\,.$$

*Proof.* Let $u = (x, v) \in \mathbb{R}^{2d}$ (resp. $\bar{u} = (\bar{x}, \bar{v}) \in \mathbb{R}^{2d}$) and denote by $(\overrightarrow{\mathbf{U}}_t^u)_{t \in [0,T]}$ (resp. $(\overrightarrow{\mathbf{U}}_t^{\bar{u}})_{t \in [0,T]}$) the solution of (9), with initial condition $\overrightarrow{\mathbf{U}}_0^u = u$ (resp. $\overrightarrow{\mathbf{U}}_0^{\bar{u}} = \bar{u}$). By Itô's lemma,

$$\mathrm{d}\left(e^{-tA}\overrightarrow{\mathbf{U}}^{x,v}\right) = e^{-tA}\Sigma_\varepsilon \mathrm{d}B_t\,.$$

Using a synchronous coupling for $(\overrightarrow{\mathbf{U}}_t^u)_{t \in [0,T]}$ and $(\overrightarrow{\mathbf{U}}_t^{\bar{u}})_{t \in [0,T]}$, we have that

$$\overrightarrow{\mathbf{U}}_t^u - \overrightarrow{\mathbf{U}}_t^{\bar{u}} = \mathrm{e}^{tA}(u - \bar{u})\,.$$

By definition of the Wasserstein-2 distance $\mathcal{W}_2(\mathcal{L}(\overrightarrow{\mathbf{U}}_t^u), \mathcal{L}(\overrightarrow{\mathbf{U}}_t^{\bar{u}})) \leq \|\overrightarrow{\mathbf{U}}_t^u - \overrightarrow{\mathbf{U}}_t^{\bar{u}}\|_{L_2}$. Then, by Lemma A.1,

$$\left\|\overrightarrow{\mathbf{U}}_t^u - \overrightarrow{\mathbf{U}}_t^{\bar{u}}\right\|_{L_2} \leq \left\|\mathrm{e}^{tA}\right\| \|u - \bar{u}\|_{L_2} \leq K_t \mathrm{e}^{-ta} \|u - \bar{u}\|_{L_2}\,, \qquad (25)$$

with

$$K_t := (1 + \max\{a + 1; a(a+1)\}t)\,.$$

Finally, assume that $\bar{u} \sim \pi_\infty$, $u \sim \pi_{\mathrm{data}} \otimes \pi_v$ and fix any coupling $\gamma \in \Pi(\pi_{\mathrm{data}} \otimes \pi_v, \pi_\infty)$. Using that $\pi_\infty$ is stationary distribution of $\overrightarrow{\mathbf{U}}_t$ and taking the infimum over $\gamma \in \Pi(\pi_{\mathrm{data}} \otimes \pi_v, \pi_\infty)$ yields,

$$\mathcal{W}_2\left(\mathcal{L}\left(\overrightarrow{\mathbf{U}}_T\right), \pi_\infty\right) \leq K_T e^{-aT}\mathcal{W}_2\left(\pi_{\mathrm{data}} \otimes \pi_v, \pi_\infty\right)\,,$$

which finishes the proof. $\qquad\square$

# B Proof of Theorem 3.1

In this section we prove Theorem 3.1. We use notations from (12) (resp. (13)) for the continuous time interpolation of the discretized backward with modified score function $\bar{U}_t$ (resp. for the continuous time interpolation of the discretized backward with approximated modified score function $\bar{U}_t^\theta$). We first establish the propagation of Lipschitz regularity, followed by the proof of Theorem 3.1. To do so, we decompose the generation error as the sum of the discretization error (Lemma B.2), the approximation error (Lemma B.3), and the mixing time error (Lemma B.4).

## B.1 Propagation of the regularity assumptions

**Proposition B.1.** *Assume that Assumption H2 holds. Then, for all $t > 0$, $\Sigma_\varepsilon^2 \nabla \log p_t$ (resp. $\Sigma_\varepsilon^2 \nabla \log \tilde{p}_t$) is $L_t$-Lipschitz (resp. $\tilde{L}_t$-Lipschitz): for all $u \in \mathbb{R}^{2d}$,*

$$\left\| \Sigma_\varepsilon^2 \nabla^2 \log p_t(u) \right\| \le L_t \,.$$

*Moreover, there exists a constant $C > 0$ such that for all $u \in \mathbb{R}^{2d}$,*

$$\left\| \Sigma_\varepsilon^2 \nabla^2 \log \tilde{p}_t(u) \right\| \le \tilde{L}_t \le C \left( 1 + \frac{1}{\sqrt{t}} \right) e^{-2at} \,. \tag{26}$$

*Proof. Step 1: Lower bound on $\nabla^2 \log p_t$.* Recall the following equality in law given by the modified kinetic OU process (9)

$$\overrightarrow{\mathbf{U}}_t \overset{\mathcal{L}}{=} e^{tA} \overrightarrow{\mathbf{U}}_0 + \sqrt{\Sigma_{0,t}} G \,,$$

with $\overrightarrow{\mathbf{U}}_0 \sim \pi_{\text{data}} \otimes \pi_v$, $G \sim \mathcal{N}(0, \mathbf{I}_{2d})$, where $G$ and $\overrightarrow{\mathbf{U}}_0$ are independent, and $\Sigma_{0,t}$ is defined in (20). Writing $q_{t|0}$ the conditional density of $\overrightarrow{\mathbf{U}}_t$ given $\overrightarrow{\mathbf{U}}_0$, we have

$$
\begin{aligned}
p_t(u_t) &= \int_{\mathbb{R}^{2d}} p_0(u_0) q_{t|0}(u_t|u_0) \mathrm{d}u_0 \\
&= \int_{\mathbb{R}^{2d}} p_0(u_0) \det(2\pi \Sigma_{0,t})^{-1/2} \exp\left( -\frac{1}{2} \left( u_t - e^{tA} u_0 \right)^\top \Sigma_{0,t}^{-1} \left( u_t - e^{tA} u_0 \right) \right) \mathrm{d}u_0 \\
&= \det\left( e^{-tA} \right) \int_{\mathbb{R}^{2d}} p_0\left( e^{-tA} z \right) \det(2\pi \Sigma_{0,t})^{-1/2} \exp\left( -\frac{1}{2} \left( u_t - z \right)^\top \Sigma_{0,t}^{-1} \left( u_t - z \right) \right) \mathrm{d}z \,.
\end{aligned}
$$

As also observed in Saumard and Wellner (Proposition 7.1, 2014), we get

$$
\begin{aligned}
\nabla^2 \log p_t(u) &= \text{Var}(\nabla \phi_{0,t}(Y_0)|Y_0 + Y_1 = u) - \mathbb{E}[\nabla^2 \phi_{0,t}(Y_0)|Y_0 + Y_1 = u] \tag{27} \\
&= \text{Var}(\nabla \phi_{1,t}(Y_1)|Y_0 + Y_1 = u) - \mathbb{E}[\nabla^2 \phi_{1,t}(Y_1)|Y_0 + Y_1 = u] \,,
\end{aligned}
$$

for $Y_0 = e^{tA} \overrightarrow{\mathbf{U}}_0$ and $Y_1 = \sqrt{\Sigma_{0,t}} G$ and for $\phi_{0,t}$ and $\phi_{1,t}$ such that for all $u \in \mathbb{R}^{2d}$,

$$
\begin{aligned}
e^{-\phi_{0,t}(u)} &:= \det\left( e^{-tA} \right) p_0\left( e^{-tA} u \right) , \\
e^{-\phi_{1,t}(u)} &:= \det(2\pi \Sigma_{0,t})^{-1/2} \exp\left( -\frac{1}{2} u^\top \Sigma_{0,t}^{-1} u \right) \,.
\end{aligned}
$$

This implies that

$$
\begin{aligned}
\nabla^2 \log p_t(u) &\succcurlyeq -\mathbb{E}[\nabla^2 \phi_{0,t}(Y_0)|Y_0 + Y_1 = u] \,, \\
\nabla^2 \log p_t(u) &\succcurlyeq -\mathbb{E}[\nabla^2 \phi_{1,t}(Y_1)|Y_0 + Y_1 = u] \,.
\end{aligned} \tag{28}
$$

Note that for all $u \in \mathbb{R}^{2d}$,

$$
\begin{aligned}
\nabla^2 \phi_{0,t}(u) &= -e^{-tA^\top} \nabla^2 \log p_0\left( e^{-tA} u \right) e^{-tA} \,, \\
\nabla^2 \phi_{1,t}(u) &= \Sigma_{0,t}^{-1} \,.
\end{aligned}
$$

From (Bouchut et al., 2005, Lemma 2.2) together with (14), we get that the one-sided Lipschitz assumption entails the following inequality over the Hessian of the log-density, since $\log p_0(u) = \log \pi_{\text{data}}(x) + \log p_v(v)$,

$$\nabla^2\left(-\log p_0\right)(u) = \begin{pmatrix} -\nabla^2 \log p_{\text{data}}(x) & 0 \\ 0 & -\nabla^2 \log p_v(v) \end{pmatrix}$$

$$= \begin{pmatrix} -\nabla^2 \log p_{\text{data}}(x) & 0 \\ 0 & v^{-2}\mathbf{I}_d \end{pmatrix} \preccurlyeq \max\left\{L_0, \frac{1}{v^2}\right\}\mathbf{I}_{2d}.$$

Therefore, for $t > 0$, from (28), we get

$$\nabla^2 \log p_t(u) \succcurlyeq -\mathfrak{h}_t\mathbf{I}_{2d},$$

where $\mathfrak{h}_t = \min\left\{\left\|e^{-tA}\right\|^2 \max\left\{L_0, v^{-2}\right\}; \left\|\Sigma_{0,t}^{-1}\right\|\right\}$.

*Bound on $\mathfrak{h}_t$.* By Lemma A.1, we have that $\left\|e^{-tA}\right\|^2 \leq \left(1 + (a+1)^2 t\right)^2 e^{2ta}$. Moreover, from Lemma A.3, it follows that

$$\left\|\Sigma_{0,t}^{-1}\right\| = \frac{1}{\lambda_{\min}(\Sigma_{0,t})} \leq \frac{1}{\left\lfloor \lambda_{\min}(\Sigma_\infty) - \lambda_{\max}(e^{tA}\Sigma_\infty e^{tA^\top})\right\rfloor_+},$$

with $\lfloor\cdot\rfloor_+$ denoting the positive part of a real number. Therefore,

$$\left\|\Sigma_{0,t}^{-1}\right\| \leq \frac{4}{\left\lfloor \sigma^2 \min\{a, 1/a\} - (\sigma^2 \max\{a, 1/a\} + 5\varepsilon^2 a^{-1})\, e^{-2at}\right\rfloor_+} =: \mathfrak{h}_{2,t}.$$

Combining the two previous bounds, we obtain

$$\mathfrak{h}_t \leq \min\left\{\mathfrak{h}_{1,t}; \mathfrak{h}_{2,t}\right\}, \tag{29}$$

where $\mathfrak{h}_{1,t} := \left(1 + (a+1)^2 t\right)^2 e^{2ta} \max\left\{L_0, v^{-2}\right\}$.

*Step 2: Upper bound on $\nabla^2 \log p_t$.* We first express the conditional density of $\overrightarrow{\mathbf{U}}_0$ given $\overrightarrow{\mathbf{U}}_t$ as follows

$$q_{t|0}((x_0, v_0)^\top | u_t) \propto \left(e^{-V(x_0)-H(x_0)} \otimes \mathcal{N}(v_0; 0_d, v^2\mathbf{I}_d)\right)\mathcal{N}(u_t; e^{tA}(x_0, v_0)^\top, \Sigma_{0,t}). \tag{30}$$

First, we consider the log-concave part of the above distribution,

$$\nu_t \propto \left(e^{-V(x_0)} \otimes \mathcal{N}(v_0; 0_d, v^2\mathbf{I}_d)\right)\mathcal{N}(u_t; e^{tA}(x_0, v_0)^\top, \Sigma_{0,t}). \tag{31}$$

Since $\nabla^2 V(x) \succcurlyeq \alpha\mathbf{I}_d$ for all $x \in \mathbb{R}^d$, we obtain

$$\nabla^2\left(-\log \nu_t\right) \succcurlyeq e^{-tA}\begin{pmatrix} \alpha\mathbf{I}_d & 0 \\ 0 & \frac{1}{v^2}\mathbf{I}_d \end{pmatrix}e^{-tA^\top} + \Sigma_{0,t}^{-1}.$$

Therefore, by Brascamp–Lieb inequality (Brascamp and Lieb, 1976),

$$\text{Cov}(\nu_t) \preccurlyeq \left(e^{-tA}\begin{pmatrix} \alpha\mathbf{I}_d & 0 \\ 0 & \frac{1}{v^2}\mathbf{I}_d \end{pmatrix}e^{-tA^\top} + \Sigma_{0,t}^{-1}\right)^{-1}.$$

Using the identity $\Sigma_{0,t} = \Sigma_\infty - e^{tA}\Sigma_\infty e^{tA^T}$ given in Lemma A.2, we now expand $\Sigma_{0,t}$ at zero as

$$\Sigma_{0,t} = t\begin{pmatrix} \varepsilon^2 & 0 \\ 0 & \sigma^2 \end{pmatrix} + \mathcal{O}(t^2),$$

which implies that

$$\Sigma_{0,t}^{-1} = \frac{1}{t}\underbrace{\begin{pmatrix} 1/\varepsilon^2 & 0 \\ 0 & 1/\sigma^2 \end{pmatrix}}_{=\Sigma_\varepsilon^{-1}} + o\left(\tfrac{1}{t}\right).$$

Therefore, the covariance matrix near zero satisfies

$$\text{Cov}(\nu_t) \preccurlyeq \begin{pmatrix} \left(\alpha + \frac{1}{\varepsilon^2 t}\right)^{-1} & 0 \\ 0 & \left(\frac{1}{v^2} + \frac{1}{\sigma^2 t}\right)^{-1} \end{pmatrix} + o(t).$$

Next, the Lipschitz perturbation term, following Brigati and Pedrotti (2025), can be bounded as

$$\mathrm{Cov}(q_t(.|u_t)) \preccurlyeq \underbrace{\begin{pmatrix} \left(\frac{L}{\alpha+(\epsilon^2 t)^{-1}} + \sqrt{\frac{1}{\alpha+(\epsilon^2 t)^{-1}}}\right)^2 & 0 \\ 0 & \left(\frac{1}{v^2} + \frac{1}{\sigma^2 t}\right)^{-1} \end{pmatrix}}_{:=M_{\varepsilon,t}} + o(t)\,.$$

Using (27), we have

$$\nabla^2 \log p_t(u) = \Sigma_{0,t}^{-1} \mathrm{Cov}(q_t(.|u)) \Sigma_{0,t}^{-1} - \Sigma_{0,t}^{-1}\,, \tag{32}$$

so that

$$\nabla^2 \log p_t(u) = \left(\frac{1}{t}\Sigma_\varepsilon^{-1} + o\left(\frac{1}{t}\right)\right)(M_t + o(t))\left(\frac{1}{t}\Sigma_\varepsilon^{-1} + o\left(\frac{1}{t}\right)\right) - \left(\frac{1}{t}\Sigma_\varepsilon^{-1} + o\left(\frac{1}{t}\right)\right)$$

$$= \begin{pmatrix} \alpha_t & 0 \\ 0 & \beta_t \end{pmatrix} + o\left(\frac{1}{t}\right)\,,$$

with

$$|\alpha_t| \le \frac{L^2}{(\alpha\epsilon^2 t + 1)^2} + \frac{2L}{(\epsilon^2 t)^{1/2}(\alpha\epsilon^2 t + 1)^{3/2}} - \frac{\alpha}{\alpha\epsilon^2 t + 1}, \qquad \beta_t := -\frac{1}{\sigma^2 t + v^2}\,.$$

Consequently, for all $\epsilon > 0$, as $t \to 0^+$,

$$\begin{pmatrix} \frac{2L}{\epsilon\sqrt{t}} & 0 \\ 0 & \frac{1}{v^2} \end{pmatrix} + o\left(\frac{1}{\sqrt{t}}\right) \le \nabla^2 \log p_t(u) \le \begin{pmatrix} \frac{2L}{\epsilon\sqrt{t}} & 0 \\ 0 & -\frac{1}{v^2} \end{pmatrix} + o\left(\frac{1}{\sqrt{t}}\right)\,. \tag{33}$$

*Step 3: Uniform bound on $\nabla^2 \log p_t$.* We now analyze the structure of the minimum in the upper bound of $\mathfrak{h}_t$ in (29). We observe that the first term is increasing, equals $\max\{L_0, v^{-2}\}$ for $t \to 0$, and diverges as $t \to +\infty$. In contrast, the second term is decreasing: it diverges as $t \to 0$ and converges to $4/(\sigma^2 \min\{a, 1/a\})$ as $t \to +\infty$. Therefore, the minimum coincides with the first term, for $t \le T_{\mathrm{change}}$, and with the second term, for $t > T_{\mathrm{change}}$. Using (33), we obtain the following uniform bound, for all $\epsilon > 0$,

$$\left\| \nabla^2 \log p_t(u) \right\| \le \max\left\{ \mathfrak{h}_{1,T_{\mathrm{change}}}; Ct^{-1/2} \right\}\,, \quad \text{for } t > 0\,. \tag{34}$$

This bound is uniform in $\varepsilon > 0$, therefore, for $\varepsilon \to 0$, we have

$$\left\| \Sigma_\epsilon^2 \nabla^2 \log p_t(u) \right\| \le \max\left\{ \mathfrak{h}_{1,T_{\mathrm{change}}}; Ct^{-1/2} \right\}\,, \quad \text{for } t > 0\,. \tag{35}$$

Since $\tilde{p}_t = p_t/p_\infty$, and $p_\infty$ is the density of a centered Gaussian vector of variance $\Sigma_\infty$, we have

$$\nabla^2 \log \tilde{p}_t(u) = \nabla^2 \log p_t(u) + \Sigma_\infty^{-1}\,. \tag{36}$$

Therefore, the same bound as in (35) holds for the modified score.

*Step 4: Exponential decay of the modified score.* From (32), we have the following equality

$$\nabla^2 \log \tilde{p}_t(u) = \nabla^2 \log p_t(u) + \Sigma_\infty^{-1} = \Sigma_{0,t}^{-1} \mathrm{Cov}(q_t(.|u)) \Sigma_{0,t}^{-1} - \Sigma_{0,t}^{-1} + \Sigma_\infty^{-1}\,,$$

where $q_t$ is defined in (30). By applying Lemma D.9, together with the decomposition (20) and the positivity of the covariance, we obtain

$$\nabla^2 \log \tilde{p}_t(u) \succcurlyeq -\Sigma_{0,t}^{-1}\left( e^{tA} \Sigma_\infty (e^{tA})^\top \right) \Sigma_\infty^{-1}\,.$$

Since $\Sigma_{0,t} = \Sigma_\infty + \mathcal{O}(e^{-2at})$ as $t \to \infty$, there exists a constant $C > 0$ such that

$$\left\| \Sigma_{0,t}^{-1}\left( e^{tA} \Sigma_\infty (e^{tA})^\top \right) \Sigma_\infty^{-1} \right\| \le Ce^{-2at}$$

On the other hand, using the fact that $\Sigma_{0,t}^{-1} \succcurlyeq \Sigma_\infty^{-1}$ (see (20)), we get

$$\nabla^2 \log \tilde{p}_t(u) \preccurlyeq \Sigma_{0,t}^{-1} \mathrm{Cov}(q_t(.|u)) \Sigma_{0,t}^{-1}\,.$$

Following the same steps as in the derivation of the upper bound on $\nabla^2 \log p_t$ (step 2), we obtain

$$\nabla^2\left(-\log\nu_t\right) \succcurlyeq \mathrm{e}^{-tA}\begin{pmatrix}\alpha\mathbf{I}_d & 0 \\ 0 & \frac{1}{v^2}\mathbf{I}_d\end{pmatrix}\mathrm{e}^{-tA^\top} + \Sigma_{0,t}^{-1} \succcurlyeq \mathrm{e}^{-tA}\begin{pmatrix}\alpha\mathbf{I}_d & 0 \\ 0 & \frac{1}{v^2}\mathbf{I}_d\end{pmatrix}\mathrm{e}^{-tA^\top},$$

where $\nu_t$ is defined in (30). By the Brascamp–Lieb inequality, this implies that $\mathrm{Cov}(\nu_t) = \mathcal{O}(e^{-2at})$. Next, similarly to "step 2", for the term involving the Lipschitz perturbation, and following Brigati and Pedrotti (2025, Theorem 1.3), we have

$$\mathrm{Cov}(q_t(.|u)) \preccurlyeq \left(LCe^{-2at} + \sqrt{Ce^{-2at}}\right)^2\mathbf{I}_d.$$

Therefore, there exist a universal constant $C > 0$ and a finite time $T_{\mathrm{change}} > 0$ such that, for all $t \geq T_{\mathrm{change}}$,

$$\left\|\nabla^2\log\tilde{p}_t(u)\right\| \leq Ce^{-2at}.$$

This implies that the modified score function is $\tilde{L}_t$-Lipschitz, with $\tilde{L}_t$ defined as

$$\tilde{L}_t := \begin{cases}\max\left\{\mathfrak{h}_{1,T_{\mathrm{change}}}; Ct^{-1/2}\right\} + \max\{a, 1/a\}, & \text{for } t \in (0, T_{\mathrm{change}}], \\ Ce^{-2at}, & \text{for } t \in (T_{\mathrm{change}}, +\infty),\end{cases}$$

which concludes the proof.

$\square$

## B.2 Proofs of the main results

**Lemma B.2** (Discretization error). *Assume that H1 and H2 hold. Then, for all $\eta > 0$ and all $h > 0$, there exists a constant $C > 0$ such that*

$$\mathcal{W}_2\left(\mathcal{L}(\overleftarrow{\mathbf{U}}_T), \mathcal{L}(\bar{\mathbf{U}}_T)\right) \leq \sqrt{h}C\sqrt{\left(h\|A\|^2B^2 + \|\Sigma_\epsilon\|^2\left(d + \mathcal{I}(\pi_{\mathrm{data}}\otimes\pi_v|\pi_\infty)\right)\right)\frac{\mathrm{e}^{Ca^{-1}}}{a-\eta}}, \quad (37)$$

*with*

$$B := \max_{t\in[0,T]}\left(1 + (a+1)^2(T-t)\right)^2\mathrm{e}^{-2a(T-t)}\left\|\overrightarrow{\mathbf{U}}_0\right\|_{L_2}^2 + \frac{d}{2}\left(\sigma^2\max\{a, 1/a\} + \frac{5\varepsilon^2}{a}\right).$$

*Proof.* Consider a synchronous coupling for $(\overleftarrow{\mathbf{U}}_t)_{t\in[0,T]}$ and $(\bar{\mathbf{U}}_t)_{t\in[0,T]}$, *i.e.*, use the same Brownian motion to drive the two processes, with the same initial point, *i.e.*, $\overleftarrow{\mathbf{U}}_0 = \bar{\mathbf{U}}_0$. Then, it holds that

$$\mathcal{W}_2\left(\mathcal{L}(\overleftarrow{\mathbf{U}}_T), \mathcal{L}(\bar{\mathbf{U}}_T)\right) \leq \left\|\overleftarrow{\mathbf{U}}_T - \bar{\mathbf{U}}_T\right\|_{L_2}.$$

Fix $0 < \Delta < h$ and let $t_N := T - \Delta$. Note that, for all $0 \leq k \leq N-1$, from (11) and (12),

$$\overleftarrow{\mathbf{U}}_{t_{k+1}} - \bar{\mathbf{U}}_{t_{k+1}}$$
$$= \overleftarrow{\mathbf{U}}_{t_k} - \bar{\mathbf{U}}_{t_k} + \int_{t_k}^{t_{k+1}}\left\{\tilde{A}_\epsilon\left(\overleftarrow{\mathbf{U}}_t - \bar{\mathbf{U}}_{t_k}\right) + \Sigma_\epsilon^2\left(\tilde{\mathsf{s}}_{T-t}\left(\overleftarrow{\mathbf{U}}_t\right) - \tilde{\mathsf{s}}_{T-t_k}\left(\bar{\mathbf{U}}_{t_k}\right)\right)\right\}\mathrm{d}t,$$

where

$$\tilde{A}_\epsilon = -A - \Sigma_\epsilon^2\Sigma_\infty^{-1}.$$

From Monmarché (2023, Lemma 5 and Proposition 4) and Achleitner et al. (2015, Lemma 2.6), there exists a symmetric positive definite matrix $\mathfrak{M} \in \mathbb{R}^{2d\times 2d}$ such that, for any fixed $\eta > 0$, we have

$$\mathfrak{M}\tilde{A}_\epsilon \preccurlyeq -(a-\eta)\mathfrak{M}. \quad (38)$$

We then prove contraction with respect to the norm associated with $\mathfrak{M}$ defined, for all $v \in \mathbb{R}^{2d}$, by $\|v\|_{\mathfrak{M}}^2 := v^\top\mathfrak{M}v$.

For $t \in [t_k, t_{k+1})$,

$$\mathrm{d}\left(\overleftarrow{\mathbf{U}}_t - \bar{\mathbf{U}}_t\right) = \tilde{A}_\epsilon\left(\overleftarrow{\mathbf{U}}_t - \bar{\mathbf{U}}_{t_k}\right)\mathrm{d}t + \Sigma_\epsilon^2\left(\tilde{s}_{T-t}\left(\overleftarrow{\mathbf{U}}_t\right) - \tilde{s}_{T-t_k}\left(\bar{\mathbf{U}}_{t_k}\right)\right)\mathrm{d}t.$$

This means that we have

$$\mathrm{d}\left(\left(\overleftarrow{\mathbf{U}}_t - \bar{\mathbf{U}}_t\right)^\top \mathfrak{M}\left(\overleftarrow{\mathbf{U}}_t - \bar{\mathbf{U}}_t\right)\right) = 2\left(\overleftarrow{\mathbf{U}}_t - \bar{\mathbf{U}}_t\right)^\top \mathfrak{M}\,\mathrm{d}\left(\overleftarrow{\mathbf{U}}_t - \bar{\mathbf{U}}_t\right).$$

It follows that,

$$\left\|\overleftarrow{\mathbf{U}}_{t_{k+1}} - \bar{\mathbf{U}}_{t_{k+1}}\right\|_{\mathfrak{M}}^2 = \left\|\overleftarrow{\mathbf{U}}_{t_k} - \bar{\mathbf{U}}_{t_k}\right\|_{\mathfrak{M}}^2 + 2\int_{t_k}^{t_{k+1}} \left(\overleftarrow{\mathbf{U}}_t - \bar{\mathbf{U}}_{t_k}\right)^\top \mathfrak{M}\tilde{A}_\epsilon \left(\overleftarrow{\mathbf{U}}_t - \bar{\mathbf{U}}_{t_k}\right)\mathrm{d}t$$

$$+ 2\int_{t_k}^{t_{k+1}} \left(\overleftarrow{\mathbf{U}}_t - \bar{\mathbf{U}}_{t_k}\right)^\top \mathfrak{M}\Sigma_\epsilon^2 \left(\tilde{\mathsf{s}}_{T-t}\left(\overleftarrow{\mathbf{U}}_t\right) - \tilde{\mathsf{s}}_{T-t_k}\left(\bar{\mathbf{U}}_{t_k}\right)\right)\mathrm{d}t$$

$$\le \left\|\overleftarrow{\mathbf{U}}_{t_k} - \bar{\mathbf{U}}_{t_k}\right\|_{\mathfrak{M}}^2 + 2\left(A_{1,k} + A_{2,k} + A_{3,k} + A_{4,k} + A_{5,k} + A_{6,k}\right),$$

where

$$A_{1,k} := h\left(\overleftarrow{\mathbf{U}}_{t_k} - \bar{\mathbf{U}}_{t_k}\right)^\top \mathfrak{M}\tilde{A}_\epsilon \left(\overleftarrow{\mathbf{U}}_{t_k} - \bar{\mathbf{U}}_{t_k}\right)$$

$$+ h\left(\overleftarrow{\mathbf{U}}_{t_k} - \bar{\mathbf{U}}_{t_k}\right)^\top \mathfrak{M}\Sigma_\epsilon^2 \left(\tilde{\mathsf{s}}_{T-t_k}\left(\overleftarrow{\mathbf{U}}_{t_k}\right) - \tilde{\mathsf{s}}_{T-t_k}\left(\bar{\mathbf{U}}_{t_k}\right)\right),$$

$$A_{2,k} := \int_{t_k}^{t_{k+1}} \left(\overleftarrow{\mathbf{U}}_t - \overleftarrow{\mathbf{U}}_{t_k}\right)^\top \mathfrak{M}\left\{\tilde{A}_\epsilon \left(\overleftarrow{\mathbf{U}}_{t_k} - \bar{\mathbf{U}}_{t_k}\right) + \Sigma_\epsilon^2 \left(\tilde{\mathsf{s}}_{T-t_k}\left(\overleftarrow{\mathbf{U}}_{t_k}\right) - \tilde{\mathsf{s}}_{T-t_k}\left(\bar{\mathbf{U}}_{t_k}\right)\right)\right\}\mathrm{d}t,$$

$$A_{3,k} := \int_{t_k}^{t_{k+1}} \left(\overleftarrow{\mathbf{U}}_t - \overleftarrow{\mathbf{U}}_{t_k}\right)^\top \tilde{A}_\epsilon^\top \mathfrak{M}\left(\overleftarrow{\mathbf{U}}_{t_k} - \bar{\mathbf{U}}_{t_k}\right)\mathrm{d}t,$$

$$A_{4,k} := \left(\overleftarrow{\mathbf{U}}_{t_k} - \bar{\mathbf{U}}_{t_k}\right)^\top \mathfrak{M}\Sigma_\epsilon^2 \int_{t_k}^{t_{k+1}} \left(\tilde{\mathsf{s}}_{T-t}\left(\overleftarrow{\mathbf{U}}_t\right) - \tilde{\mathsf{s}}_{T-t_k}\left(\overleftarrow{\mathbf{U}}_{t_k}\right)\right)\mathrm{d}t,$$

$$A_{5,k} := \int_{t_k}^{t_{k+1}} \left(\overleftarrow{\mathbf{U}}_t - \overleftarrow{\mathbf{U}}_{t_k}\right)^\top \mathfrak{M}\tilde{A}_\epsilon \left(\overleftarrow{\mathbf{U}}_t - \overleftarrow{\mathbf{U}}_{t_k}\right)\mathrm{d}t,$$

$$A_{6,k} := \int_{t_k}^{t_{k+1}} \left(\overleftarrow{\mathbf{U}}_t - \overleftarrow{\mathbf{U}}_{t_k}\right)^\top \mathfrak{M}\Sigma_\epsilon^2 \left(\tilde{\mathsf{s}}_{T-t}\left(\overleftarrow{\mathbf{U}}_t\right) - \tilde{\mathsf{s}}_{T-t_k}\left(\overleftarrow{\mathbf{U}}_{t_k}\right)\right)\mathrm{d}t.$$

Next, we bound each term of the above decomposition separately.

*Bound of* $\mathbb{E}[A_{1,k}]$. By Assumption H2, applying Proposition B.1, there exists a constant $C$ (that depends on the eigenvalues of $\mathfrak{M}$ or constant terms and that may vary from line to line) such that

$$\left(\overleftarrow{\mathbf{U}}_{t_k} - \bar{\mathbf{U}}_{t_k}\right)^\top \mathfrak{M}\Sigma_\epsilon^2 \left(\tilde{\mathsf{s}}_{T-t_k}\left(\overleftarrow{\mathbf{U}}_{t_k}\right) - \tilde{\mathsf{s}}_{T-t_k}\left(\bar{\mathbf{U}}_{t_k}\right)\right) \le C\tilde{L}_{T-t_k}\left\|\overleftarrow{\mathbf{U}}_{t_k} - \bar{\mathbf{U}}_{t_k}\right\|_{\mathfrak{M}}^2,$$

and using (38),

$$\left(\overleftarrow{\mathbf{U}}_{t_k} - \bar{\mathbf{U}}_{t_k}\right)^\top \mathfrak{M}\tilde{A}_\epsilon \left(\overleftarrow{\mathbf{U}}_{t_k} - \bar{\mathbf{U}}_{t_k}\right) \le -(a - \eta)\left\|\overleftarrow{\mathbf{U}}_{t_k} - \bar{\mathbf{U}}_{t_k}\right\|_{\mathfrak{M}}^2.$$

Combining this with (38) yields

$$\mathbb{E}\left[A_{1,k}\right] \le h\left(C\tilde{L}_{T-t_k} - (a - \eta)\right)\mathbb{E}\left[\left\|\overleftarrow{\mathbf{U}}_{t_k} - \bar{\mathbf{U}}_{t_k}\right\|_{\mathfrak{M}}^2\right].$$

*Bound of* $\mathbb{E}[A_{2,k}]$. Using the Cauchy-Schwarz inequality,

$$\mathbb{E}\left[A_{2,k}\right] \le \mathbb{E}\left[\left\|\int_{t_k}^{t_{k+1}} \left(\overleftarrow{\mathbf{U}}_t - \overleftarrow{\mathbf{U}}_{t_k}\right)\mathrm{d}t\right\|^2\right]^{1/2}$$

$$\times \mathbb{E}\left[\left\|\mathfrak{M}\left\{\tilde{A}_\epsilon \left(\overleftarrow{\mathbf{U}}_{t_k} - \bar{\mathbf{U}}_{t_k}\right) + \Sigma_\epsilon^2 \left(\tilde{\mathsf{s}}_{T-t_k}\left(\overleftarrow{\mathbf{U}}_{t_k}\right) - \tilde{\mathsf{s}}_{T-t_k}\left(\bar{\mathbf{U}}_{t_k}\right)\right)\right\}\right\|^2\right]^{1/2}$$

$$\le C\mathbb{E}\left[\left\|\int_{t_k}^{t_{k+1}} \left(\overleftarrow{\mathbf{U}}_t - \overleftarrow{\mathbf{U}}_{t_k}\right)\mathrm{d}t\right\|^2\right]^{1/2} \mathbb{E}\left[\left\|\sqrt{\mathfrak{M}}\left(\tilde{b}_{t_k}\left(\overleftarrow{\mathbf{U}}_{t_k}\right) - \tilde{b}_{t_k}\left(\bar{\mathbf{U}}_{t_k}\right)\right)\right\|^2\right]^{1/2},$$

with $\tilde{b}_t$ the backward drift in (11) defined by $\tilde{b}_t : u \mapsto \tilde{A}_\epsilon u + \Sigma_\epsilon^2 \tilde{s}_{T-t}(u)$. On the one hand, the Cauchy-Schwarz inequality implies

$$\mathbb{E}\left[\left\|\int_{t_k}^{t_{k+1}} \left(\overleftarrow{\mathbf{U}}_t - \overleftarrow{\mathbf{U}}_{t_k}\right) \mathrm{d}t\right\|^2\right]^{1/2} \leq \sqrt{h} \left(\int_{t_k}^{t_{k+1}} \mathbb{E}\left[\left\|\overleftarrow{\mathbf{U}}_t - \overleftarrow{\mathbf{U}}_{t_k}\right\|^2\right] \mathrm{d}t\right)^{1/2}.$$

Using the time-reversal property, Lemma D.3, together with Cauchy-Schwarz inequality and Itô's isometry,

$$\begin{aligned}
\mathbb{E}\left[\left\|\overleftarrow{\mathbf{U}}_t - \overleftarrow{\mathbf{U}}_{t_k}\right\|^2\right] &\leq \mathbb{E}\left[\left\|\int_{T-t}^{T-t_k} A\overrightarrow{\mathbf{U}}_s \mathrm{d}s + \Sigma_\epsilon \mathrm{d}B_s\right\|^2\right] \\
&\leq C\left(h\|A\|^2 \int_{T-t}^{T-t_k} \mathbb{E}\left[\left\|\overrightarrow{\mathbf{U}}_s\right\|^2\right] \mathrm{d}s + hd\left\|\Sigma_\epsilon\right\|^2\right) \\
&\leq C\left(h^2\|A\|^2 B^2 + hd\left\|\Sigma_\epsilon\right\|^2\right),
\end{aligned} \tag{39}$$

where $B$ is defined in (56). It follows that

$$\mathbb{E}\left[\left\|\int_{t_k}^{t_{k+1}} \left(\overleftarrow{\mathbf{U}}_t - \overleftarrow{\mathbf{U}}_{t_k}\right) \mathrm{d}t\right\|^2\right]^{1/2} \leq h\sqrt{h}C\left(h\|A\|^2 B^2 + \|\Sigma_\epsilon\|^2 d\right)^{1/2}.$$

On the other hand,

$$\mathbb{E}\left[\left\|\sqrt{\mathfrak{M}}\left(\tilde{b}_{t_k}\left(\overleftarrow{\mathbf{U}}_{t_k}\right) - \tilde{b}_{t_k}\left(\bar{\mathbf{U}}_{t_k}\right)\right)\right\|^2\right]^{1/2}$$
$$\leq C\left(\|\tilde{A}_\epsilon\| + \tilde{L}_{T-t_k}\right) \mathbb{E}\left[\left\|\overleftarrow{\mathbf{U}}_{t_k} - \bar{\mathbf{U}}_{t_k}\right\|_{\mathfrak{M}}^2\right]^{1/2}.$$

Therefore,

$$\begin{aligned}
\mathbb{E}[A_{2,k}] &\leq Ch\sqrt{h}\left(\|\tilde{A}_\epsilon\| + \tilde{L}_{T-t_k}\right)\left(h\|A\|^2 B^2 + \|\Sigma_\epsilon\|^2 d\right)^{1/2} \mathbb{E}\left[\left\|\overleftarrow{\mathbf{U}}_{t_k} - \bar{\mathbf{U}}_{t_k}\right\|_{\mathfrak{M}}^2\right]^{1/2} \\
&= Ch\sqrt{h}\|\tilde{A}_\epsilon\|\left(h\|A\|^2 B^2 + \|\Sigma_\epsilon\|^2 d\right)^{1/2} \mathbb{E}\left[\left\|\overleftarrow{\mathbf{U}}_{t_k} - \bar{\mathbf{U}}_{t_k}\right\|_{\mathfrak{M}}^2\right]^{1/2} \\
&+ Ch\sqrt{h}\tilde{L}_{T-t_k}\left(h\|A\|^2 B^2 + \|\Sigma_\epsilon\|^2 d\right)^{1/2} \mathbb{E}\left[\left\|\overleftarrow{\mathbf{U}}_{t_k} - \bar{\mathbf{U}}_{t_k}\right\|_{\mathfrak{M}}^2\right]^{1/2}.
\end{aligned}$$

Moreover, from Young's inequality, we get that, for all $a, b \geq 0$ and $\alpha > 0$,

$$ab \leq \frac{\alpha}{2}a^2 + \frac{1}{2\alpha}b^2. \tag{40}$$

It follows that,

$$\begin{aligned}
\mathbb{E}[A_{2,k}] &\leq \frac{a-\eta}{6}h\,\mathbb{E}\left[\left\|\overleftarrow{\mathbf{U}}_{t_k} - \bar{\mathbf{U}}_{t_k}\right\|_{\mathfrak{M}}^2\right] + h^2 C\frac{\|\tilde{A}_\epsilon\|^2\left(h\|A\|^2 B^2 + \|\Sigma_\epsilon\|^2 d\right)}{a-\eta} \\
&+ Ch\tilde{L}_{T-t_k}\mathbb{E}\left[\left\|\overleftarrow{\mathbf{U}}_{t_k} - \bar{\mathbf{U}}_{t_k}\right\|_{\mathfrak{M}}^2\right] + Ch^2\tilde{L}_{T-t_k}\left(h\|A\|^2 B^2 + \|\Sigma_\epsilon\|^2 d\right).
\end{aligned}$$

*Bound of* $\mathbb{E}[A_{3,k}]$. Using Cauchy-Schwarz inequality,

$$\mathbb{E}[A_{3,k}] \leq \mathbb{E}\left[\left\|\int_{t_k}^{t_{k+1}} \left(\overleftarrow{\mathbf{U}}_t - \overleftarrow{\mathbf{U}}_{t_k}\right) \mathrm{d}t\right\|^2\right]^{1/2} \mathbb{E}\left[\left\|\tilde{A}_\epsilon^\top \mathfrak{M}\left(\overleftarrow{\mathbf{U}}_{t_k} - \bar{\mathbf{U}}_{t_k}\right)\right\|^2\right]^{1/2}.$$

On the one hand,

$$\mathbb{E}\left[\left\|\tilde{A}_\epsilon^\top \mathfrak{M}\left(\overleftarrow{\mathbf{U}}_{t_k} - \bar{\mathbf{U}}_{t_k}\right)\right\|^2\right]^{1/2} \leq C\|\tilde{A}_\epsilon\|\mathbb{E}\left[\left\|\overleftarrow{\mathbf{U}}_{t_k} - \bar{\mathbf{U}}_{t_k}\right\|^2\right]^{1/2},$$

and, on the other hand, using (39) yields,

$$\mathbb{E}\left[\left\|\int_{t_k}^{t_{k+1}}\left(\overleftarrow{\mathbf{U}}_t - \overleftarrow{\mathbf{U}}_{t_k}\right)\mathrm{d}t\right\|^2\right]^{1/2} \le Ch\sqrt{h}\left(h\|A\|^2 B^2 + \|\Sigma_\epsilon\|^2 d\right)^{1/2}.$$

Combining both and using (40) yield,

$$\mathbb{E}\left[A_{3,k}\right] \le Ch\sqrt{h}\|\tilde{A}_\epsilon\|\sqrt{h\|A\|^2 B^2 + \|\Sigma_\epsilon\|^2 d} \times \mathbb{E}\left[\left\|\overleftarrow{\mathbf{U}}_{t_k} - \bar{\mathbf{U}}_{t_k}\right\|_{\mathfrak{M}}^2\right]^{1/2}$$

$$\le \frac{a-\eta}{6}h\mathbb{E}\left[\left\|\overleftarrow{\mathbf{U}}_{t_k} - \bar{\mathbf{U}}_{t_k}\right\|_{\mathfrak{M}}^2\right] + h^2 C \frac{\|\tilde{A}_\epsilon\|^2\left(h\|A\|^2 B^2 + \|\Sigma_\epsilon\|^2 d\right)}{a-\eta}.$$

*Bound of $\mathbb{E}[A_{4,k}]$.* Using Cauchy-Schwarz inequality,

$$\mathbb{E}\left[A_{4,k}\right] \le C\mathbb{E}\left[\left\|\overleftarrow{\mathbf{U}}_{t_k} - \bar{\mathbf{U}}_{t_k}\right\|_{\mathfrak{M}}^2\right]^{1/2}\mathbb{E}\left[\left\|\int_{t_k}^{t_{k+1}}\Sigma_\epsilon^2\left(\tilde{\mathsf{s}}_{T-t}\left(\overleftarrow{\mathbf{U}}_t\right) - \tilde{\mathsf{s}}_{T-t_k}\left(\overleftarrow{\mathbf{U}}_{t_k}\right)\right)\mathrm{d}t\right\|^2\right]^{1/2}.$$

Therefore, using Cauchy–Schwarz inequality again,

$$\mathbb{E}\left[\left\|\int_{t_k}^{t_{k+1}}\Sigma_\epsilon^2\left(\tilde{\mathsf{s}}_{T-t}\left(\overleftarrow{\mathbf{U}}_t\right) - \tilde{\mathsf{s}}_{T-t_k}\left(\overleftarrow{\mathbf{U}}_{t_k}\right)\right)\mathrm{d}t\right\|^2\right]^{1/2}$$

$$\le \sqrt{h}\left(\int_{t_k}^{t_{k+1}}\mathbb{E}\left[\left\|\Sigma_\epsilon^2\left(\tilde{\mathsf{s}}_{T-t}\left(\overleftarrow{\mathbf{U}}_t\right) - \tilde{\mathsf{s}}_{T-t_k}\left(\overleftarrow{\mathbf{U}}_{t_k}\right)\right)\right\|^2\right]\mathrm{d}t\right)^{1/2}.$$

Then, using Lemma D.8,

$$\mathbb{E}\left[\left\|\Sigma_\epsilon^2\left(\tilde{\mathsf{s}}_{T-t}\left(\overleftarrow{\mathbf{U}}_t\right) - \tilde{\mathsf{s}}_{T-t_k}\left(\overleftarrow{\mathbf{U}}_{t_k}\right)\right)\right\|^2\right] \le \|\Sigma_\epsilon\|^2\mathbb{E}\left[\left\|\nabla\log\tilde{p}_{T-t}\left(\overrightarrow{\mathbf{U}}_t\right) - \nabla\log\tilde{p}_{T-t_k}\left(\overrightarrow{\mathbf{U}}_{t_k}\right)\right\|^2\right]$$

$$\le C\|\Sigma_\epsilon\|^2\left(g(t_{k+1}) - g(t_k)\right), \tag{41}$$

with the function $g$ defined in (59). This yields

$$\mathbb{E}\left[A_{4,k}\right] \le Ch\|\Sigma_\epsilon\|\sqrt{g(t_{k+1}) - g(t_k)} \times \mathbb{E}\left[\left\|\overleftarrow{\mathbf{U}}_{t_k} - \bar{\mathbf{U}}_{t_k}\right\|_{\mathfrak{M}}^2\right]^{1/2}$$

$$\le h\frac{a-\eta}{6}\mathbb{E}\left[\left\|\overleftarrow{\mathbf{U}}_{t_k} - \bar{\mathbf{U}}_{t_k}\right\|_{\mathfrak{M}}^2\right] + h\frac{C\|\Sigma_\epsilon\|^2}{a-\eta}\left(g(t_{k+1}) - g(t_k)\right),$$

where we have used Young's inequality in the last inequality.

*Bound of $\mathbb{E}[A_{5,k}]$.* Using Cauchy-Schwarz inequality,

$$\mathbb{E}\left[A_{5,k}\right] = \int_{t_k}^{t_{k+1}}\mathbb{E}\left[\left(\overleftarrow{\mathbf{U}}_t - \overleftarrow{\mathbf{U}}_{t_k}\right)^\top\mathfrak{M}\tilde{A}_\epsilon\left(\overleftarrow{\mathbf{U}}_t - \overleftarrow{\mathbf{U}}_{t_k}\right)\right]\mathrm{d}t$$

$$\le C\left\|\tilde{A}_\epsilon\right\|\mathbb{E}\left[\int_{t_k}^{t_{k+1}}\left\|\overleftarrow{\mathbf{U}}_s - \overleftarrow{\mathbf{U}}_{t_k}\right\|^2\mathrm{d}s\right].$$

Then using Lemma D.3 as in (39),

$$\mathbb{E}\left[A_{5,k}\right] \le Ch^2\left\|\tilde{A}_\epsilon\right\|\left(h\|A\|^2 B^2 + \|\Sigma_\epsilon\|^2 d\right),$$

with $B$ defined as in 56.

*Bound of $\mathbb{E}[A_{6,k}]$.* Using Cauchy-Schwarz inequality,

$$\mathbb{E}\left[A_{6,k}\right] = \int_{t_k}^{t_{k+1}}\mathbb{E}\left[\left(\overleftarrow{\mathbf{U}}_t - \overleftarrow{\mathbf{U}}_{t_k}\right)^\top\mathfrak{M}\Sigma_\epsilon^2\left(\tilde{\mathsf{s}}_{T-t}\left(\overleftarrow{\mathbf{U}}_t\right) - \tilde{\mathsf{s}}_{T-t_k}\left(\overleftarrow{\mathbf{U}}_{t_k}\right)\right)\right]\mathrm{d}t$$

$$\le C\int_{t_k}^{t_{k+1}}\mathbb{E}\left[\left\|\overleftarrow{\mathbf{U}}_t - \overleftarrow{\mathbf{U}}_{t_k}\right\|^2\right]^{1/2}\mathbb{E}\left[\left\|\Sigma_\epsilon^2\left(\tilde{\mathsf{s}}_{T-t}\left(\overleftarrow{\mathbf{U}}_t\right) - \tilde{\mathsf{s}}_{T-t_k}\left(\overleftarrow{\mathbf{U}}_{t_k}\right)\right)\right\|^2\right]^{1/2}\mathrm{d}t.$$

Controlling the first term as in (39) and the second term as in (41), using Lemma D.8, together with Young's inequality, yields,

$$\mathbb{E}\left[A_{6,k}\right] \le Ch^2(a-\eta)\left(h\|A\|^2B^2 + \|\Sigma_\epsilon\|^2 d\right) + Ch\frac{\|\Sigma_\epsilon\|^2\left(g(t_{k+1}) - g(t_k)\right)}{a-\eta} \; .$$

*Final bound.* Combining the upper bounds for $A_{1,k}$, $A_{2,k}$, $A_{3,k}$, $A_{4,k}$, $A_{5,k}$ and $A_{6,k}$, there exists a constant $C > 0$ such that

$$
\begin{aligned}
\mathbb{E}\left[\left\|\overleftarrow{\mathbf{U}}_{t_{k+1}} - \bar{\mathbf{U}}_{t_{k+1}}\right\|_{\mathfrak{M}}^2\right] &\le \delta_k \mathbb{E}\left[\left\|\overleftarrow{\mathbf{U}}_{t_k} - \bar{\mathbf{U}}_{t_k}\right\|_{\mathfrak{M}}^2\right] + Ch\frac{\|\Sigma_\epsilon\|^2}{a-\eta}(g(t_{k+1}) - g(t_k)) \\
&\quad + Ch^2\left(h\|A\|^2B^2 + \|\Sigma_\epsilon\|^2 d\right)\left((a-\eta) + \frac{a-\eta+2}{a-\eta}\|\tilde{A}_\epsilon\| \vee \|\tilde{A}_\epsilon\|^2\right) \\
&\quad + Ch^2\tilde{L}_{T-t_k}\left(h\|A\|^2B^2 + \|\Sigma_\epsilon\|^2 d\right),
\end{aligned}
$$

with $\delta_k := 1 + h(C\tilde{L}_{T-t_k} - (a-\eta)/2)$. Therefore,

$$
\begin{aligned}
\mathbb{E}\left[\left\|\overleftarrow{\mathbf{U}}_{t_N} - \bar{\mathbf{U}}_{t_N}\right\|_{\mathfrak{M}}^2\right] &\le \left(\prod_{k=0}^{N-1}\delta_k\right)\mathbb{E}\left[\left\|\overleftarrow{\mathbf{U}}_0 - \bar{\mathbf{U}}_0\right\|_{\mathfrak{M}}^2\right] \\
&\quad + Ch^2\left(h\|A\|^2B^2 + \|\Sigma_\epsilon\|^2 d\right)\left((a-\eta) + \frac{a-\eta+2}{a-\eta}\|\tilde{A}_\epsilon\| \vee \|\tilde{A}_\epsilon\|^2\right)\sum_{k=0}^{N-1}\prod_{j=k+1}^{N-1}\delta_j \\
&\quad + Ch\frac{\|\Sigma_\epsilon\|^2}{a-\eta}\sum_{k=0}^{N-1}(g(t_{k+1}) - g(t_k))\prod_{j=k+1}^{N-1}\delta_j \\
&\quad + Ch^2\left(h\|A\|^2B^2 + \|\Sigma_\epsilon\|^2 d\right)\sum_{k=0}^{N-1}\tilde{L}_{T-t_k}\prod_{j=k+1}^{N-1}\delta_j \; .
\end{aligned}
\tag{42}
$$

First recall that the two processes share the same initialization, *i.e.* $\overleftarrow{\mathbf{U}}_0 = \bar{\mathbf{U}}_0$. Note that, since $\exp(x) \ge 1 + x$, for $x \in \mathbb{R}$,

$$
\begin{aligned}
\prod_{j=k+1}^{N-1}\delta_j &\le \exp\left(\sum_{j=k+1}^{N-1}h\left(C\tilde{L}_{T-t_j} - \frac{a-\eta}{2}\right)\right) \\
&\le \exp\left(-\frac{a-\eta}{2}h(N-k-1) + C\sum_{j=k+1}^{N-1}h\tilde{L}_{T-t_j}\right) \\
&\le \exp\left(-\frac{a-\eta}{2}h(N-k-1) + C\int_0^\infty \tilde{L}_s\mathrm{d}s\right) \\
&\le \exp\left(-\frac{a-\eta}{2}h(N-k-1) + Ca^{-1}\right),
\end{aligned}
$$

where we use the bound (26) from Proposition B.1. Combining this bound with the fact that $h \le (1 - \mathrm{e}^{-h})\mathrm{e}^h$, we obtain

$$h\exp\left(-\frac{a-\eta}{2}(N-k)h\right) \le \frac{2\mathrm{e}^h}{a-\eta}\left(\exp\left(-\frac{a-\eta}{2}(N-k)h\right) - \exp\left(-\frac{a-\eta}{2}(N-k+1)h\right)\right),$$

which then implies that

$$
\begin{aligned}
h\sum_{k=0}^{N-1}\prod_{j=k+1}^{N-1}\delta_j &\le h\sum_{k=0}^{N-1}\mathrm{e}^{-\frac{a-\eta}{2}(N-k-2)h} \times \mathrm{e}^{Ca^{-1}} \\
&\le \frac{2\mathrm{e}^h}{a-\eta}\sum_{k=0}^{N-1}\left(\mathrm{e}^{-\frac{a-\eta}{2}(N-k-2)h} - \mathrm{e}^{-\frac{a-\eta}{2}(N-k-1)h}\right) \times \mathrm{e}^{Ca^{-1}} \le \frac{C\mathrm{e}^{Ca^{-1}}}{a-\eta},
\end{aligned}
$$

increasing the value of the constant $C$ if necessary. For the term involving $\tilde{L}_{T-t_k}$, note that

$$
\begin{aligned}
h\tilde{L}_{T-t_k} &\exp(-\frac{a-\eta}{2}(N-k)h) \\
&\leq h\frac{C}{\sqrt{(N-k)h}} \exp(-\frac{a-\eta}{2}(N-k)h) \\
&\leq \frac{2\mathrm{e}^h}{a-\eta}\left(\Gamma\left(\frac{1}{2},\frac{a-\eta}{2}(N-k)h\right) - \Gamma\left(\frac{1}{2},\frac{a-\eta}{2}(N-k+1)h\right)\right),
\end{aligned}
$$

where $\Gamma(a,b)$ denotes the Gamma function. Consequently,

$$
\begin{aligned}
h\sum_{k=0}^{N-1}\tilde{L}_{T-t_k}&\prod_{j=k+1}^{N-1}\delta_j \\
&\leq \frac{2\mathrm{e}^h}{a-\eta}\sum_{k=0}^{N-1}\left(\Gamma\left(\frac{1}{2},\frac{a-\eta}{2}(N-k-2)h\right) - \Gamma\left(\frac{1}{2},\frac{a-\eta}{2}(N-k-1)h\right)\right)\times\mathrm{e}^{Ca^{-1}} \\
&\leq \frac{C\mathrm{e}^{Ca^{-1}}}{a-\eta}\,.
\end{aligned}
$$

Moreover, we have that

$$
\sum_{k=0}^{N-1}(g(t_{k+1})-g(t_k))\prod_{j=k+1}^{N-1}\delta_j \leq \mathrm{e}^{Ca^{-1}}\sum_{k=0}^{N-1}(g(t_{k+1})-g(t_k)) \leq \mathrm{e}^{Ca^{-1}}g(t_N)
$$

$$
\leq C\mathrm{e}^{Ca^{-1}}\mathbb{E}\left[\left\|\tilde{\mathsf{s}}_\Delta\left(\vec{\mathbf{U}}_\Delta\right)\right\|^2\right]\,.
$$

Note that $\mathbb{E}[\|\tilde{\mathsf{s}}_\Delta(\vec{\mathbf{U}}_\Delta)\|^2]$ corresponds to the relative Fisher information between $p_\Delta$ and $\pi_\infty$. We can conclude for $\Delta \to 0$, following the argument of Conforti et al. (Lemma 3.9, 2025) and using Assumption H1, that $\mathcal{I}(\pi_{\mathrm{data}}\otimes\pi_v|\pi_\infty) = \mathbb{E}[\|\tilde{\mathsf{s}}_0(\vec{\mathbf{U}}_0)\|^2] < \infty$. Then, applying (42) directly yields

$$
\mathbb{E}\left[\left\|\overleftarrow{\mathbf{U}}_{t_N} - \bar{\mathbf{U}}_{t_N}\right\|_{\mathfrak{M}}^2\right] \leq h \times C\left(h\|A\|^2 B^2 + \|\Sigma_\epsilon\|^2\left(d + \mathcal{I}(\pi_{\mathrm{data}}\otimes\pi_v|\pi_\infty)\right)\right)\frac{\mathrm{e}^{Ca^{-1}}}{a-\eta}\,,
$$

which concludes the proof.

$\square$

**Lemma B.3** (Approximation error). *Assume that Assumptions H2 and H3 hold. Then, for any $\eta > 0$, there exists a constant $C > 0$ such that*

$$
\mathcal{W}_2\left(\mathcal{L}\left(\bar{\mathbf{U}}_T^\infty\right), \mathcal{L}\left(\bar{\mathbf{U}}_T^\theta\right)\right) \leq C\frac{\|\Sigma_\epsilon\|^2}{a-\eta}M\,. \tag{43}
$$

*Proof.* As in the proof of Lemma B.2, consider the synchronous coupling of the two processes $\bar{\mathbf{U}}^\infty$ and $\bar{\mathbf{U}}^\theta$ with the same initial condition $\bar{\mathbf{U}}_0^\infty = \bar{\mathbf{U}}_0^\theta$. We have

$$
\mathcal{W}_2\left(\mathcal{L}\left(\bar{\mathbf{U}}_T^\infty\right), \mathcal{L}\left(\bar{\mathbf{U}}_T^\theta\right)\right) \leq \left\|\bar{\mathbf{U}}_T^\infty - \bar{\mathbf{U}}_T^\theta\right\|_{L_2}\,.
$$

Fix $\Delta \geq 0$ such that $t_N = T - \Delta$ and note that for all $0 \leq k \leq N-1$, from (12) and (13), we get

$$
\begin{aligned}
\bar{\mathbf{U}}_{t_{k+1}}^\infty &- \bar{\mathbf{U}}_{t_{k+1}}^\theta \\
&= \bar{\mathbf{U}}_{t_k}^\infty - \bar{\mathbf{U}}_{t_k}^\theta + \int_{t_k}^{t_{k+1}}\left\{\tilde{A}_\epsilon\left(\bar{\mathbf{U}}_{t_k}^\infty - \bar{\mathbf{U}}_{t_k}^\theta\right) + \Sigma_\epsilon^2\left(\tilde{\mathsf{s}}_{T-t_k}\left(\bar{\mathbf{U}}_{t_k}^\infty\right) - \tilde{s}_\theta\left(T-t_k, \bar{\mathbf{U}}_{t_k}^\theta\right)\right)\right\}\mathrm{d}t\,.
\end{aligned}
$$

Taking $\mathfrak{M}$ as in the proof of Lemma B.2, we have

$$
\left\|\bar{\mathbf{U}}_{t_{k+1}}^\infty - \bar{\mathbf{U}}_{t_{k+1}}^\theta\right\|_{\mathfrak{M}}^2 = \left\|\bar{\mathbf{U}}_{t_k}^\infty - \bar{\mathbf{U}}_{t_k}^\theta\right\|_{\mathfrak{M}}^2 + 2B_{1,k} + 2B_{2,k}\,,
$$

with

$$B_{1,k} = h \left( \bar{\mathbf{U}}_{t_k}^{\infty} - \bar{\mathbf{U}}_{t_k}^{\theta} \right)^{\top} \mathfrak{M} \tilde{A}_{\epsilon} \left( \bar{\mathbf{U}}_{t_k}^{\infty} - \bar{\mathbf{U}}_{t_k}^{\theta} \right),$$

$$B_{2,k} = h \left( \bar{\mathbf{U}}_{t_k}^{\infty} - \bar{\mathbf{U}}_{t_k}^{\theta} \right)^{\top} \mathfrak{M} \Sigma_{\epsilon}^2 \left( \tilde{\mathsf{s}}_{T-t_k} \left( \bar{\mathbf{U}}_{t_k}^{\infty} \right) - \tilde{s}_{\theta} \left( T - t_k, \bar{\mathbf{U}}_{t_k}^{\theta} \right) \right).$$

*Bound of* $\mathbb{E}[B_{1,k}]$. From (38), we note that

$$\mathbb{E}\left[ B_{1,k} \right] \leq -h \left( a - \eta \right) \mathbb{E}\left[ \left\| \bar{\mathbf{U}}_{t_k}^{\infty} - \bar{\mathbf{U}}_{t_k}^{\theta} \right\|_{\mathfrak{M}}^2 \right].$$

*Bound of* $\mathbb{E}[B_{2,k}]$. We decompose the second term into the score and approximation components:

$$\mathbb{E}\left[ B_{2,k} \right]$$
$$= h \mathbb{E}\left[ \left( \bar{\mathbf{U}}_{t_k}^{\infty} - \bar{\mathbf{U}}_{t_k}^{\theta} \right)^{\top} \mathfrak{M} \Sigma_{\epsilon}^2 \left( \tilde{\mathsf{s}}_{T-t_k} \left( \bar{\mathbf{U}}_{t_k}^{\infty} \right) - \tilde{\mathsf{s}}_{T-t_k} \left( \bar{\mathbf{U}}_{t_k}^{\theta} \right) \right) \right]$$
$$\quad + h \mathbb{E}\left[ \left( \bar{\mathbf{U}}_{t_k}^{\infty} - \bar{\mathbf{U}}_{t_k}^{\theta} \right)^{\top} \mathfrak{M} \Sigma_{\epsilon}^2 \left( \tilde{\mathsf{s}}_{T-t_k} \left( \bar{\mathbf{U}}_{t_k}^{\theta} \right) - \tilde{s}_{\theta} \left( T - t_k, \bar{\mathbf{U}}_{t_k}^{\theta} \right) \right) \right]$$
$$\leq h C \tilde{L}_{T-t_k} \mathbb{E}\left[ \left\| \bar{\mathbf{U}}_{t_k}^{\infty} - \bar{\mathbf{U}}_{t_k}^{\theta} \right\|_{\mathfrak{M}}^2 \right] + h \left\| \Sigma_{\epsilon} \right\| M \mathbb{E}\left[ \left\| \bar{\mathbf{U}}_{t_k}^{\infty} - \bar{\mathbf{U}}_{t_k}^{\theta} \right\|_{\mathfrak{M}} \right]$$
$$\leq h C \tilde{L}_{T-t_k} \mathbb{E}\left[ \left\| \bar{\mathbf{U}}_{t_k}^{\infty} - \bar{\mathbf{U}}_{t_k}^{\theta} \right\|_{\mathfrak{M}}^2 \right] + h \frac{a - \eta}{2} \mathbb{E}\left[ \left\| \bar{\mathbf{U}}_{t_k}^{\infty} - \bar{\mathbf{U}}_{t_k}^{\theta} \right\|_{\mathfrak{M}}^2 \right] + h C \left\| \Sigma_{\epsilon} \right\|^4 M^2,$$

where we have used Young's inequality in the last inequality, for $C > 0$ a universal constant (which may change from line to line) depending only on the eigenvalues of the matrix $\mathfrak{M}$ or constant factors. *Final bound.* Combining the bounds on $B_{1,k}$ and $B_{2,k}$, there exists a constant $C > 0$ such that

$$\mathbb{E}\left[ \left\| \bar{\mathbf{U}}_{t_{k+1}}^{\infty} - \bar{\mathbf{U}}_{t_{k+1}}^{\theta} \right\|_{\mathfrak{M}}^2 \right] \leq \delta_k \mathbb{E}\left[ \left\| \bar{\mathbf{U}}_{t_k}^{\infty} - \bar{\mathbf{U}}_{t_k}^{\theta} \right\|_{\mathfrak{M}}^2 \right] + h \frac{C}{a - \eta} \left\| \Sigma_{\epsilon} \right\|^4 M^2,$$

with $\delta_k := 1 + h \left( C \tilde{L}_{T-t_k} - (a - \eta)/2 \right)$. Therefore, we have

$$\mathbb{E}\left[ \left\| \bar{\mathbf{U}}_{t_N}^{\infty} - \bar{\mathbf{U}}_{t_N}^{\theta} \right\|_{\mathfrak{M}}^2 \right]$$
$$\leq \prod_{j=k+1}^{N-1} \delta_j \mathbb{E}\left[ \left\| \bar{\mathbf{U}}_0^{\infty} - \bar{\mathbf{U}}_0^{\theta} \right\|_{\mathfrak{M}}^2 \right] + h \frac{C}{a - \eta} \left\| \Sigma_{\epsilon} \right\|^4 M^2 \sum_{k=0}^{N-1} \prod_{j=k+1}^{N-1} \delta_j$$
$$\leq h \frac{C}{a - \eta} \left\| \Sigma_{\epsilon} \right\|^4 M^2 \sum_{k=0}^{N-1} \prod_{j=k+1}^{N-1} \delta_j,$$

where we used that $\bar{\mathbf{U}}_0^{\infty} = \bar{\mathbf{U}}_0^{\theta}$. Following the same argument as in Lemma B.2 (discretization error term), we obtain

$$\mathbb{E}\left[ \left\| \bar{\mathbf{U}}_{t_N}^{\infty} - \bar{\mathbf{U}}_{t_N}^{\theta} \right\|_{\mathfrak{M}}^2 \right] \leq C \frac{\left\| \Sigma_{\epsilon} \right\|^4 M^2}{(a - \eta)^2}.$$

We conclude the proof by taking the limit as $\Delta \to 0$ together with Fatou's lemma. $\qquad\square$

**Lemma B.4** (Mixing time). *Assume that H2 holds. Then, for all $\eta > 0$, there exists a constant $C > 0$ such that*

$$\mathcal{W}_2 \left( \mathcal{L} \left( \bar{\mathbf{U}}_T \right), \mathcal{L} \left( \bar{\mathbf{U}}_T^{\infty} \right) \right) \leq C e^{C a^{-1}} \times T e^{-\frac{3}{2}(a - \eta)T} \mathcal{W}_2 \left( \pi_{\mathrm{data}} \otimes \pi_v, \pi_{\infty} \right). \tag{44}$$

*Proof.* Consider a synchronous coupling of the continuous-time interpolations $(\bar{\mathbf{U}}_t)_{t \in [0,T]}$ and $(\bar{\mathbf{U}}_t^{\infty})_{t \in [0,T]}$, defined in (12), with initialization

$$\mathcal{W}_2 \left( \pi_{\infty}, \mathcal{L} \left( \overrightarrow{\mathbf{U}}_T \right) \right) = \left\| \bar{\mathbf{U}}_0 - \bar{\mathbf{U}}_0^{\infty} \right\|_{L_2}.$$

By definition of the $\mathcal{W}_2$ distance,

$$\mathcal{W}_2 \left( \mathcal{L} \left( \bar{\mathbf{U}}_T \right), \mathcal{L} \left( \bar{\mathbf{U}}_T^{\infty} \right) \right) \leq \left\| \bar{\mathbf{U}}_T - \bar{\mathbf{U}}_T^{\infty} \right\|_{L_2}.$$

Analogously to the proof of Lemma B.2 and Lemma B.3, fix $\Delta \geq 0$ such that $t_N = T - \delta$, and note that for $t \in [t_k, t_{k+1}]$, we have that

$$\left\| \bar{\mathbf{U}}_{t_{k+1}} - \bar{\mathbf{U}}_{t_{k+1}}^\infty \right\|_{\mathfrak{M}}^2 = \left\| \bar{\mathbf{U}}_{t_k} - \bar{\mathbf{U}}_{t_k}^\infty \right\|_{\mathfrak{M}}^2 + C_k \, ,$$

with

$$C_k = h 2 \left( \bar{\mathbf{U}}_{t_k} - \bar{\mathbf{U}}_{t_k}^\infty \right)^\top \mathfrak{M} \left\{ \tilde{A}_\epsilon \left( \bar{\mathbf{U}}_{t_k} - \bar{\mathbf{U}}_{t_k}^\infty \right) + \Sigma_\epsilon^2 \left( \tilde{\mathbf{s}}_{T-t_k} \left( \bar{\mathbf{U}}_{t_k} \right) - \tilde{\mathbf{s}}_{T-t_k} \left( \bar{\mathbf{U}}_{t_k}^\infty \right) \right) \right\} \, .$$

Similarly to Lemma B.2, we use that, for any fixed $\eta > 0$, we have

$$\mathfrak{M}\tilde{A}_\epsilon \preccurlyeq -(a - \eta)\mathfrak{M} \, .$$

Therefore,

$$\left( \bar{\mathbf{U}}_{t_k} - \bar{\mathbf{U}}_{t_k}^\infty \right)^\top \mathfrak{M}\tilde{A}_\epsilon \left( \bar{\mathbf{U}}_{t_k} - \bar{\mathbf{U}}_{t_k}^\infty \right) \leq -(a - \eta) \left\| \bar{\mathbf{U}}_{t_k} - \bar{\mathbf{U}}_{t_k}^\infty \right\|_{\mathfrak{M}}^2 \, ,$$

and using Proposition B.1 there exists $C > 0$ a universal constant (depending only on the eigenvalues of the matrix $\mathfrak{M}$ or constant factors) such that

$$\left( \bar{\mathbf{U}}_{t_k} - \bar{\mathbf{U}}_{t_k}^\infty \right)^\top \mathfrak{M}\Sigma^2 \left( \tilde{\mathbf{s}}_{T-t_k} \left( \bar{\mathbf{U}}_{t_k} \right) - \tilde{\mathbf{s}}_{T-t_k} \left( \bar{\mathbf{U}}_{t_k}^\infty \right) \right) \leq C\tilde{L}_{T-t_k} \left\| \bar{\mathbf{U}}_{t_k} - \bar{\mathbf{U}}_{t_k}^\infty \right\|_{\mathfrak{M}}^2 \, ,$$

it follows that

$$\mathbb{E}\left[ C_k \right] \leq h \left( C\tilde{L}_{T-t_k} - (a - \eta) \right) \mathbb{E}\left[ \left\| \bar{\mathbf{U}}_{t_k} - \bar{\mathbf{U}}_{t_k}^\infty \right\|_{\mathfrak{M}}^2 \right] \, .$$

As a consequence,

$$\mathbb{E}\left[ \left\| \bar{\mathbf{U}}_{t_N} - \bar{\mathbf{U}}_{t_N}^\infty \right\|_{\mathfrak{M}}^2 \right] \leq \mathbb{E}\left[ \left\| \bar{\mathbf{U}}_0 - \bar{\mathbf{U}}_0^\infty \right\|_{\mathfrak{M}}^2 \right] \prod_{\ell=0}^{N-1} \delta_\ell' \, ,$$

with $\delta_\ell' = 1 + h(C\tilde{L}_{T-t_k} - (a - \eta))$. Since $\exp(x) \geq 1 + x$, for $x \in \mathbb{R}$, we have that

$$\prod_{\ell=0}^{N-1} \delta_\ell' \leq \mathrm{e}^{\sum_{k=0}^{N-1} h \left( C\tilde{L}_{T-t_k} - (a-\eta) \right)}$$

$$\leq \mathrm{e}^{-(a-\eta)T + C \sum_{k=0}^{N-1} h\tilde{L}_{T-t_k}}$$

$$\leq \mathrm{e}^{-(a-\eta)T + C \int_0^\infty \tilde{L}_s \mathrm{d}s}$$

$$\leq \mathrm{e}^{-(a-\eta)T + Ca^{-1}} \, ,$$

thus,

$$\mathbb{E}\left[ \left\| \bar{\mathbf{U}}_{t_N} - \bar{\mathbf{U}}_{t_N}^\infty \right\|_{\mathfrak{M}}^2 \right] \leq \mathrm{e}^{Ca^{-1}} \mathrm{e}^{-(a-\eta)T} \mathbb{E}\left[ \left\| \bar{\mathbf{U}}_0 - \bar{\mathbf{U}}_0^\infty \right\|_{\mathfrak{M}}^2 \right] \, ,$$

which implies, taking the limit as $\Delta \to 0$ together with Fatou's lemma, that

$$\mathcal{W}_2^2 \left( \mathcal{L}\left( \bar{\mathbf{U}}_T \right), \mathcal{L}\left( \bar{\mathbf{U}}_T^\infty \right) \right) \leq C\mathrm{e}^{Ca^{-1}} \mathrm{e}^{-(a-\eta)T} \left\| \bar{\mathbf{U}}_0 - \bar{\mathbf{U}}_0^\infty \right\|_{L_2}^2 \, .$$

Moreover, similarly to the backward, the forward process also satisfies the following contraction property (Lemma A.4 ),

$$\left\| \bar{\mathbf{U}}_0 - \bar{\mathbf{U}}_0^\infty \right\|_{L_2} = \mathcal{W}_2 \left( \pi_\infty, \mathcal{L}\left( \vec{\mathbf{U}}_T \right) \right) \leq CT\mathrm{e}^{-(a-\eta)T} \mathcal{W}_2 \left( \pi_{\mathrm{data}} \otimes \pi_v, \pi_\infty \right) \, ,$$

yielding (44). $\qquad \square$

## C    Proof of Theorem 3.2

In this section, we prove Theorem 3.2. To establish this result, we work with the (unmodified) score function rather than the modified one used previously. Similarly to the previous section, we introduce

the continuous time interpolation $(\bar{\mathbf{U}}_t)_{t\in[0,T]}$ of the Euler scheme for the time-reversed process $(\overleftarrow{\mathbf{U}}_t)_{t\in[0,T]}$ defined as the Itô process, for $t \in [t_k, t_{k+1}]$,

$$\bar{\mathbf{U}}_t = \bar{\mathbf{U}}_{t_k} + \left(-A\bar{\mathbf{U}}_{t_k} + \Sigma_\varepsilon^2 \mathsf{s}_{T-t_k}\left(\bar{\mathbf{U}}_{t_k}\right)\right)(t - t_k) + \Sigma_\varepsilon\left(B_t - B_{t_k}\right), \tag{45}$$

when initialized at $p_T$ (*i.e.*, $\bar{\mathbf{U}}_0 \sim p_T$). When initialized at $\pi_\infty$, we write $(\bar{\mathbf{U}}_t^\infty)_{t\in[0,T]}$ this Itô process. We also introduce the continuous time Euler scheme $(\bar{\mathbf{U}}_t^\theta)_{t\in[0,T]}$ in which the true, unknown score function is replaced by a neural network approximation $s_\theta$, and defined for $t \in [t_k, t_{k+1}]$ as

$$\bar{\mathbf{U}}_t^\theta = \bar{\mathbf{U}}_{t_k}^\theta + \left(-A\bar{\mathbf{U}}_{t_k}^\theta + \Sigma_\varepsilon^2 s_\theta\left(t_k, \bar{\mathbf{U}}_{t_k}^\theta\right)\right)(t - t_k) + \Sigma_\varepsilon\left(B_t - B_{t_k}\right), \tag{46}$$

where $\bar{\mathbf{U}}_0^\theta \sim \pi_\infty$.

We first establish the propagation of regularity properties: strong log-concavity propagation (Proposition C.1) and Lipschitz regularity propagation (Proposition C.2), followed by the proof of Theorem 3.2. To this end, we decompose the generation error into the sum of the discretization error (Lemma C.3), the approximation error (Lemma C.4), and the mixing time error (Lemma C.5), as in Theorem 3.1.

### C.1 Propagation of the regularity assumptions

**Proposition C.1.** *Assume that H2′ holds. Then for all $t \in [0, T]$ and all $u \in \mathbb{R}^{2d}$,*

$$\nabla^2 \log p_t(u) \preccurlyeq -\alpha_t \mathbf{I}_{2d},$$

*where*

$$\alpha_t = \left(\frac{1}{(\alpha_0 \wedge v^{-2})\sigma_{\min}^2(e^{-tA})} + \lambda_{\max}(\Sigma_{0,t})\right)^{-1}. \tag{47}$$

*Proof.* Similar to Proposition B.1 recall the following equality in law given by the modified kinetic OU process (9)

$$\overrightarrow{\mathbf{U}}_t \stackrel{\mathcal{L}}{=} e^{tA}\overrightarrow{\mathbf{U}}_0 + \sqrt{\Sigma_{0,t}}G,$$

with $\overrightarrow{\mathbf{U}}_0 \sim \pi_{\text{data}} \otimes \pi_v$, $G \sim \mathcal{N}(0, \mathbf{I}_{2d})$, where $G$ and $\overrightarrow{\mathbf{U}}_0$ are independent, and $\Sigma_{0,t}$ is defined in (20). Writing $q_{t|0}$ the conditional density of $\overrightarrow{\mathbf{U}}_t$ given $\overrightarrow{\mathbf{U}}_0$, we have

$$p_t(u_t) = \det\left(e^{-tA}\right)\int_{\mathbb{R}^{2d}} p_0\left(e^{-tA}z\right)\det\left(2\pi\Sigma_{0,t}\right)^{-1/2}\exp\left(-\frac{1}{2}\left(u_t - z\right)^\top \Sigma_{0,t}^{-1}\left(u_t - z\right)\right)\mathrm{d}z.$$

Since $\pi_{\text{data}}$ is $\alpha_0$–strongly log-concave and $\pi_v$ is a centered Gaussian with covariance $v^2\mathbf{I}_d$, their product (*i.e.* the probability density function of $\overrightarrow{\mathbf{U}}_0$) satisfies

$$\nabla^2 \log p_0(z) \preccurlyeq -\left(\alpha_0 \wedge v^{-2}\right)\mathbf{I}_{2d}.$$

Consequently, for any $z \in \mathbb{R}^{2d}$,

$$\nabla^2 \log p_0(e^{-tA}z) \preccurlyeq -(\alpha_0 \wedge v^{-2})(e^{-tA})^\top e^{-tA}.$$

Finally, using Saumard and Wellner (2014), $p_t$ is strongly log-concave with constant

$$\alpha_t = \left(\frac{1}{(\alpha_0 \wedge v^{-2})\sigma_{\min}^2(e^{-tA})} + \lambda_{\max}(\Sigma_{0,t})\right)^{-1}.$$

$\square$

**Proposition C.2.** *Assume that H2′ holds. Then, for all $t > 0$, $\nabla \log p_t$ is $L_t$-Lipschitz: for all $u \in \mathbb{R}^{2d}$,*

$$\left\|\nabla^2 \log p_t(u)\right\| \leq L_t \leq \min\left\{\mathfrak{h}_{1,t}; \mathfrak{h}_{2,t}\right\}.$$

*where*

$$\mathfrak{h}_{1,t} = \left(1 + (a+1)^2 t\right)^2 e^{2ta}\max\left\{L_0, v^{-2}\right\}$$

$$\mathfrak{h}_{2,t} = \frac{4}{\lfloor \sigma^2\min\{a, 1/a\} - (\sigma^2\max\{a, 1/a\} + 5\varepsilon^2 a^{-1})e^{-2at}\rfloor_+}.$$

*Proof.* Following "*Step 1: Lower bound on $\nabla^2 \log p_t$*" in the proof of Proposition B.1, we obtain for all $t > 0$

$$\nabla^2 \log p_t(u) \succcurlyeq -\min\{\mathfrak{h}_{1,t}; \mathfrak{h}_{2,t}\}\,\mathbf{I}_{2d}\,,$$

where

$$\mathfrak{h}_{1,t} = \left(1 + (a+1)^2 t\right)^2 \mathrm{e}^{2ta}\max\{L_0, v^{-2}\}$$

$$\mathfrak{h}_{2,t} = \frac{4}{\lfloor \sigma^2 \min\{a, 1/a\} - (\sigma^2 \max\{a, 1/a\} + 5\varepsilon^2 a^{-1})\,\mathrm{e}^{-2at}\rfloor_+}\,.$$

Moreover, Proposition C.1 implies that

$$\nabla^2 \log p_t(u) \preccurlyeq -\alpha_t \mathbf{I}_{2d}$$
$$\preccurlyeq 0_{2d \times 2d}\,,$$

where $\alpha_t$ defined as in (47). Consequently,

$$\left\|\nabla^2 \log p_t(u)\right\| \le \min\{\mathfrak{h}_{1,t}; \mathfrak{h}_{2,t}\}\,.$$

$\square$

## C.2 Proofs of the main results

**Lemma C.3** (Discretization error). *Assume that H2′ holds and let $\varepsilon > 0$. If the step size $h$ satisfies*

$$0 < h < \frac{2\min_k \alpha_{t_k}\left(\sigma^2 \wedge \varepsilon^2\right) - (\sigma - \varepsilon)^2 \max_k L_{t_k} - (a+1)^2}{\|A\|^2 + (\varepsilon^4 + \sigma^4)\max_k L_{t_k}^2 + 2\left(\sigma^2 \vee \varepsilon^2\right)\|A\|\max_k L_{t_k}}\,,$$

*then, there exists $\delta_\varepsilon > 0$ such that $\mathcal{W}_2\left(\mathcal{L}(\overleftarrow{\mathbf{U}}_T), \mathcal{L}(\bar{\mathbf{U}}_T)\right) \le 2\sqrt{h}C_a(\varepsilon)/\delta_\varepsilon$ where*

$$C_a(\varepsilon) = \left(2\|A\|^4 B_\varepsilon + 4d(a^2\sigma^2 + \varepsilon)^2 \Lambda_\varepsilon^*(T)\right)h + 4d\left(\|A\|^2 + \sigma^4 \sup_{t \in [0,T]} L_{T-t}^2\right), \qquad (48)$$

*with*

$$\Lambda_\varepsilon^*(T) = \min\left\{\frac{2a\left(1 + (a+1)^2 T\right)^2}{\min\{\varepsilon^2, \sigma^2\}}, \frac{4}{\sigma^2 \min\{a, 1/a\} - (\sigma^2 \max\{a, 1/a\} + 5a\varepsilon^{-2})\mathrm{e}^{-2aT}}\right\},$$

*such that $\sup_{T>0} \Lambda_\varepsilon^*(T) < +\infty$ and*

$$B_\varepsilon := \max_{t \in [0,T]}\left(1 + (a+1)^2(T-t)\right)^2 \mathrm{e}^{-2a(T-t)}\|\overrightarrow{\mathbf{U}}_0\|_{L^2}^2 + \frac{d}{2}\left(\sigma^2 \max\{a, 1/a\} + \frac{5\varepsilon^2}{a}\right). \qquad (49)$$

*Proof.* Consider a synchronous coupling for $(\overleftarrow{\mathbf{U}}_t)_{t \in [0,T]}$ and $(\bar{\mathbf{U}}_t)_{t \in [0,T]}$ *i.e.*, use the same Brownian motion to drive the two processes, with the same initial point, *i.e.*, $\overleftarrow{\mathbf{U}}_0 = \bar{\mathbf{U}}_0$. Then it holds, that

$$\mathcal{W}_2\left(\mathcal{L}(\overleftarrow{\mathbf{U}}_T), \mathcal{L}(\bar{\mathbf{U}}_T)\right) \le \left\|\overleftarrow{\mathbf{U}}_T - \bar{\mathbf{U}}_T\right\|_{L_2}\,.$$

Fix $\Delta \ge 0$ such that $t_N = T - \Delta$ and note that for all $0 \le k \le N-1$,

$$\left\|\overleftarrow{\mathbf{U}}_{t_{k+1}} - \bar{\mathbf{U}}_{t_{k+1}}\right\|_{L_2}$$

$$= \left\|\overleftarrow{\mathbf{U}}_{t_k} - \bar{\mathbf{U}}_{t_k} + \int_{t_k}^{t_{k+1}}\left\{-A\left(\overleftarrow{\mathbf{U}}_t - \bar{\mathbf{U}}_{t_k}\right) + \Sigma_\varepsilon^2\left(\mathsf{s}_{T-t}\left(\overleftarrow{\mathbf{U}}_t\right) - \mathsf{s}_{T-t_k}\left(\bar{\mathbf{U}}_{t_k}\right)\right)\right\}\mathrm{d}t\right\|_{L_2}$$

$$\le A_{1,k} + A_{2,k}\,,$$

where

$$A_{1,k} = \left\|\overleftarrow{\mathbf{U}}_{t_k} - \bar{\mathbf{U}}_{t_k} + \int_{t_k}^{t_{k+1}}\left\{-A\left(\overleftarrow{\mathbf{U}}_{t_k} - \bar{\mathbf{U}}_{t_k}\right) + \Sigma_\varepsilon^2\left(\mathsf{s}_{T-t_k}\left(\overleftarrow{\mathbf{U}}_{t_k}\right) - \mathsf{s}_{T-t_k}\left(\bar{\mathbf{U}}_{t_k}\right)\right)\right\}\mathrm{d}t\right\|_{L_2}\,,$$

$$A_{2,k} = \left\|\int_{t_k}^{t_{k+1}}\left\{-A\left(\overleftarrow{\mathbf{U}}_t - \overleftarrow{\mathbf{U}}_{t_k}\right) + \Sigma_\varepsilon^2\left(\mathsf{s}_{T-t}\left(\overleftarrow{\mathbf{U}}_t\right) - \mathsf{s}_{T-t_k}\left(\overleftarrow{\mathbf{U}}_{t_k}\right)\right)\right\}\mathrm{d}t\right\|_{L_2}\,.$$

For the first term, note that,

$$A_{1,k}^2 = \left\| (\mathbf{I}_{2d} - hA)\left(\overleftarrow{\mathbf{U}}_{t_k} - \bar{\mathbf{U}}_{t_k}\right) + h\Sigma_\varepsilon^2\left(\mathsf{s}_{T-t_k}\left(\overleftarrow{\mathbf{U}}_{t_k}\right) - \mathsf{s}_{T-t_k}\left(\bar{\mathbf{U}}_{t_k}\right)\right)\right\|_{L_2}^2$$

$$= \left\| (\mathbf{I}_{2d} - hA)\left(\overleftarrow{\mathbf{U}}_{t_k} - \bar{\mathbf{U}}_{t_k}\right)\right\|_{L_2}^2 + \left\| h\Sigma_\varepsilon^2\left(\mathsf{s}_{T-t_k}\left(\overleftarrow{\mathbf{U}}_{t_k}\right) - \mathsf{s}_{T-t_k}\left(\bar{\mathbf{U}}_{t_k}\right)\right)\right\|_{L_2}^2$$

$$+ 2h\mathbb{E}\left[\left(\overleftarrow{\mathbf{U}}_{t_k} - \bar{\mathbf{U}}_{t_k}\right)^\top (\mathbf{I}_{2d} - hA)^\top \Sigma_\varepsilon^2\left(\mathsf{s}_{T-t_k}\left(\overleftarrow{\mathbf{U}}_{t_k}\right) - \mathsf{s}_{T-t_k}\left(\bar{\mathbf{U}}_{t_k}\right)\right)\right].$$

By Proposition C.2, it follows that the score at time $t$ is $L_t-$Lipschitz continuous for all $t \in [0,T]$, in particular,

$$\left\| h\Sigma_\varepsilon^2\left(\mathsf{s}_{T-t_k}\left(\overleftarrow{\mathbf{U}}_{t_k}\right) - \mathsf{s}_{T-t_k}\left(\bar{\mathbf{U}}_{t_k}\right)\right)\right\|_{L_2}^2 \leq h^2(\varepsilon^4 + \sigma^4)L_{T-t_k}^2\left\|\overleftarrow{\mathbf{U}}_{t_k} - \bar{\mathbf{U}}_{t_k}\right\|_{L_2}^2.$$

Therefore,

$$A_{1,k}^2 \leq \mathbb{E}\left[\left(\overleftarrow{\mathbf{U}}_{t_k} - \bar{\mathbf{U}}_{t_k}\right)^\top \left((\mathbf{I}_{2d} - hA)^\top(\mathbf{I}_{2d} - hA) + h^2(\varepsilon^4 + \sigma^4)L_{T-t_k}^2\mathbf{I}_{2d}\right)\left(\overleftarrow{\mathbf{U}}_{t_k} - \bar{\mathbf{U}}_{t_k}\right)\right]$$

$$+ 2h\mathbb{E}\left[\left(\overleftarrow{\mathbf{U}}_{t_k} - \bar{\mathbf{U}}_{t_k}\right)^\top (\mathbf{I}_{2d} - hA)^\top \Sigma_\varepsilon^2\left(\mathsf{s}_{T-t_k}\left(\overleftarrow{\mathbf{U}}_{t_k}\right) - \mathsf{s}_{T-t_k}\left(\bar{\mathbf{U}}_{t_k}\right)\right)\right].$$

For all $0 \leq t \leq T$, let $C_{t,k} := \int_0^1 \nabla^2 \log p_t(\overleftarrow{\mathbf{U}}_{t_k} - \gamma(\overleftarrow{\mathbf{U}}_{t_k} - \bar{\mathbf{U}}_{t_k}))\mathrm{d}\gamma$ and write $\mathbf{A}_h = \mathbf{I}_{2d} - hA$ so that,

$$A_{1,k}^2 \leq \mathbb{E}\left[\left(\overleftarrow{\mathbf{U}}_{t_k} - \bar{\mathbf{U}}_{t_k}\right)^\top \left(\mathbf{A}_h^\top\mathbf{A}_h + h^2(\varepsilon^4 + \sigma^4)L_{T-t_k}^2\mathbf{I}_{2d} + 2h\mathbf{A}_h^\top\Sigma_\varepsilon^2 C_{T-t_k,k}\right)\left(\overleftarrow{\mathbf{U}}_{t_k} - \bar{\mathbf{U}}_{t_k}\right)\right]$$

$$\leq \left\|\overleftarrow{\mathbf{U}}_{t_k} - \bar{\mathbf{U}}_{t_k}\right\|_{L_2}^2 + h\left(\overleftarrow{\mathbf{U}}_{t_k} - \bar{\mathbf{U}}_{t_k}\right)^\top M_h(\varepsilon)\left(\overleftarrow{\mathbf{U}}_{t_k} - \bar{\mathbf{U}}_{t_k}\right),$$

where

$$M_h(\varepsilon) = -(A^\top + A) + 2\Sigma_\varepsilon^2 C_{T-t_k,k} + h\left(A^\top A + (\varepsilon^4 + \sigma^4)L_{T-t_k}^2\mathbf{I}_{2d} - 2A^\top\Sigma_\varepsilon^2 C_{T-t_k,k}\right).$$

In order to control $(\overleftarrow{\mathbf{U}}_{t_k} - \bar{\mathbf{U}}_{t_k})^\top M_h(\varepsilon)(\overleftarrow{\mathbf{U}}_{t_k} - \bar{\mathbf{U}}_{t_k})$, it is enough to control the eigenvalues of $\tilde{M}_h(\varepsilon)$ where

$$\tilde{M}_h(\varepsilon) = \frac{1}{2}(M_h(\varepsilon) + M_h(\varepsilon)^\top)$$

$$= -(A^\top + A) + (\Sigma_\varepsilon^2 C_{T-t_k,k} + C_{T-t_k,k}\Sigma_\varepsilon^2)$$

$$+ h\left\{A^\top A + (\varepsilon^4 + \sigma^4)L_{T-t_k}^2\mathbf{I}_{2d} - \left(A^\top\Sigma_\varepsilon^2 C_{T-t_k,k} + C_{T-t_k,k}\Sigma_\varepsilon^2 A\right)\right\}.$$

Noting that,

$$(\Sigma_\varepsilon^2 C_{T-t_k,k} + C_{T-t_k,k}\Sigma_\varepsilon^2) = 2\Sigma_\varepsilon C_{T-t_k,k}\Sigma_\varepsilon + \Sigma_\varepsilon^2 C_{T-t_k,k} + C_{T-t_k,k}\Sigma_\varepsilon^2 - 2\Sigma_\varepsilon C_{T-t_k,k}\Sigma_\varepsilon$$

By Proposition C.1,

$$\Sigma_\varepsilon C_{T-t_k,k}\Sigma_\varepsilon \preccurlyeq -\alpha_{T-t_k}\lambda_{\min}\left(\Sigma_\varepsilon^2\right)\mathbf{I}_{2d}$$

$$\preccurlyeq -\alpha_{T-t_k}\left(\sigma^2 \wedge \varepsilon^2\right)\mathbf{I}_{2d},$$

and simple calculations yields

$$\Sigma_\varepsilon^2 C_{T-t_k,k} + C_{T-t_k,k}\Sigma_\varepsilon^2 - 2\Sigma_\varepsilon C_{T-t_k,k}\Sigma_\varepsilon = (\sigma - \varepsilon)^2\begin{pmatrix} 0_{d\times d} & C_{T-t_k,k}^{12} \\ C_{T-t_k,k}^{12}{}^\top & 0_{d\times d} \end{pmatrix},$$

where $C_{T-t_k,k}^{12}$ denotes the block anti diagonal elements of $C_{T-t_k}$. Hence,

$$\Sigma_\varepsilon^2 C_{T-t_k,k} + C_{T-t_k,k}\Sigma_\varepsilon^2 - 2\Sigma_\varepsilon C_{T-t_k,k}\Sigma_\varepsilon \preccurlyeq (\sigma - \varepsilon)^2\|C_{T-t_k,k}\|\mathbf{I}_{2d}$$

$$\preccurlyeq (\sigma - \varepsilon)^2 L_{T-t_k}\mathbf{I}_{2d},$$

where we used Proposition C.2 in the last line. It follows that,

$$\tilde{M}_h(\varepsilon) \preceq -\lambda_{\min}(A^\top + A)\mathbf{I}_{2d} - 2\alpha_{T-t_k}\left(\sigma^2 \wedge \varepsilon^2\right)\mathbf{I}_{2d} + (\sigma - \varepsilon)^2 L_{T-t_k}\mathbf{I}_{2d}$$
$$+ h\left(\|A\|^2 + (\varepsilon^4 + \sigma^4)L_{T-t_k}^2 + 2\left(\sigma^2 \vee \varepsilon^2\right)\|A\|L_{T-t_k}\right)\mathbf{I}_{2d}.$$

Therefore, using that $\lambda_{\min}(A^\top + A) = -(a+1)^2$, $\tilde{M}_h(\varepsilon)$ is negative when $h$ is chosen so that

$$h < \frac{2\min_k \alpha_{t_k}\left(\sigma^2 \wedge \varepsilon^2\right) - (\sigma - \varepsilon)^2 \max_k L_{t_k} - (a+1)^2}{\|A\|^2 + (\varepsilon^4 + \sigma^4)\max_k L_{t_k}^2 + 2\left(\sigma^2 \vee \varepsilon^2\right)\|A\|\max_k L_{t_k}}. \tag{50}$$

It follows that when $h$ satisfies (50), there exists $\delta_\varepsilon > 0$ such that

$$A_{1,k} \leq \sqrt{1 - h\delta_\varepsilon}\left\|\overleftarrow{\mathbf{U}}_{t_k} - \bar{\mathbf{U}}_{t_k}\right\|_{L_2}.$$

For the second term $A_{2,k}$, note the backward drift function as $b(t,u) = -Au + \Sigma_\varepsilon^2 \mathsf{s}_{T-t}(u)$ so that

$$A_{2,k}^2 = \mathbb{E}\left[\left\|\int_{t_k}^{t_{k+1}}\left(b(t,\overleftarrow{\mathbf{U}}_t) - b(t_k,\overleftarrow{\mathbf{U}}_{t_k})\right)\mathrm{d}t\right\|^2\right].$$

Applying Lemma D.6 combining with Itô's formula, we obtain

$$\mathrm{d}b(t,\overleftarrow{\mathbf{U}}_t) = -A\mathrm{d}\overleftarrow{\mathbf{U}}_t + \Sigma_\varepsilon^2\mathrm{d}\mathsf{s}_{T-t}(\overleftarrow{\mathbf{U}}_t)$$
$$= \left\{AA\overleftarrow{\mathbf{U}}_t - A\Sigma_\varepsilon^2\mathsf{s}_{T-t}(\overleftarrow{\mathbf{U}}_t) + \Sigma_\varepsilon^2 A^T\mathsf{s}_{T-t}(\overleftarrow{\mathbf{U}}_t)\right\}\mathrm{d}t + \left(A + \Sigma_\varepsilon^2\nabla^2\log p_{T-t}(\overleftarrow{\mathbf{U}}_t)\right)\Sigma_\varepsilon\mathrm{d}B_t$$
$$= AA\overleftarrow{\mathbf{U}}_t\mathrm{d}t + \left(\Sigma_\varepsilon^2 A^\top - A\Sigma_\varepsilon^2\right)\mathsf{s}_{T-t}(\overleftarrow{\mathbf{U}}_t)\mathrm{d}t + \left(A + \Sigma_\varepsilon^2\nabla^2\log p_{T-t}(\overleftarrow{\mathbf{U}}_t)\right)\Sigma_\varepsilon\mathrm{d}B_t.$$

Using $H_s = AA\overleftarrow{\mathbf{U}}_s + \left(\Sigma_\varepsilon^2 A^\top - A\Sigma_\varepsilon^2\right)\mathsf{s}_{T-s}(\overleftarrow{\mathbf{U}}_s)$ and $K_s = \left(A + \Sigma_\varepsilon^2\nabla^2\log p_{T-s}(\overleftarrow{\mathbf{U}}_s)\right)\Sigma_\varepsilon$ we have that

$$A_{2,k}^2 = \mathbb{E}\left[\left\|\int_{t_k}^{t_{k+1}}\int_{t_k}^t H_s\mathrm{d}s\mathrm{d}t + \int_{t_k}^{t_{k+1}}\int_{t_k}^t K_s\mathrm{d}B_s\mathrm{d}t\right\|^2\right],$$
$$\leq 2h\int_{t_k}^{t_{k+1}}\mathbb{E}\left[\left\|\int_{t_k}^t H_s\mathrm{d}s\right\|^2\mathrm{d}t\right] + 2h^2\mathbb{E}\left[\sup_{t\in[t_k,t_{k+1}]}\left\|\int_{t_k}^t K_s\mathrm{d}B_s\right\|^2\right],$$

by convexity of $\|\cdot\|^2$. Using again the convexity (or applying Cauchy-Schwartz inequality) we have $\mathbb{E}[\|\int_{t_k}^t H_s\mathrm{d}s\|^2] \leq h\int_{t_k}^t\mathbb{E}[\|H_s\|^2]\mathrm{d}s$ and then

$$A_{2,k}^2 \leq h^4\sup_{t\in[t_k,t_{k+1}]}\mathbb{E}\left[\|H_t\|^2\right] + 2h^2\mathbb{E}\left[\sup_{t\in[t_k,t_{k+1}]}\left\|\int_{t_k}^t K_s\mathrm{d}B_s\right\|^2\right]. \tag{51}$$

First we have for $t \in [0,T]$,

$$\mathbb{E}\left[\|H_t\|^2\right] \leq 2\|A\|_2^4\mathbb{E}\left[\|\overleftarrow{\mathbf{U}}_t\|^2\right] + 2\left\|\Sigma_\varepsilon^2 A^\top - A\Sigma_\varepsilon^2\right\|^2\mathbb{E}\left[\left\|\mathsf{s}_{T-t}(\overleftarrow{\mathbf{U}}_t)\right\|^2\right],$$

and by Lemma D.3 and Lemma D.5 we get

$$\mathbb{E}\left[\|H_t\|^2\right] \leq 2\|A\|^4 B_\varepsilon + 2\left\|\Sigma_\varepsilon^2 A^\top - A\Sigma_\varepsilon^2\right\|^2\frac{2d}{\lambda_{\min}(\Sigma_{0,T-t})},$$

where $B_\varepsilon$ is defined in (49). By Lemma A.3 we get

$$\max_{t\in[0,T]}\mathbb{E}\left[\|H_t\|^2\right] \leq 2\|A\|^4 B_\varepsilon + 4d(a^2\sigma^2 + \varepsilon)^2\Lambda_\varepsilon^*(T), \tag{52}$$

with

$$\Lambda_\varepsilon^*(T) = \min\left\{\frac{2a\left(1 + (a+1)^2 T\right)^2}{\min\{\varepsilon^2,\sigma^2\}}, \frac{4}{\sigma^2\min\{a,1/a\} - \left(\sigma^2\max\{a,1/a\} + 5a^{-1}\varepsilon^2\right)\mathrm{e}^{-2aT}}\right\},$$

such that $\sup_{T>0} \Lambda_\varepsilon^*(T) < +\infty$.

Now by Doob's inequality and Itô's isometry, we have

$$\mathbb{E}\left[\sup_{t\in[t_k,t_{k+1}]}\left\|\int_{t_k}^t K_s \mathrm{d}B_s\right\|^2\right] = \int_{t_k}^{t_{k+1}} \mathbb{E}\left[\left\|A + \Sigma_\varepsilon^2 \nabla^2 \log p_{T-s}(\overleftarrow{\mathbf{U}}_s)\right\|_F^2\right]\mathrm{d}s\,,$$

where $\|\cdot\|_F$ is the Frobenius norm, so that

$$\mathbb{E}\left[\sup_{t\in[t_k,t_{k+1}]}\left\|\int_{t_k}^t K_s \mathrm{d}B_s\right\|^2\right] \le hd \max_{t\in[t_k,t_{k+1}]} \mathbb{E}\left[\left\|A + \Sigma_\varepsilon^2 \nabla^2 \log p_{T-t}(\overleftarrow{\mathbf{U}}_t)\right\|^2\right]\,.$$

Using the $L_t$–Lipschitz continuous property of the score at time $t$ we have

$$\mathbb{E}\left[\sup_{t\in[t_k,t_{k+1}]}\left\|\int_{t_k}^t K_s \mathrm{d}B_s\right\|^2\right] \le 2hd\left(\|A\|^2 + \max\{\sigma^4,\varepsilon^4\} \sup_{t\in[t_k,t_{k+1}]} L_{T-t_k}^2\right). \tag{53}$$

Plugging (52) and (53) into (51) we obtain

$$A_{2,k}^2 \le C_a(\varepsilon)h^3 \tag{54}$$

with $C_a(\varepsilon)$ defined in (48).

Combining the bound on $A_{1,k}$ to the bound on $A_{2,k}$ yields,

$$\left\|\overleftarrow{\mathbf{U}}_{t_{k+1}} - \bar{\mathbf{U}}_{t_{k+1}}\right\|_{L_2} \le \sqrt{1-h\delta_\varepsilon}\left\|\overleftarrow{\mathbf{U}}_{t_k} - \bar{\mathbf{U}}_{t_k}\right\|_{L_2} + h\sqrt{h}C_a(\varepsilon)\,.$$

Using that $\overleftarrow{\mathbf{U}}_0 - \bar{\mathbf{U}}_0 = 0$ we have by induction

$$\left\|\overleftarrow{\mathbf{U}}_{t_N} - \bar{\mathbf{U}}_{t_N}\right\|_{L_2} \le \sum_{k=0}^{N-1}\prod_{j=k+1}^{N-1}\left(1-h\delta_\varepsilon\right)^{1/2}h\sqrt{h}C_a(\varepsilon)\,,$$

$$\le \frac{2}{\delta_\varepsilon}\sqrt{h}C_a(\varepsilon)\,,$$

since $\sqrt{1-\delta_\varepsilon h} \le 1 - h\delta_\varepsilon/2$. Letting $\Delta \to 0$ together with Fatou's lemma finishes the proof.

$\square$

**Lemma C.4** (Approximation error)**.** *Assume that H2′ and H3 hold. Then, there exists $\delta_\varepsilon > 0$ such that*

$$\mathcal{W}_2\left(\mathcal{L}\left(\bar{\mathbf{U}}_T^\infty\right),\mathcal{L}\left(\bar{\mathbf{U}}_T^\theta\right)\right) \le \frac{2}{\delta_\varepsilon}\max\left\{\varepsilon^2,\sigma^2\right\}M\,.$$

*Proof.* Note that

$$\mathcal{W}_2\left(\mathcal{L}\left(\bar{\mathbf{U}}_T^\infty\right),\mathcal{L}\left(\bar{\mathbf{U}}_T^\theta\right)\right) \le \left\|\bar{\mathbf{U}}_T^\infty - \bar{\mathbf{U}}_T^\theta\right\|_{L_2}\,.$$

Using a decomposition similar to that in C.3, with $t_N = T - \Delta$, we obtain:

$$\left\|\bar{\mathbf{U}}_{t_{k+1}}^\infty - \bar{\mathbf{U}}_{t_{k+1}}^\theta\right\|_{L_2}$$

$$= \left\|\bar{\mathbf{U}}_{t_k}^\infty - \bar{\mathbf{U}}_{t_k}^\theta + \int_{t_k}^{t_{k+1}} -A\left(\bar{\mathbf{U}}_{t_k}^\infty - \bar{\mathbf{U}}_{t_k}^\theta\right) + \Sigma_\varepsilon^2\left(\mathsf{s}_{T-t_k}\left(\bar{\mathbf{U}}_{t_k}^\infty\right) - s_\theta\left(T-t_k,\bar{\mathbf{U}}_{t_k}^\theta\right)\right)\mathrm{d}t\right\|_{L_2}$$

$$\le \left\|\bar{\mathbf{U}}_{t_k}^\infty - \bar{\mathbf{U}}_{t_k}^\theta + \int_{t_k}^{t_{k+1}} -A\left(\bar{\mathbf{U}}_{t_k}^\infty - \bar{\mathbf{U}}_{t_k}^\theta\right) + \Sigma_\varepsilon^2\left(\mathsf{s}_{T-t_k}\left(\bar{\mathbf{U}}_{t_k}^\infty\right) - \mathsf{s}_{T-t_k}\left(\bar{\mathbf{U}}_{t_k}^\theta\right)\right)\mathrm{d}t\right\|_{L_2}$$

$$+ \left\|\int_{t_k}^{t_{k+1}} \Sigma_\varepsilon^2\left(\mathsf{s}_{T-t_k}\left(\bar{\mathbf{U}}_{t_k}^\theta\right) - s_\theta\left(T-t_k,\bar{\mathbf{U}}_{t_k}^\theta\right)\right)\mathrm{d}t\right\|_{L_2}$$

$$\le B_{1,k} + B_{2,k}\,.$$

For the first term, note that,

$$B_{1,k}^2 = \left\| (\mathbf{I}_{2d} - hA)\left(\bar{\mathbf{U}}_{t_k}^\infty - \bar{\mathbf{U}}_{t_k}^\theta\right) + h\Sigma_\varepsilon^2\left(\mathsf{s}_{T-t_k}\left(\bar{\mathbf{U}}_{t_k}^\infty\right) - \mathsf{s}_{T-t_k}\left(\bar{\mathbf{U}}_{t_k}^\theta\right)\right) \right\|_{L_2}^2$$

$$= \left\| (\mathbf{I}_{2d} - hA)\left(\bar{\mathbf{U}}_{t_k}^\infty - \bar{\mathbf{U}}_{t_k}^\theta\right) \right\|_{L_2}^2 + \left\| h\Sigma_\varepsilon^2\left(\mathsf{s}_{T-t_k}\left(\bar{\mathbf{U}}_{t_k}^\infty\right) - \mathsf{s}_{T-t_k}\left(\bar{\mathbf{U}}_{t_k}^\theta\right)\right) \right\|_{L_2}^2$$

$$+ 2h\mathbb{E}\left[\left(\bar{\mathbf{U}}_{t_k}^\infty - \bar{\mathbf{U}}_{t_k}^\theta\right)^\top (\mathbf{I}_{2d} - hA)^\top \Sigma_\varepsilon^2\left(\mathsf{s}_{T-t_k}\left(\bar{\mathbf{U}}_{t_k}^\infty\right) - \mathsf{s}_{T-t_k}\left(\bar{\mathbf{U}}_{t_k}^\theta\right)\right)\right] .$$

It follows that $B_{1,k}$ can be treated similarly to $A_{1,k}$. Using H2, H2′, and for $h$ satisfying (50), we have

$$B_{1,k} \leq \sqrt{1 - h\delta_\varepsilon} \left\| \bar{\mathbf{U}}_{t_k}^\infty - \bar{\mathbf{U}}_{t_k}^\theta \right\|_{L_2} ,$$

where $\delta_\varepsilon$ is defined as in the proof of Lemma C.3. For $B_{2,k}$, using Assumption H3, we get

$$B_{2,k} \leq h \left\| \Sigma_\epsilon \right\|^2 M \leq h \max\left\{\varepsilon^2, \sigma^2\right\} M .$$

Finally, for $h$ satisfying (50), it follows from the same argument as in the proof of Lemma C.3 that

$$\left\| \bar{\mathbf{U}}_{t_N}^\infty - \bar{\mathbf{U}}_{t_N}^\theta \right\|_{L_2} \leq \frac{2}{\delta_\varepsilon} \max\left\{\varepsilon^2, \sigma^2\right\} M .$$

Taking the limit as $\Delta \to 0$, toghether with Fatou's lemma finishes the proof. □

**Lemma C.5** (Mixing time). *Assume that H2′ holds. Then*

$$\mathcal{W}_2\left(\mathcal{L}\left(\bar{\mathbf{U}}_T\right), \mathcal{L}\left(\bar{\mathbf{U}}_T^\infty\right)\right) \leq K_T e^{-aT} \mathcal{W}_2\left(\pi_{\text{data}} \otimes \pi_v, \pi_\infty\right) ,$$

*with*

$$K_T := (1 + \max\{a + 1; a(a+1)\}T) .$$

*Proof.* Consider a synchronous coupling of the continuous-time interpolations $(\bar{\mathbf{U}}_t)_{t \in [0,T]}$ and $(\bar{\mathbf{U}}_t^\infty)_{t \in [0,T]}$, with initialization

$$\mathcal{W}_2\left(\pi_\infty, \mathcal{L}\left(\overrightarrow{\mathbf{U}}_T\right)\right) = \left\| \bar{\mathbf{U}}_0 - \bar{\mathbf{U}}_0^\infty \right\|_{L_2} .$$

By definition of the $\mathcal{W}_2$ distance,

$$\mathcal{W}_2\left(\mathcal{L}\left(\bar{\mathbf{U}}_T\right), \mathcal{L}\left(\bar{\mathbf{U}}_T^\infty\right)\right) \leq \left\| \bar{\mathbf{U}}_T - \bar{\mathbf{U}}_T^\infty \right\|_{L_2} .$$

For $t \in [t_k, t_{k+1}]$ and with $t_N = T - \Delta$ we have that,

$$\left\| \bar{\mathbf{U}}_{t_{k+1}} - \bar{\mathbf{U}}_{t_{k+1}}^\infty \right\|_{L_2}$$

$$= \left\| \bar{\mathbf{U}}_{t_k} - \bar{\mathbf{U}}_{t_k}^\infty + \int_{t_k}^{t_{k+1}} -A\left(\bar{\mathbf{U}}_{t_k} - \bar{\mathbf{U}}_{t_k}^\infty\right) + \Sigma^2\left(\mathsf{s}_{T-t_k}\left(\bar{\mathbf{U}}_{t_k}\right) - \mathsf{s}_{T-t_k}\left(\bar{\mathbf{U}}_{t_k}^\infty\right)\right) dt \right\|_{L_2}$$

$$\leq \left\| \bar{\mathbf{U}}_{t_k} - \bar{\mathbf{U}}_{t_k}^\infty \right\|_{L_2} \delta_k ,$$

where $\delta_k$ is defined as in (50). As a consequence,

$$\left\| \bar{\mathbf{U}}_T - \bar{\mathbf{U}}_T^\infty \right\|_{L_2} \leq \left\| \bar{\mathbf{U}}_0 - \bar{\mathbf{U}}_0^\infty \right\|_{L_2} \prod_{\ell=0}^{N-1} \delta_\ell ,$$

where we let $\Delta \to 0$ together with Fatou's lemma. Finally, using Lemma A.4, yields

$$\left\| \bar{\mathbf{U}}_0 - \bar{\mathbf{U}}_0^\infty \right\|_{L_2} = \mathcal{W}_2\left(\pi_\infty, \mathcal{L}\left(\overrightarrow{\mathbf{U}}_T\right)\right) \leq K_T e^{-aT} \mathcal{W}_2\left(\pi_{\text{data}} \otimes \pi_v, \pi_\infty\right) ,$$

which finishes the proof. □

# D  Technical Lemmata

**Lemma D.1.** *Assume that H2 holds. Then, the data distribution $p_{\mathrm{data}}(x) \propto \exp(-(V(x) + H(x)))$ has sub-Gaussian tails,* i.e., *there exist constants $C, \kappa > 0$ such that*

$$p_{\mathrm{data}}(x) \leq C \, \exp(-\kappa \|x\|^2), \qquad x \in \mathbb{R}^d.$$

*In particular, $\pi_{\mathrm{data}}$ admits finite moments of all orders.*

*Proof.* By $\alpha$–strong convexity of $V$, for all $x, y \in \mathbb{R}^d$,

$$V(x) \geq V(y) + \nabla V(y)^\top (x - y) + \frac{\alpha}{2}\|x - y\|^2 .$$

Let $x^*$ denote the unique minimizer of $V$, so that $\nabla V(x^*) = 0$. Then,

$$V(x) \geq V(x^*) + \frac{\alpha}{2}\|x - x^*\|^2 \geq \frac{\alpha}{4}\|x\|^2 - c_1 ,$$

for some constant $c_1 \in \mathbb{R}$. Since $H$ is $L$–Lipschitz, we have

$$H(x) \geq H(x^*) - L\|x - x^*\| \geq -L\|x\| + c_2 ,$$

for some $c_2 \in \mathbb{R}$. Combining these two inequalities yields, for some $C \in \mathbb{R}$,

$$V(x) + H(x) \geq \frac{\alpha}{4}\|x\|^2 - L\|x\| + C .$$

Using Young's inequality $L\|x\| \leq \alpha\|x\|^2/8 + 2L^2/\alpha$, we obtain

$$V(x) + H(x) \geq \frac{\alpha}{8}\|x\|^2 - \frac{2L^2}{\alpha} + C .$$

Hence, up to a multiplicative constant,

$$p_{\mathrm{data}}(x) \; \propto \; \exp(-(V(x) + H(x))) \leq C' \, \exp\!\left( -\frac{\alpha}{8}\|x\|^2 \right),$$

for some $C' > 0$ which concludes the proof. $\qquad\square$

**Lemma D.2.** *Assume that H2 holds and that there exist $m \in \mathbb{N}$ and $C > 0$ such that, for all $x \in \mathbb{R}^d$,*

$$\|\nabla V(x)\| \leq C\,(1 + \|x\|^m) . \tag{55}$$

*Then, the relative Fisher Information between $\pi_0 = \pi_{\mathrm{data}} \otimes \pi_v$ (i.e. the initialization of the stochastic process defined in (4)) and $\pi_\infty$ is finite,* i.e.

$$\mathcal{I}(\pi_0|\pi_\infty) := \int \left\| \nabla \log\left( \frac{\mathrm{d}\pi_0}{\mathrm{d}\pi_\infty}(u) \right) \right\|^2 \pi_0(\mathrm{d}u) < \infty .$$

*Proof.* From Assumption H2, together with the fact that $\pi_0 = \pi_{\mathrm{data}} \otimes \pi_v$ and $\pi_v \sim \mathcal{N}\left(0_d, v^2 \mathbf{I}_d\right)$,

$$p_0(u) = p_{\mathrm{data}}(x)\mathcal{N}(y; 0_d, v^2 \mathbf{I}_d) \propto \mathrm{e}^{-(V(x)+H(x))} \mathrm{e}^{-\frac{\|y\|^2}{2v^2}} .$$

Therefore, the relative Fisher Information satisfies

$$\mathcal{I}(\pi_0|\pi_\infty) = \mathbb{E}\left[ \left\| -\begin{pmatrix} \nabla V(\overrightarrow{X}_0) + \nabla H(\overrightarrow{X}_0) \\ v^{-2}\overrightarrow{V}_0 \end{pmatrix} + \Sigma_\infty^{-1}\begin{pmatrix} \overrightarrow{X}_0 \\ \overrightarrow{V}_0 \end{pmatrix} \right\|^2 \right]$$

$$\leq 2\mathbb{E}\left[ \left\| -\begin{pmatrix} \nabla V(\overrightarrow{X}_0) + \nabla H(\overrightarrow{X}_0) \\ v^{-2}\overrightarrow{V}_0 \end{pmatrix} \right\|^2 \right] + 2\mathbb{E}\left[ \left\| \Sigma_\infty^{-1}\begin{pmatrix} \overrightarrow{X}_0 \\ \overrightarrow{V}_0 \end{pmatrix} \right\|^2 \right] .$$

By Lemma D.1, $\pi_{\mathrm{data}}$ has sub-Gaussian tails, hence

$$\mathbb{E}\left[ \left\| \Sigma_\infty^{-1}\begin{pmatrix} \overrightarrow{X}_0 \\ \overrightarrow{V}_0 \end{pmatrix} \right\|^2 \right] < \infty .$$

Moreover,

$$\mathbb{E}\left[\left\|-\begin{pmatrix}\nabla V(\overrightarrow{X}_0) + \nabla H(\overrightarrow{X}_0) \\ v^{-2}\overrightarrow{V}_0\end{pmatrix}\right\|^2\right] \leq 2\mathbb{E}\left[\left\|\nabla V(\overrightarrow{X}_0)\right\|^2\right] + 2\mathbb{E}\left[\left\|\nabla H(\overrightarrow{X}_0)\right\|^2\right] + v^{-4}\mathbb{E}\left[\left\|\overrightarrow{V}_0\right\|^2\right]$$

Since $\overrightarrow{V}_0$ is Gaussian, $\mathbb{E}[\|\overrightarrow{V}_0\|^2] < \infty$, and by Assumption H2, $H$ is $L$-Lipschitz, so that $\mathbb{E}[\|\nabla H(\overrightarrow{X}_0)\|^2] \leq L^2$. Using (55), there exist $m \in \mathbb{N}$ and $C > 0$ such that

$$\mathbb{E}\left[\left\|\nabla V(\overrightarrow{X}_0)\right\|\right] \leq C\left(1 + \mathbb{E}\left[\left\|\overrightarrow{X}_0\right\|^m\right]\right) < \infty,$$

using sub-Gaussianity of $\pi_{\mathrm{data}}$, which concludes the proof. $\qquad\square$

**Lemma D.3.** *Assume that $(\overrightarrow{\mathbf{U}}_t)_{t\in[0,T]}$ is solution to (9) and that $\overrightarrow{\mathbf{U}}_0$ admits a second order moment, then for all $0 \leq t \leq T$, then for all $\varepsilon \geq 0$,*

$$\left\|\overleftarrow{\mathbf{U}}_t\right\|_{L_2}^2 \leq \left(1 + (a+1)^2(T-t)\right)^2 e^{-2a(T-t)}\left\|\overrightarrow{\mathbf{U}}_0\right\|_{L_2}^2 + \frac{d}{2}\left(\sigma^2 \max\{a, 1/a\} + \frac{5\varepsilon^2}{a}\right) =: B\,. \tag{56}$$

*Proof.* Note that, in distribution,

$$\overrightarrow{\mathbf{U}}_t \overset{\mathcal{L}}{=} e^{tA}\overrightarrow{\mathbf{U}}_0 + \Sigma_{0,t}^{1/2}G\,,$$

with $\overrightarrow{\mathbf{U}}_0 \sim \pi_{\mathrm{data}} \otimes \pi_v$, $G \sim \mathcal{N}(0, \mathbf{I}_{2d})$, and where $G$ and $\overrightarrow{\mathbf{U}}_0$ are independent. Since $G$ and $\overrightarrow{\mathbf{U}}_0$ are independent, using time-reversal and sub-multiplicativity of matrix norms, we have that

$$\begin{aligned}\mathbb{E}\left[\left\|\overleftarrow{\mathbf{U}}_{T-t}\right\|^2\right] = \mathbb{E}\left[\left\|\overrightarrow{\mathbf{U}}_t\right\|^2\right] &= \mathbb{E}\left[\left\|e^{tA}\overrightarrow{\mathbf{U}}_0\right\|^2\right] + \mathbb{E}\left[\left\|\Sigma_{0,t}^{1/2}G\right\|^2\right] \\ &\leq \left\|e^{tA}\right\|^2 \mathbb{E}\left[\left\|\overrightarrow{\mathbf{U}}_0\right\|^2\right] + \left\|\Sigma_{0,t}^{1/2}\right\|^2 \mathbb{E}\left[\|G\|^2\right] \\ &= \left\|e^{tA}\right\|^2 \mathbb{E}\left[\left\|\overrightarrow{\mathbf{U}}_0\right\|^2\right] + 2d\lambda_{\max}(\Sigma_{0,t})\,.\end{aligned}$$

We conclude by applying Lemma A.1 to bound $\left\|e^{tA}\right\|^2$ and Lemma A.3 to bound $\lambda_{\max}(\Sigma_{0,t})$. $\quad\square$

*Remark* D.4. Lemma D.3 holds true when $\overleftarrow{\mathbf{U}}_t$ is defined as in (4) by setting $\varepsilon = 0$.

**Lemma D.5.** *Assume that $(\overrightarrow{\mathbf{U}}_t)_{t\in[0,T]}$ is solution to (9), then,*

$$\mathbb{E}\left[\left\|\mathsf{s}_{T-t}\left(\overleftarrow{\mathbf{U}}_t\right)\right\|^2\right] \leq \frac{2d}{\lambda_{\min}(\Sigma_{0,T-t})}\,,$$

*where $\Sigma_{0,t}$ is defined in (20).*

*Proof.* By the time-reversal property,

$$\mathbb{E}\left[\left\|\mathsf{s}_{T-t}\left(\overleftarrow{\mathbf{U}}_t\right)\right\|^2\right] = \mathbb{E}\left[\left\|\mathsf{s}_{T-t}\left(\overrightarrow{\mathbf{U}}_{T-t}\right)\right\|^2\right]\,.$$

Note that

$$\mathsf{s}_{T-t}(\overrightarrow{\mathbf{U}}_{T-t}) = \mathbb{E}\left[\Sigma_{0,T-t}^{-1}(e^{(T-t)A}\overrightarrow{\mathbf{U}}_0 - \overrightarrow{\mathbf{U}}_{T-t})|\overrightarrow{\mathbf{U}}_{T-t}\right]\,,$$

then, using Jensen's inequality and the tower property,

$$\mathbb{E}\left[\left\|\mathsf{s}_{T-t}\left(\overleftarrow{\mathbf{U}}_t\right)\right\|^2\right] \leq \mathbb{E}\left[\left\|\Sigma_{0,T-t}^{-1}\left(e^{(T-t)A}\overrightarrow{\mathbf{U}}_0 - \overrightarrow{\mathbf{U}}_{T-t}\right)\right\|^2\right]\,.$$

Since $\overrightarrow{\mathbf{U}}_t \overset{\mathcal{L}}{=} e^{tA}\overrightarrow{\mathbf{U}}_0 + \Sigma_{0,t}^{1/2}G$ with $\overrightarrow{\mathbf{U}}_0 \sim \pi_{\mathrm{data}} \otimes \pi_v$, $G \sim \mathcal{N}(0, \mathbf{I}_{2d})$, and where $G$ and $\overrightarrow{\mathbf{U}}_0$ are independent, we have

$$\mathbb{E}\left[\left\|\mathsf{s}_{T-t}\left(\overleftarrow{\mathbf{U}}_t\right)\right\|^2\right] \leq \mathbb{E}\left[\left\|\Sigma_{0,T-t}^{-1/2}G\right\|^2\right] = \mathrm{Tr}\left(\Sigma_{0,T-t}^{-1}\right)\,,$$

which completes the proof.

$\qquad\square$

**Lemma D.6.** *Assume that* $(\overleftarrow{\mathbf{U}}_t)_{t\in[0,T]}$ *is solution to the backward SDE associated with (9). Then,*

$$\mathrm{d}(\nabla \log p_{T-t}(\overleftarrow{\mathbf{U}}_t)) = A^\top \nabla \log p_{T-t}(\overleftarrow{\mathbf{U}}_t)\mathrm{d}t + \nabla^2 \log p_{T-t}(\overleftarrow{\mathbf{U}}_t)\Sigma_\varepsilon \mathrm{d}B_t\,.$$

*Proof.* The Fokker-Plank equation for the SDE defined in (4) yields, for $u \in \mathbb{R}^{2d}$,

$$\partial_t p_t(u) = -\mathrm{div}(Au p_t(u)) + \frac{1}{2}\mathrm{div}(\Sigma_\varepsilon^2 \nabla p_t(u))\,. \tag{57}$$

First, using the notation introduced in (10),

$$
\begin{aligned}
\mathrm{div}(Au p_t(u)) &= \sum_{i=1}^{2d} \frac{\partial Au p_t(u)}{\partial u_i}\\
&= \sum_{i=1}^{2d}\sum_{j=1}^{2d} \frac{\partial}{\partial u_i} A_{ij}u_j p_t(u)\\
&= \sum_{i=1}^{2d} A_{ii} p_t(u) + \sum_{i=1}^{2d}\sum_{j=1}^{2d} A_{ij} u_j \frac{\partial}{\partial_i} p_t(u)\\
&= \sum_{i=1}^{2d} A_{ii} p_t(u) + (Au)^\top \nabla p_t(u)\\
&= \mathrm{Tr}(A) p_t(u) + (Au)^\top \nabla p_t(u)\\
&= p_t(u)\left(\mathrm{Tr}(A) + (Au)^\top \mathsf{s}_t(u)\right)\,.
\end{aligned}
$$

Second, using the product rule for divergence,

$$
\begin{aligned}
\frac{1}{2}\mathrm{div}(\Sigma_\varepsilon^2 \nabla p_t(u)) &= \frac{1}{2}\mathrm{div}(\Sigma_\varepsilon^2 p_t(u)\mathsf{s}_t(u))\\
&= \frac{1}{2}\mathrm{div}(p_t(u)\Sigma_\varepsilon^2 \mathsf{s}_t(u))\\
&= \frac{1}{2}\left(p_t(u)\mathrm{div}(\Sigma_\varepsilon^2 \mathsf{s}_t(u)) + (\Sigma_\varepsilon^2 \mathsf{s}_t(u))^\top \nabla p_t(u)\right)\\
&= \frac{1}{2}p_t(u)\left(\mathrm{div}(\Sigma_\varepsilon^2 \mathsf{s}_t(u)) + \mathsf{s}_t(u)^\top \Sigma_\varepsilon^2 \mathsf{s}_t(u)\right)\,.
\end{aligned}
$$

Hence, dividing (57) by $p_t$ yields

$$\partial_t \log p_t(u) = -\mathrm{Tr}(A) - (Au)^\top \mathsf{s}_t(u) + \frac{1}{2}\left[\left(\mathrm{div}(\Sigma_\varepsilon^2 \mathsf{s}_t(u)) + \mathsf{s}_t(u)^\top \Sigma_\varepsilon^2 \mathsf{s}_t(u)\right)\right]\,,$$

so that,

$$\partial_t \log p_{T-t}(u) = \mathrm{Tr}(A) + (Au)^\top \mathsf{s}_{T-t}(u) - \frac{1}{2}\left[\left(\mathrm{div}(\Sigma_\varepsilon^2 \mathsf{s}_{T-t}(u)) + \mathsf{s}_{T-t}(u)^\top \Sigma_\varepsilon^2 \mathsf{s}_{T-t}(u)\right)\right]\,.$$

Recall that the backward process can be written as

$$\mathrm{d}\overleftarrow{\mathbf{U}}_t = (-A\overleftarrow{\mathbf{U}}_t + \Sigma_\varepsilon^2 \mathsf{s}_{T-t}(\overleftarrow{\mathbf{U}}_t))\mathrm{d}t + \Sigma_\varepsilon \mathrm{d}B_t\,.$$

Hence, by Itô's formula,

$$
\begin{aligned}
\mathrm{d}(\mathsf{s}_{T-t}(\overleftarrow{\mathbf{U}}_t)) &= \partial_t \mathsf{s}_{T-t}(\overleftarrow{\mathbf{U}}_t)\mathrm{d}t + \nabla \mathsf{s}_{T-t}(\overleftarrow{\mathbf{U}}_t)\mathrm{d}U_t + \frac{1}{2}\mathrm{Tr}\left(\Sigma_\varepsilon^2 \nabla^2 \mathsf{s}_{T-t}(\overleftarrow{\mathbf{U}}_t)\right)\mathrm{d}t\\
&= \nabla \partial_t \log p_{T-t}(\overleftarrow{\mathbf{U}}_t)\mathrm{d}t - \nabla \mathsf{s}_{T-t}(\overleftarrow{\mathbf{U}}_t)A\overleftarrow{\mathbf{U}}_t\mathrm{d}t + \nabla \mathsf{s}_{T-t}(\overleftarrow{\mathbf{U}}_t)\Sigma_\varepsilon^2 \mathsf{s}_{T-t}(\overleftarrow{\mathbf{U}}_t)\mathrm{d}t\\
&\quad + \frac{1}{2}\mathrm{Tr}\left(\Sigma_\varepsilon^2 \nabla^2 \mathsf{s}_{T-t}(\overleftarrow{\mathbf{U}}_t)\right)\mathrm{d}t + \nabla \mathsf{s}_{T-t}(\overleftarrow{\mathbf{U}}_t)\Sigma_\varepsilon \mathrm{d}B_t\\
&= \nabla\left(\partial_t \log p_{T-t}(\overleftarrow{\mathbf{U}}_t) + \frac{1}{2}\mathsf{s}_{T-t}(\overleftarrow{\mathbf{U}}_t)^\top \Sigma_\varepsilon^2 \mathsf{s}_{T-t}(\overleftarrow{\mathbf{U}}_t) + \frac{1}{2}\mathrm{div}(\Sigma_\varepsilon^2 \mathsf{s}_{T-t}(\overleftarrow{\mathbf{U}}_t))\right)\mathrm{d}t\\
&\quad - \nabla \mathsf{s}_{T-t}(\overleftarrow{\mathbf{U}}_t)A\overleftarrow{\mathbf{U}}_t\mathrm{d}t + \nabla \mathsf{s}_{T-t}(\overleftarrow{\mathbf{U}}_t)\Sigma_\varepsilon \mathrm{d}B_t\\
&= A^\top \mathsf{s}_{T-t}(\overleftarrow{\mathbf{U}}_t)\mathrm{d}t + \nabla \mathsf{s}_{T-t}(\overleftarrow{\mathbf{U}}_t)\Sigma_\varepsilon \mathrm{d}B_t\,,
\end{aligned}
$$

which completes the proof and where we used that for $u \in \mathbb{R}^{2d}$, $2\nabla^2 \log p_t(u)\Sigma_\varepsilon^2 \mathsf{s}_t(u) = \nabla(\mathsf{s}_t(u)^\top \Sigma_\varepsilon^2 \mathsf{s}_t(u))$ and $\nabla\mathrm{div}(\Sigma_\varepsilon^2 \mathsf{s}_t(u)) = \nabla\mathrm{Tr}(\Sigma_\varepsilon^2 \nabla^2 \mathsf{s}_t(u))$. Indeed, for $k \in \{1,...,2d\}$, with $g(u) = \nabla \log p_t(u)$, and therefore $g_i(u) = \frac{\partial}{\partial u_i}g(u)$

$$
\begin{aligned}
\frac{\partial}{\partial u_k}\left(\nabla g(u)^\top \Sigma_\varepsilon^2 \nabla g(u)\right) &= \frac{\partial}{\partial u_k}\sum_{i,j} g_i(u)\Sigma_{\varepsilon,ij}^2 g_j(u) \\
&= \sum_{i,j}\Sigma_{\varepsilon,ij}^2\left(g_j(u)\frac{\partial}{\partial u_k}g_i(u) + g_i(u)\frac{\partial}{\partial u_k}g_j(u)\right) \\
&= 2\sum_{i=1}^{2d}\Sigma_{\varepsilon,ii}^2\left(g_i(u)\frac{\partial}{\partial u_k}g_i(u)\right) \\
&= 2\sum_{i=1}^{2d}\Sigma_{\varepsilon,ii}^2\left(\frac{\partial}{\partial u_i}g(u)\frac{\partial}{\partial u_k}\frac{\partial}{\partial u_i}g(u)\right) \\
&= \left[2\nabla^2 g(u)\Sigma_\varepsilon^2 \nabla g(u)\right]_k.
\end{aligned}
$$

$\square$

**Lemma D.7.** *Assume that* $(\overleftarrow{\mathbf{U}}_t)_{t\in[0,T]}$ *is solution to the backward SDE associated with* (9). *Then,*

$$
\mathrm{d}(\tilde{\mathsf{s}}_{T-t}(\overleftarrow{\mathbf{U}}_t)) = -\tilde{A}_\epsilon^\top \tilde{\mathsf{s}}_{T-t}(\overleftarrow{\mathbf{U}}_t)\mathrm{d}t + \nabla^2 \log \tilde{p}_{T-t}(\overleftarrow{\mathbf{U}}_t)\Sigma_\epsilon \mathrm{d}B_t. \tag{58}
$$

*Proof.* Recall that $p_\infty$ is the stationary distribution of (4) so that using Fokker-Planck equation we get, for $u \in \mathbb{R}^{2d}$,

$$
\begin{aligned}
0 = &-\mathrm{Tr}(A) - (Au)^\top \nabla \log p_\infty(u) \\
&+ \frac{1}{2}\left[\mathrm{div}\left(\Sigma^2 \nabla \log p_\infty(u)\right) + \nabla \log p_\infty(u)^\top \Sigma^2 \nabla \log p_\infty(u)\right].
\end{aligned}
$$

Using that $\tilde{p}_t = p_t/p_\infty$, and Fokker-Planck as in Lemma D.6

$$
\begin{aligned}
\partial_t \log \tilde{p}_t(u) = &-(Au)^\top \tilde{\mathsf{s}}_t(u) \\
&+ \frac{1}{2}\left[\mathrm{div}\left(\Sigma^2 \tilde{\mathsf{s}}_t(u)\right) + \tilde{\mathsf{s}}_t(u)^\top \Sigma^2 \tilde{\mathsf{s}}_t(u)\right] \\
&+ \tilde{\mathsf{s}}_t(u)^\top \Sigma^2 \nabla \log p_\infty(u).
\end{aligned}
$$

Using the definition of $\tilde{A}_\epsilon$, we have,

$$
\partial_t \log \tilde{p}_t(u) = (\tilde{A}_\epsilon u)^\top \tilde{\mathsf{s}}_t(u) + \frac{1}{2}\left[\mathrm{div}\left(\Sigma^2 \tilde{\mathsf{s}}_t(u)\right) + \tilde{\mathsf{s}}_t(u)^\top \Sigma^2 \tilde{\mathsf{s}}_t(u)\right],
$$

and therefore,

$$
\partial_t \log \tilde{p}_{T-t}(u) = -(\tilde{A}_\epsilon u)^\top \tilde{\mathsf{s}}_{T-t}(u) - \frac{1}{2}\left[\mathrm{div}\left(\Sigma^2 \tilde{\mathsf{s}}_{T-t}(u)\right) + \tilde{\mathsf{s}}_{T-t}(u)^\top \Sigma^2 \tilde{\mathsf{s}}_{T-t}(u)\right].
$$

Recall that the modified backward process can be written as

$$
\mathrm{d}\overleftarrow{\mathbf{U}}_t = (\tilde{A}_\epsilon \overleftarrow{\mathbf{U}}_t + \Sigma^2 \tilde{\mathsf{s}}_{T-t}(\overleftarrow{\mathbf{U}}_t))\mathrm{d}t + \Sigma_\epsilon \mathrm{d}B_t.
$$

Hence, by Itô's formula,

$$
\begin{aligned}
&\mathrm{d}(\tilde{\mathsf{s}}_{T-t}(\overleftarrow{\mathbf{U}}_t)) \\
&= \partial_t \tilde{\mathsf{s}}_{T-t}(\overleftarrow{\mathbf{U}}_t)\mathrm{d}t + \nabla^2 \log \tilde{p}_{T-t}(\overleftarrow{\mathbf{U}}_t)\mathrm{d}\overleftarrow{\mathbf{U}}_t + \frac{1}{2}\mathrm{Tr}\left(\Sigma^2 \nabla^2 \tilde{\mathsf{s}}_{T-t}(\overleftarrow{\mathbf{U}}_t)\right)\mathrm{d}t \\
&= \nabla\left(\partial_t \log \tilde{p}_{T-t}(\overleftarrow{\mathbf{U}}_t)\mathrm{d}t + \frac{1}{2}\tilde{\mathsf{s}}_{T-t}(\overleftarrow{\mathbf{U}}_t)^\top \Sigma^2 \tilde{\mathsf{s}}_{T-t}(\overleftarrow{\mathbf{U}}_t) + \frac{1}{2}\mathrm{div}\left(\Sigma^2 \tilde{\mathsf{s}}_{T-t}(\overleftarrow{\mathbf{U}}_t)\right)\right) \\
&\quad + \nabla^2 \log \tilde{p}_{T-t}(\overleftarrow{\mathbf{U}}_t)\tilde{A}_\epsilon \overleftarrow{\mathbf{U}}_t \mathrm{d}t + \nabla^2 \log \tilde{p}_{T-t}(\overleftarrow{\mathbf{U}}_t)\Sigma_\epsilon \mathrm{d}B_t \\
&= -\tilde{A}_\epsilon^\top \tilde{\mathsf{s}}_{T-t}(\overleftarrow{\mathbf{U}}_t) + \nabla^2 \log \tilde{p}_{T-t}(\overleftarrow{\mathbf{U}}_t)\Sigma_\epsilon \mathrm{d}B_t,
\end{aligned}
$$

which completes the proof.

$\square$

**Lemma D.8.** *Let $\Delta$ be an arbitrary fixed positive constant, and assume that $(\overleftarrow{\mathbf{U}}_t)_{t \in [0, T-\Delta]}$ is the solution to (11). Then, there exists a universal constant $C > 0$ such that*

$$\mathbb{E}\left[\left\|\nabla \log \tilde{p}_{T-t}\left(\overleftarrow{\mathbf{U}}_t\right) - \nabla \log \tilde{p}_{T-t_k}\left(\overleftarrow{\mathbf{U}}_{t_k}\right)\right\|^2\right] \le C\left(g(t_{k+1}) - g(t_k)\right),$$

*for $t \in [t_k, t_{k+1}]$, with*

$$g(t) := \mathbb{E}\left[\left\|\nabla \log \tilde{p}_{T-t}\left(\overleftarrow{\mathbf{U}}_t\right)\right\|^2\right]. \tag{59}$$

*Proof.* The argument follows from an adaptation of Conforti et al. (Proposition 3.2, 2025) to our setting. Let $Y_t := \nabla \log \tilde{p}_{T-t}(\overleftarrow{\mathbf{U}}_t)$ and $Z_t := \nabla^2 \log \tilde{p}_{T-t}(\overleftarrow{\mathbf{U}}_t)$. From (58), the process $(Y_t)_{t \in [0,T]}$ satisfies

$$\mathrm{d}Y_t = -\tilde{A}_\epsilon^\top Y_t \mathrm{d}t + Z_t \Sigma_\epsilon \mathrm{d}B_t.$$

Applying Itô's formula to $\|Y_t\|^2$ yields

$$\mathrm{d}\|Y_t\|^2 = -2\langle Y_t, \tilde{A}_\epsilon^\top Y_t\rangle \, \mathrm{d}t + 2\langle Y_t, Z_t \Sigma_\epsilon \, \mathrm{d}B_t\rangle + \|Z_t \Sigma_\epsilon\|_{\mathrm{Fr}}^2 \, \mathrm{d}t.$$

Therefore, there exists a constant $c > 0$, depending only on $a$, such that

$$\mathrm{d}\|Y_t\|^2 \ge c\left(\|Y_t\|^2 + \|Z_t \Sigma_\epsilon\|_{\mathrm{Fr}}^2\right) \mathrm{d}t + \tilde{H}_t \mathrm{d}B_t,$$

where $\tilde{H}_t$ denotes a stochastic process. Moreover, following the argument of Conforti et al. (Lemma 3.3, 2025), the stochastic integral $\int_0^t \tilde{H}_r \mathrm{d}B_r$ is a true martingale. Using this and integrating over $[t_k, t]$, we deduce that there exists a universal constant $C > 0$ (whose value may change in the course of the argument) such that

$$\mathbb{E}\left[\|Y_t - Y_{t_k}\|^2\right] \le C \int_{t_k}^{t_{k+1}} \mathbb{E}\left[\|Y_s\|^2 + \|Z_s \Sigma_\epsilon\|_{\mathrm{Fr}}^2\right] \mathrm{d}s \le C(g(t_{k+1}) - g(t_k)).$$

$\square$

**Lemma D.9.** *Let $A \in \mathbb{R}^{n \times n}$ be an invertible matrix, and let $B \in \mathbb{R}^{n \times n}$ be such that $A - B$ is also invertible. Then,*

$$(A - B)^{-1} - A^{-1} = (A - B)^{-1} B A^{-1}.$$

*Proof.* Note that

$$(A - B)^{-1} - A^{-1} = (A - B)^{-1} A A^{-1} - A^{-1} = \left[(A - B)^{-1} A - \mathbf{I}_n\right] A^{-1},$$

and

$$(A - B)^{-1} A = (A - B)^{-1}\left((A - B) + B\right) = \mathbf{I}_n + (A - B)^{-1} B,$$

so that

$$\left[(A - B)^{-1} A - \mathbf{I}_n\right] A^{-1} = (A - B)^{-1} B A^{-1},$$

which completes the proof. $\square$

# E  Numerical Illustration

This section provides additional details on the numerical implementation described in Section 4.

## E.1  CLD training and sampling

Algorithms 1 and 2 show the training and sampling procedures for the CLD-based approaches, respectively.

---

**Algorithm 1** CLD Training

---

**Require:** Dataset $\mathcal{D}$, batch size $B$, network $s_\theta(\cdot, t)$, a positive weight function $\lambda : [0, T] \to \mathbb{R}_+$ and $\epsilon \geq 0$.
1: Precompute $\tilde{\Sigma}_{0,t} = \Sigma_{0,t} + e^{tA}\mathrm{diag}(0\mathbf{I}_d, v^2\mathbf{I}_d)(e^{tA})^\top$.   ▷ The value of $\Sigma_{0,t}$ depends on $\epsilon$, see Lemma A.2. (eq 23).
2: **while** not converged **do**
3:     Sample $\{x^{(i)}\}_{i=1}^B \sim \mathcal{D}$
4:     Sample $\{t^{(i)}\}_{i=1}^B \sim \mathcal{U}([0, T])$
5:     Sample $\{\varepsilon^{(i)}\}_{i=1}^B \sim \mathcal{N}(0, \mathbf{I}_{2d})$
6:     $\overrightarrow{\mathbf{U}}_{t^{(i)}} = e^{t^{(i)}A}\left(\overrightarrow{X}_0, 0_d\right)^\top + (\tilde{\Sigma}_{0,t^{(i)}})^{1/2}\varepsilon^{(i)}$
7:     $\mathcal{L} \leftarrow \frac{1}{B}\sum_{i=1}^B \lambda(t^{(i)})\left\|s_\theta\left(t^{(i)}, \overrightarrow{\mathbf{U}}_{t^{(i)}}\right) + (\tilde{\Sigma}_{0,t^{(i)}})^{-1/2}\varepsilon^{(i)}\right\|^2$
8:     Update $\theta$ by taking gradient step on $\nabla_\theta \mathcal{L}$
9: **end while**

---

**Algorithm 2** CLD Sampling

---

**Require:** Learned network $s_\theta$, number of discretization steps $N$ and $\epsilon \geq 0$.
1: $h \leftarrow T/N$
2: $\bar{\mathbf{U}}_0 \sim \pi_\infty$
3: **for** $k = 0$ down to $N - 1$ **do**
4:     $t_k \leftarrow k\,h$
5:     Sample $Z_k \sim \mathcal{N}(0, \mathbf{I}_{2d})$                ▷ $\pi_\infty$ depends on $\epsilon$, see (21) in Lemma A.2.
6:     $\bar{\mathbf{U}}_{t_{k+1}}^\theta = \bar{\mathbf{U}}_{t_k}^\theta + h\left(\tilde{A}_\epsilon \bar{\mathbf{U}}_{t_k}^\theta + \Sigma_\epsilon^2 s_\theta(t_k, \bar{\mathbf{U}}_{t_k}^\theta)\right) + \sqrt{h}\Sigma_\epsilon Z_k$
7: **end for**
8: **return** First $d$ coordinates of $\bar{\mathbf{U}}_{t_N}^\theta$                ▷ Return position only, discard velocity.

---

## E.2  Time-rescaling of the forward SDE

Following Dockhorn et al. (2022), one often implements in practice a time-rescaled version of

$$d\overrightarrow{\mathbf{U}}_t = A\overrightarrow{\mathbf{U}}_t dt + \Sigma_\epsilon dB_t\,,$$

by introducing a positive noise schedule $\beta\colon [0, 1] \to [0, \infty)$ and setting

$$\tilde{\overrightarrow{\mathbf{U}}}_t = \overrightarrow{\mathbf{U}}_{\tau(t)} \quad \text{and} \quad \tau(t) = \int_0^t \beta(s)\mathrm{d}s\,.$$

Equivalently, $\tilde{\overrightarrow{\mathbf{U}}}_t$ satisfies the inhomogeneous SDE

$$d\tilde{\overrightarrow{\mathbf{U}}}_t = \underbrace{\beta(t)A}_{=\tilde{A}(t)}\tilde{\overrightarrow{\mathbf{U}}}_t dt + \underbrace{\sqrt{\beta(t)}\,\Sigma_\epsilon}_{=\tilde{\Sigma}_\epsilon(t)} dB_t\,,$$

In the critically-damped example (Equation (4)), we have

$$\tilde{A}(t) = \begin{pmatrix} 0 & \beta(t)\,a^2 \\ -\beta(t) & -2a\,\beta(t) \end{pmatrix} \otimes \mathbf{I}_d, \quad \tilde{\Sigma}_\epsilon(t) = \sqrt{\beta(t)}\Sigma_\epsilon \otimes \mathbf{I}_d\,.$$

**Mean factor.** Since $\overset{\sim}{\overrightarrow{\mathbf{U}}}_t = \overrightarrow{\mathbf{U}}_{\tau(t)}$, we can deduce from the homogeneous solution the mean factor,

$$\mathbb{E}\left[\overset{\sim}{\overrightarrow{\mathbf{U}}}_t \mid \overrightarrow{\mathbf{U}}_0\right] = e^{-a\,\tau(t)}\left(\begin{pmatrix} 1 + a\,\tau(t) & a^2\,\tau(t) \\ -\tau(t) & 1 - a\,\tau(t) \end{pmatrix} \otimes \mathbf{I}_d\right)\overrightarrow{\mathbf{U}}_0\,.$$

**Covariance.** Again by the time-change $\tau(t)$, one has

$$\mathrm{Cov}\big(\overset{\sim}{\overrightarrow{\mathbf{U}}}_t \mid \overrightarrow{\mathbf{U}}_0\big) = \mathrm{Cov}\big(\overrightarrow{\mathbf{U}}_{\tau(t)} \mid \overrightarrow{\mathbf{U}}_0\big) = \int_0^{\tau(t)} e^{sA}\Sigma_\epsilon\,\Sigma_\epsilon^T e^{sA^T}\,\mathrm{d}s\,.$$

**Affine schedule.** A popular and simple choice of noise schedule is an affine noise schedule given by

$$\beta(t) = \beta_1 t + \beta_0\,, \quad \tau(t) = \frac{\beta_1}{2}t^2 + \beta_0 t\,.$$

### E.3 Score approximation

**Denoising Score Matching (DSM).** Recall that the conditional score function of the forward process (4) given the initial data distribution is Gaussian,

$$\nabla \log p_t(\overrightarrow{\mathbf{U}}_t | \overrightarrow{\mathbf{U}}_0) = -\Sigma_{0,t}^{-1}\left(\overrightarrow{\mathbf{U}}_t - e^{tA}\overrightarrow{\mathbf{U}}_0\right)\,.$$

Hence, following Vincent (2011) the conditional denoising score matching loss $\mathcal{L}_{\mathrm{cond}}$, for $\theta \in \Theta$, $s_\theta(t,x) : [0,T] \times \mathbb{R}^{2d} \mapsto \mathbb{R}^{2d}$ and $Z_{2d} \sim \mathcal{N}(0, \mathbf{I}_{2d})$ can be written as

$$\mathcal{L}_{\mathrm{DSM}}(\theta) = \mathbb{E}\left[\lambda(t)\left\|s_\theta\left(\tau, \overrightarrow{\mathbf{U}}_\tau\right) - \nabla \log p_\tau\left(\overrightarrow{\mathbf{U}}_\tau | \overrightarrow{\mathbf{U}}_0\right)\right\|^2\right]$$

$$= \mathbb{E}\left[\lambda(t)\left\|s_\theta\left(\tau, e^{\tau A}\overrightarrow{\mathbf{U}}_0 + \sqrt{\Sigma_{0,\tau}}Z_{2d}\right) + \Sigma_{0,t}^{-1/2}Z_{2d}\right\|^2\right]\,,$$

where $\tau \sim \mathcal{U}[0,T]$, $\tau \perp Z_{2d}$ and $\lambda : [0,T] \mapsto \mathbb{R}_{>0}$.

**Hybrid Score Matching (HSM).** It has been shown in Dockhorn et al. (2022) that another loss, potentially more stable numerically can be obtained by conditioning only on $\overrightarrow{X}_0$ rather than on the full state $\overrightarrow{\mathbf{U}}_0 = (\overrightarrow{X}_0, \overrightarrow{V}_0)^\top$. This hybrid score matching loss can be derived by marginalizing out the velocity component $\overrightarrow{V}_0 \sim \mathcal{N}(0_d, v^2\mathbf{I}_d)$, $\overrightarrow{V}_0 \perp \overrightarrow{X}_0$ in the conditional score function,

$$\mathcal{L}_{\mathrm{HSM}}(\theta) = \mathbb{E}\left[\lambda(t)\left\|s_\theta(\tau, \overrightarrow{\mathbf{U}}_\tau) - \nabla \log p_\tau(\overrightarrow{\mathbf{U}}_\tau \mid \overrightarrow{X}_0)\right\|^2\right]$$

$$= \mathbb{E}\left[\lambda(t)\left\|s_\theta\left(\tau, e^{\tau A}\begin{pmatrix}\overrightarrow{X}_0 \\ 0_d\end{pmatrix} + \sqrt{\Sigma'_{0,\tau}}Z_{2d}\right) + (\Sigma'_{0,\tau})^{-1/2}Z_{2d}\right\|^2\right]\,,$$

with $Z_{2d} \sim \mathcal{N}(0, \mathbf{I}_{2d})$ independent of $\tau \sim \mathcal{U}[0,T]$ and

$$\Sigma'_{0,\tau} = \Sigma_{0,\tau} + e^{\tau A}\begin{pmatrix} 0 & 0 \\ 0 & v^2\mathbf{I}_d \end{pmatrix}(e^{\tau A})^\top\,.$$

### E.4 Neural network architectures

In Figure 3 we detail the neural network used in the illustration. The input layer is composed of a vector $x$ in dimension $2d$ and the time $t$. Both are respectively embedded using a linear transformation or a sine/cosine transformation (Nichol and Dhariwal, 2021) of width `mid_features`. Then, 3 dense layers of constant width `mid_features` followed by SiLu activations and skip connections regarding the time embedding. The output layer is linear resulting in a vector of dimension $d$ (when $\varepsilon = 0$) and $2d$ (when $\varepsilon \neq 0$).

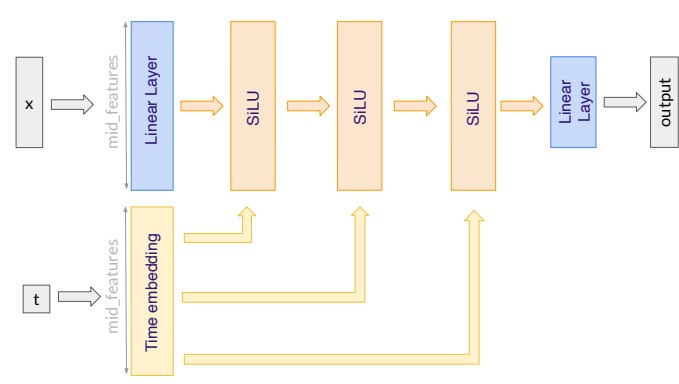

Figure 3: Neural network architecture.

## E.5   Additional experiments

We present additional experimental results for the MG25 distribution in dimension 100 and the 2D-diamond dataset. The MG25 distribution is defined as a Gaussian mixture model with 25 modes in dimension 100, defined as

$$\pi_{\mathrm{data}}(x) = \frac{1}{25} \sum_{(j,k) \in \{-2,...,2\}^2} \varphi_{\mu_{jk}, \Sigma_d}(x)$$

with $\varphi_{\mu_{jk}, \Sigma_d}$ denoting the probability density function of the Gaussian distribution with covariance matrix $\Sigma_d = \mathrm{diag}\,(0.01, 0.01, 0.1, ..., 0.1)$ and mean vector $\mu_{jk} = [j, k, 0, 0, 0..., 0]^\top$. This dataset has been previously used in Thin et al. (2021); Strasman et al. (2025). The 2D-diamond distribution is a two-dimensional dataset with well-separated modes, used as a synthetic dataset in Dockhorn et al. (2022).

Tables 1, 2 and 3 report the sliced-Wasserstein error for different values of the regularization parameter $\varepsilon \in \{0, 0.1, 0.25, 0.5, 1\}$ and drift coefficient $a \in \{0.1, 0.25, 0.5, 1, 2\}$, using the same experimental setup as for the Funnel dataset described in Section 4. Both tables 1 and 2 highlight the improvement in generation quality achieved with smaller regularization values of $\varepsilon$. Table 3 report the values displayed in Figure 1 with the associated standard deviations.

Table 1: Comparison of mean Wasserstein distance for different noise levels $\varepsilon$ on the MG25-100D (mean $\pm$ standard deviation across 5 runs; lower is better).

| $\varepsilon$ | $a = 0.1$ | $a = 0.25$ | $a = 0.5$ | $a = 1.0$ | $a = 2.0$ |
|---|---|---|---|---|---|
| 0 | $0.284 \pm 0.002$ | $0.199 \pm 0.001$ | $0.034 \pm 0.002$ | $0.009 \pm 0.001$ | $0.009 \pm 0.001$ |
| 0.1 | $0.192 \pm 0.001$ | $0.159 \pm 0.001$ | $0.026 \pm 0.001$ | $\mathbf{0.005 \pm 0.001}$ | $0.008 \pm 0.001$ |
| 0.25 | $\mathbf{0.013 \pm 0.001}$ | $0.065 \pm 0.001$ | $0.015 \pm 0.001$ | $0.007 \pm 0.001$ | $\mathbf{0.007 \pm 0.001}$ |
| 0.5 | $0.191 \pm 0.007$ | $\mathbf{0.004 \pm 0.001}$ | $\mathbf{0.009 \pm 0.001}$ | $0.008 \pm 0.001$ | $0.008 \pm 0.001$ |
| 1 | $0.389 \pm 0.030$ | $0.045 \pm 0.003$ | $0.011 \pm 0.002$ | $0.006 \pm 0.001$ | $0.008 \pm 0.001$ |

Table 2: Comparison of mean Wasserstein distance for different noise levels $\varepsilon$ on the Diamond-2D (mean $\pm$ standard deviation across 5 runs; lower is better).

| $\varepsilon$ | $a = 0.1$ | $a = 0.25$ | $a = 0.5$ | $a = 1.0$ | $a = 2.0$ |
|---|---|---|---|---|---|
| 0 | $0.322 \pm 0.001$ | $0.256 \pm 0.004$ | $0.039 \pm 0.002$ | $0.007 \pm 0.001$ | $0.007 \pm 0.002$ |
| 0.1 | $0.234 \pm 0.001$ | $0.198 \pm 0.003$ | $0.026 \pm 0.004$ | $0.004 \pm 0.001$ | $0.005 \pm 0.001$ |
| 0.25 | $\mathbf{0.048 \pm 0.001}$ | $0.074 \pm 0.003$ | $0.021 \pm 0.002$ | $\mathbf{0.004 \pm 0.001}$ | $\mathbf{0.005 \pm 0.001}$ |
| 0.5 | $0.073 \pm 0.002$ | $\mathbf{0.008 \pm 0.001}$ | $\mathbf{0.008 \pm 0.002}$ | $0.006 \pm 0.002$ | $0.006 \pm 0.002$ |
| 1 | $0.095 \pm 0.002$ | $0.029 \pm 0.002$ | $0.014 \pm 0.001$ | $0.013 \pm 0.001$ | $0.011 \pm 0.001$ |

Table 3: Comparison of mean Wasserstein distance for different noise levels $\varepsilon$ on the Funnel-100D (mean $\pm$ standard deviation across 5 runs; lower is better).

| $\varepsilon$ | $a = 0.1$ | $a = 0.25$ | $a = 0.5$ | $a = 1.0$ | $a = 2.0$ |
|---|---|---|---|---|---|
| 0 | $0.991 \pm 0.001$ | $0.73 \pm 0.002$ | $0.291 \pm 0.005$ | $0.225 \pm 0.056$ | $0.223 \pm 0.011$ |
| 0.1 | $0.705 \pm 0.001$ | $0.632 \pm 0.002$ | $0.278 \pm 0.001$ | $0.158 \pm 0.027$ | $0.198 \pm 0.004$ |
| 0.25 | $\mathbf{0.277 \pm 0.002}$ | $0.409 \pm 0.003$ | $0.248 \pm 0.012$ | $\mathbf{0.137 \pm 0.005}$ | $\mathbf{0.179 \pm 0.006}$ |
| 0.5 | $1.171 \pm 0.015$ | $\mathbf{0.248 \pm 0.002}$ | $0.228 \pm 0.005$ | $0.157 \pm 0.002$ | $0.203 \pm 0.003$ |
| 1 | $2.885 \pm 0.016$ | $0.785 \pm 0.011$ | $\mathbf{0.191 \pm 0.008}$ | $0.253 \pm 0.006$ | $0.233 \pm 0.002$ |

