# OpenReview forum: "Wasserstein Convergence of Critically Damped Langevin Diffusions"
_NeurIPS.cc/2025/Conference — NeurIPS 2025 poster_

### Official Review · Reviewer_N5Lx · 2025-06-29

**Clarity:** 3
**Significance:** 3
**Originality:** 3
**Rating:** 5
**Confidence:** 2

**Summary:**

This paper studies Critically Damped Langevin Diffusions (CLDs), a second-order extension of score-based generative models, and provides the first W2 convergence guarantees under weak assumptions.

The authors first prove that vanilla CLD samplers converge in W2 with an error bound decomposed into mixing, discretization ($O(\sqrt{h})$), and score-approximation terms.

The authors then further introduce a new hyperparameter $\epsilon$ controlling noise on the position coordinate and prove an improved $O(h)$ discretization rate under stronger log-concavity assumptions, and give practical tuning guidance.

Experiments on the 50-dimensional Funnel distribution validate that small $\epsilon$ yields lower W2 error and variance than no $\epsilon$.

**Questions:**

Can the additional noise hyperparameter $\epsilon$ in Section 4 be interpreted as a form of warm-up or preconditioning?

**Ethical Concerns:**

["NO or VERY MINOR ethics concerns only"]

**Final Justification:**

The authors addressed all my concerns appropriately. I would like to maintain my current positive scores.

**Limitations:**

There are no societal impact considerations.

**Paper Formatting Concerns:**

The font size in Figure 1 is small.

Pseudo-code for the proposed algorithm should be provided in the main text.

**Quality:**

3

**Strengths And Weaknesses:**

**Strengths:**

1. The theoretical results are nontrivial and fill a gap in the convergence theory of CLD-based samplers. *The reviewer acknowledges here unfamiliarity with the latest advances in Langevin diffusion analysis and did not check all derivations.*
2. This is the first paper to derive W2 convergence rates for CLDs (previous work focused on KL divergence). The regularized variant in Section 4 is a novel extension with provable benefits.
3. The main text is generally well organized.

**Weaknesses:**
1. There is no analysis or experiment on warm-up (burn-in) strategies or initialization sensitivity.
2. It would be great to include an explicit pseudocode block in the main text to summarize the algorithm for analysis.
3. Experimental validation is limited to synthetic tasks. As this work mainly focuses on theoretical advances, the reviewer thinks it is acceptable.

---

> ### Author Rebuttal · Authors · 2025-07-31
>
> We thank the reviewer for the positive assessment and for recognizing the novelty of our $\mathcal{W}_2$ analysis for CLD. Below, we address all concerns and describe the additions in the revision.
>
> **Explicit pseudocode**
>
> For clarity, we now include a concise pseudocode block for both training (Algorithm 1) and sampling (Algorithm 2). They are now integrated into the manuscript. We are happy to provide a more detailed version if the reviewers find it necessary.
>
> ---
>
> ### Algorithm: CLD Training
>
> **Inputs:** Dataset $\mathcal{D}$, batch size $B$, network $s_\theta(\cdot, t)$, a positive weight function $\lambda : [0,T] \to \mathbb{R}_+$, and noise level $\epsilon \geq 0$.
>
> **Precomputation:**  Compute  $\tilde \Sigma_{0,t} = \Sigma_{0,t} + e^{t A} \, \mathrm{diag}(0 \cdot I_d, v^2 I\_d) \, (e^{t A})^\top$  (Note: The value of $\Sigma_{0,t}$ depends on $\epsilon$, see Lemma A.2 (Eq. 23).)
>
> **While not converged:**
> 1. Sample {$x^{(i)}$}$_{i=1}^{B} \sim \mathcal{D}$
> 2. Sample {$t^{(i)}$}$_{i=1}^{B} \sim \mathcal{U}([0,T])$
> 3. Sample {$\varepsilon^{(i)}$}$\_{i=1}^{B} \sim \mathcal{N}(0, I_{2d})$
> 4. Compute
>    $$
> \overrightarrow{\mathbf{U}}\_{t^{(i)}} = e^{t^{(i)} A}
> ( \overrightarrow{X}\_0 \\ 0\_d )^\top + ( \tilde{\Sigma}\_{0,t^{(i)}} )^{1/2} \varepsilon^{(i)}
> $$
> 5. Compute loss
>    $$
>    \mathcal{L} \gets \frac{1}{B} \sum_{i=1}^{B} \lambda(t^{(i)}) || s\_\theta(t^{(i)}, \overrightarrow{\mathbf{U}}\_{t^{(i)}}) + (\tilde \Sigma\_{0,t^{(i)}})^{-1/2} \varepsilon^{(i)}||^2
>    $$
> 6. Update $\theta$ via gradient descent on $\nabla_\theta \mathcal{L}$
>
> ---
>
> ### Algorithm: CLD Sampling
>
> **Inputs:** Learned network $s_\theta$, number of discretization steps $N$, and noise level $\epsilon \geq 0$.
>
> 1. Set step size: $h \leftarrow T/N$
> 2. Sample initial state: $\mathbf{\bar{U}}_0 \sim \pi_\infty$
> 3. **For** $k = 0$ to $N-1$:
>    - Compute current time: $t_k \leftarrow k h$
>    - Sample noise: $Z_k \sim \mathcal{N}(0, I_{2d})$  *(Note: $\pi_\infty$ depends on $\epsilon$, see Lemma A.2, Equation (21).)*
>    - Update state:
>      $$
>      \bar{\mathbf{U}}^\theta_{t_{k+1}} = \bar{\mathbf{U}}^\theta_{t_k} + h \left( \tilde{A} \bar{\mathbf{U}}^\theta_{t_k} + \Sigma_\epsilon^2 s_\theta(t_k, \bar{\mathbf{U}}^\theta_{t_k}) \right) + \sqrt{h} \Sigma_\epsilon Z_k
>      $$
> 4. **Return** the first $d$ coordinates of $\bar{\mathbf{U}}^\theta_{t_N}$  *(i.e., return position only, discard velocity)*
>
> ---
>
> **Experiments beyond the original synthetic tasks**
>
> In addition to our initial experimental results, we have extended the evaluation on the Funnel distribution in dimension 100 by varying the parameters $a$ and $\epsilon$. We analyze the impact of these key parameters and report the corresponding results in Table 1. The revised paper also includes an updated figure. Our regularized method consistently yields improved sliced Wasserstein-2 distances, with the performance gap increasing notably as $a$ decreases.
>
> Moreover, we have added results on two additional challenging synthetic datasets: MG-25 in dimension 100 (a 25-mode Gaussian mixture) and Diamonds (a 2D Gaussian mixture arranged in a diamond-shaped configuration). These additional experiments (Tables 2 and 3) further support our claims and exhibit trends consistent with our earlier findings. Due to time constraints, the computation of standard deviations is still in progress; we will include them in the final version once available. Moreover, note that in our ablation study, we set $\sigma$ to $\sqrt{2}$ so that the case where $a=2$ and $\epsilon=0$ exactly matches the setting recommended by Dockhorn et al. (2022). We also adopted the same time renormalization as them to make the comparison as fair as possible (see Appendix C.1).
>
> **Table 1: Comparison of mean Wasserstein distance for different noise levels $\varepsilon$ on the Funnel‑100D**
> *(Mean ± standard deviation across 5 runs; lower is better)*
>
> | $\varepsilon$ |        | $a=0.25$             | $a=0.5$              | $a=1.0$                | $a=2.0$                |
> |:-------------:|--------|:--------------------:|:--------------------:|:----------------------:|:----------------------:|
> | 0             |        | 0.73 ± 0.0016        | 0.291 ± 0.0047       | 0.225 ± 0.056          | 0.223 ± 0.011          |
> | 0.1           |        | 0.632 ± 0.0016       | 0.278 ± 0.0012       | 0.158 ± 0.027          | 0.198 ± 0.004          |
> | 0.25          |        | 0.409 ± 0.0031       | 0.248 ± 0.012        | **0.137 ± 0.0046**     | **0.179 ± 0.0057**     |
> | 0.5           |        | **0.248 ± 0.0016**   | 0.228 ± 0.0051       | 0.157 ± 0.0022         | 0.203 ± 0.0028         |
> | 1             |        | 0.785 ± 0.011        | **0.191 ± 0.0079**   | 0.253 ± 0.006          | 0.233 ± 0.002          |
>
>
> **Table 2: Comparison of mean Wasserstein distance for different noise levels $\varepsilon$ on the MG25‑100D**
>
> | $\varepsilon$ |        | $a=0.1$        | $a=0.25$       | $a=0.5$         | $a=1.0$         | $a=2.0$         |
> |:-------------:|:------:|:--------------:|:--------------:|:---------------:|:---------------:|:---------------:|
> | 0             |        | 0.285          | 0.199          | 0.0346          | 0.0083          | 0.009           |
> | 0.1           |        | 0.191          | 0.158          | 0.0262          | **0.0063**      | 0.008           |
> | 0.25          |        | **0.013**      | 0.0654         | 0.0157          | 0.0076          | **0.007**       |
> | 0.5           |        | 0.188          | **0.005**      | **0.0093**      | 0.0071          | 0.0081          |
> | 1             |        | 0.381          | 0.0491         | 0.0121          | 0.0064          | **0.007**       |
>
>
> **Table 3: Comparison of mean Wasserstein distance for different noise levels $\varepsilon$ on the Diamond‑2D**
>
> | $\varepsilon$ |        | $a=0.1$        | $a=0.25$       | $a=0.5$         | $a=1.0$         | $a=2.0$         |
> |:-------------:|:------:|:--------------:|:--------------:|:---------------:|:---------------:|:---------------:|
> | 0             |        | 0.323          | 0.252          | 0.0325          | 0.0058          | 0.0077          |
> | 0.1           |        | 0.233          | 0.198          | 0.0237          | **0.0038**      | 0.0047          |
> | 0.25          |        | **0.0474**     | 0.0737         | 0.0214          | 0.0049          | **0.004**       |
> | 0.5           |        | 0.0722         | **0.0078**     | **0.0077**      | 0.0057          | 0.0056          |
> | 1             |        | 0.0958         | 0.0286         | 0.0143          | 0.0136          | 0.012           |
>
>
> **Initialization sensitivity**
>
> We are not completely certain we have interpreted the question regarding the initialization sensitivity correctly. Our error decomposition makes the effect of initialization of the forward process explicit. In particular,the mixing term error, the one that decays as $e^{-c_2 T}$ is proportional to the Wasserstein distance between the initial law of the forward process ($\pi_{\mathrm{data}} \otimes \pi_v$) and the stationary distribution $\pi_{\infty}$. Therefore, choosing any initial velocity variance $v$ that decreases $\mathcal{W}\_2 (\pi\_{\mathrm{data}} \otimes \pi\_v, \pi\_{\infty})$ reduces the constant. However, discussing the full impact of the initial velocity variance $v$ on the whole bound is more subtle. Indeed, $v$ affects the regularity of the forward distribution $p_t$ that governs the contraction properties. For example, in the Gaussian CLD setting (with $X_0 \sim \mathcal N(\mu_0,s_0^2)$, $V_0 \sim \mathcal N(0,v^2)$), the regularity constants can directly be derived from the covariance $\Sigma_{0,t}$ of $U_t=(X_t, V_t)$. A straightforward computation shows that a larger $v$ yields a lower log-concavity constant $c_t$ and a lower smoothness constant $L_t$ for $p_t$, yielding a clear trade‑off. We will add ablation studies for the sensitivity to the initial distribution in the revised version for different synthetic data distributions. Should the interpretation of the question be incorrect, we would be happy to provide additional details if necessary.
>
> Regarding the question of whether adding $\varepsilon$ can be interpreted as a warm-up strategy: this can be considered true, to some extent. The addition of $\varepsilon$ affects the initialization of the backward sampler by modifying the stationary distribution $\pi_\infty$ of the forward process. However, beyond initialization, it also acts as a noise injection mechanism that impacts the regularity of the backward process---namely the constants $c_t$ and $L_t$---as well as the admissible step size $h$.
>
> We thank the reviewer once again for their insightful comments. The revisions above, along with the additional experiments, directly address all points raised and further strengthen the manuscript. We hope these changes meet the expectations and would be happy to clarify any remaining questions.

---

### Official Review · Reviewer_N2VV · 2025-07-01

**Clarity:** 2
**Significance:** 2
**Originality:** 2
**Rating:** 4
**Confidence:** 3

**Summary:**

The paper studies Wasserstein convergence of score-based generative diffusion models where the forward process is governed by a critically damped Langevin diffusion. A modified dynamic is proposed that incorporates an additional hyperparameter that controls the noise applied to the data. Some numerical experiments are also provided.

**Questions:**

(1) I am still a bit confused with Theorem 3.1. and Theorem 4.1. In Theorem 3.1., you obtained Wasserstein convergence guarantees without H4, the strongly-log-concave assumption. But then in Theorem 4.1. you need H4. It would be helpful if you can explain a bit why for Theorem 4.1. you need H4, but not for Theorem 3.1.

(2) Right before Theorem 4.1., you mentioned your results match existing results for SGMs in the literature. But if so, what is the advantage using your proposed modified dynamics compared to SGMs?

**Ethical Concerns:**

["NO or VERY MINOR ethics concerns only"]

**Final Justification:**

I have raised my score. I'm satisfied with the author(s)' response.

**Limitations:**

It seems the author(s) did not discuss limitations in the paper.

**Paper Formatting Concerns:**

No.

**Quality:**

2

**Strengths And Weaknesses:**

Strengths:

(1) The paper is a theoretical paper. The analysis seems to be rigorous.

(2) The Wasserstein convergence guarantees for critically damped Langevin diffusions on the Euclidean space seem to be new.

Weaknesses:

(1) The theoretical convergence guarantees for critically damped Langevin diffusion models have already been studied in the literature, including Chen et al. (2023) that is cited in the paper. There, the authors claimed that critically damped Langevin diffusion models cannot improve upon the theoretical guarantees for using the classical score-based diffusion model based on overdamped Langevin diffusion. By reading this paper, it is not clear what is the advantage using critically damped Langevin diffusions and whether acceleration can be achieved.

(2) The paper mentions that Chen et al. (2023) and Conforti et al. (2025) analyzed the convergence of SGMs in the extended space using kinetic Langevin-dynamics. Their results use other measures, such as KL divergence in Conforti et al. (2025). KL sometimes can imply Wasserstein, for example on a compact domain, or when the target distribution is strongly-log-concave, which is assumed in H4 in the paper.  The paper should highlight why for the settings considered in this paper,  Wasserstein convergence cannot be obtained using existing results or at least using the existing approaches in the literature.

(3) I find the assumption H4 very strong, because it is not just strong-log-concavity, but at every $t$.

---

> ### Author Rebuttal · Authors · 2025-07-31
>
> We thank the reviewer for their valuable feedback and have addressed each point below. The proposed revisions have been incorporated into the revised paper.
>
> **On the benefit of deriving $\mathcal{W}\_2$-bound.**
>
> Previous empirical work (Dockhorn et al., 2022) shows that CLD can outperform standard SGMs forwards on image benchmarks yet the only theoretical results were given in KL (Chen et al. (2023) and Conforti et al.(2025)) and not in $\mathcal{W}\_{2}$ -distance
> even though the latter enjoys nice geometric properties and is a true mathematical distance. The goal of this work is to close this gap and place CLD on the same footing as VP/VE SDEs in Wasserstein distance, while maintaining state‑of‑the‑art, minimal, regularity assumptions on the data distribution.
> In this way, as training in the coupled phase space has been shown by Dockhorn et al. (2022) to be empirically more stable (in particular the Hybrid Score Matching loss), our results certify that the associated Euler-Maruyama sampler remains $\mathcal{W}\_{2}$‑stable even when the score is learned only approximately (Assumption H3).
>
> As also mentioned by the reviewer, our contribution concerns a different metric and setting as Chen et al. (2023) and Conforti et al.(2025). We give here the first $\mathcal{W}\_2$ guarantees for CLD, showing it attains the same convergence rates as SGMs. Note that the proof technique is completely different and rely on coupling techniques tailored to $\mathcal{W}\_2$ analysis. We also agree that KL divergence upper bounds can sometimes be translated to $\mathcal{W}\_2$ upper bounds; however, this is non-trivial and the price to pay may be high. In the case of compactly supported data distribution, the KL bounds constitute a valid upper bound to the $\mathcal{W}\_2$ metric. Nevertheless, this upper bound depends on the diameter of the space (Villani, 2008, Thm. 6.12, p. 82), which could therefore grow very large, especially in high dimensional settings, such as those encountered in image datasets. Regarding the log-concave case, we are not sure what was meant. If the reference was meant to Talagrand's inequality, this typically requires to verify regularity on the measure to the right of the KL (i.e. $\nu$ for $\mathrm{KL}(\mu||\nu)$) which in standard convergence results (Chen et al(2023), Conforti et al.(2025)) is usually the distribution generated by the algorithm and not the data distribution.
>
> Note that Gao et al. (2025) assume strong log-concavity of the data distribution (among other conditions) in $\mathbb{R}^d$, and explicitly state that their results are not implied by existing KL-based convergence bounds without further assumptions:
> "our results are not implied by the existing convergence results for SDE-based samplers." (Gao et al. (2025), p.5).
>
> On the practical side, as noted in Strasman et al. (2025), Wasserstein upper bounds are often more tractable to approximate empirically, whereas KL divergences and Fisher information are difficult to estimate in high dimensions. It is also worth noting that previous results focus on specific parametrization choices ($a=1$ and $\sigma = 2$ so that the drift and diffusion coefficients are coupled in a specific manner), which limits their generality and expressiveness.
>
> **Response to Q1.**
>
> Note that our main contribution is Theorem 3.1, which establishes a quantitative bound for generic Langevin dynamics under weak Lipschtiz-like assumptions. Moreover, we propose in Section 4 an upper bound in the specific and widespread strongly log-concave setting. This framework allows to target more precisely the role of the parameters $a$, $\epsilon$ and $\sigma$. We agree that the initial version was a bit confusing. We now highlight that Section 4 is a particular case of Theorem 3.1. In this way, we turn the title of Section 4 into: "Specific analysis of $\epsilon$-CLD under strong log-concavity".
>
>
> **On the log-concavity at all times.**
>
>  Regarding the question on Assumption H4, we agree that this is stronger than necessary. In fact, there is no need to require strong log‑concavity at all times. Similar to Gao et al. (2025) and Strasman (2025) (in the VPSDE case), it is enough to assume that the data probability density function is $c_0$-strongly log concave for the property to propagate at all times, i.e. for $\log p_t$ to be strongly concave for every $t \geq 0$. In fact, $c_t$ is explicitly available and given by,
>
> \begin{align*}
> c_t = \left( \frac{1}{ \min (c_0, v) \sigma^2_{\min}(e^{-tA})} + \lambda_{\max}(\Sigma_{0,t}) \right)^{-1} .
> \end{align*}
>
> In particular, this expression also highlights the impact of the calibration of the initial variance $v$ of the velocity component, with respect to the data log-concavity constant and the implicit impact of $\epsilon$ through $\Sigma_{0,t}$. In addition, note that Section 4 provides quantitative upper bounds in the widespread setting of strongly log-concave distributions but Theorem 3.1 does not require this assumption.
>
> **Clarification on the assumptions.**
>
> To take into account this remark on the discussion of the assumptions and to better position our work in the literature, we have added a paragraph discussing our assumptions. Please refer to our answer to reviewer bmjs where we discuss in detail the assumptions of our work and relation to other results in the literature.
>
> **Regarding the benefit of CLDs and their $\epsilon$-variant.**
>
> The kinetic framework offers practical advantages that standard VP/VE SDEs do not offer:
>
> -**Forward tuning:** the position–noise level $\varepsilon$, the velocity–noise level $\sigma$, and the initial velocity scale $v$ can be adjusted independently of the drift parameter $a$, whereas in VP/VE the drift and diffusion are rigidly coupled. This gives extra freedom to practitioners.
>
> -**Training objective:** the extended phase space admits Hybrid Score Matching (HSM) in addition to classical DSM training, and Dockhorn et al. showed empirically that HSM  lowers gradient variance during the training.
>
> -**Backward solver:** because the reverse flow is Hamiltonian‑like, one can employ structure‑preserving integrators such as symplectic or leapfrog schemes; on our toy problems leapfrog consistently outperforms Euler–Maruyama, even in the $\varepsilon$‑regularized variant. We leave for future work the theoretical analysis of such schemes.
>
> -**$\epsilon$-regularized version:**  the introduction of the $\epsilon$ parameter was initially motivated theoretically by Lemma B.1, where we saw that larger $\epsilon$ loosens the stepsize condition on $h$—i.e., strengthens the effective contraction, although a full global trade‑off analysis remains open.We however, illustrated this effect empirically on toy datasets and saw that a small regularization  yield improvement in the $\mathcal{W}_2$ metric on a Funnel dataset in dimension 50. We further illustrate this claim by extending our empirical analysis to dimension 100. Moreover, we have added results on two additional challenging synthetic datasets: MG-25 in dimension 100 (a 25-mode Gaussian mixture) and Diamonds (a 2D Gaussian mixture arranged in a diamond-shaped configuration). All these additional experiments (Tables 1, 2 and 3 in our response to reviewer N5Lx) further support our claims and exhibit trends consistent with our earlier findings.
>
>  We hope these comments meet the expectations and would be happy to clarify any remaining questions.
>
> **References:**
> - Villani, C. (2008). Optimal Transport: Old and Young. Springer.

---

> > ### Comment · Reviewer_N2VV · 2025-08-04
> > **Response**
> >
> > Thanks for the detailed response. I do not have further questions at this stage. I will raise the score.

---

### Official Review · Reviewer_8HZL · 2025-07-02

**Clarity:** 3
**Significance:** 3
**Originality:** 3
**Rating:** 5
**Confidence:** 3

**Summary:**

Score-based generative models rely on a forward diffusion process that reduces the data distribution into an easy-to-sample target distribution (often a Gaussian). Therefore, a wide variety of diffusion processes can be used in principle. This paper focuses on the study of the Wasserstein convergence of the critically-damped Langevin diffusion, which often showcases a smoother sampling trajectory when compared to the more standard choices.

This paper studies the Wasserstein convergence of diffusion models that seek to invert the critically-damped Langevin diffusion, and provides a convergence result (Theorem 3.1) that characterizes Wasserstein distance on the extended phase space $\pi\_{\text{data}} \otimes \pi\_v$. Building upon this result, this paper posits that the convergence property of this forward process can be improved by a slight modification of the diffusion coefficient, and illustrates empirically via a synthetic dataset that the introduction of this parameter leads to better Wasserstein convergence.

**Questions:**

None

**Ethical Concerns:**

["NO or VERY MINOR ethics concerns only"]

**Final Justification:**

I have interacted with the authors on my reviews of the paper, and I am mostly satisfied with the response.

My initial assessment of the paper was positive, and as I already recommend acceptance for this paper, I maintain my current assessment.

**Limitations:**

Yes

**Quality:**

3

**Strengths And Weaknesses:**

Overall, I think this paper is well-written with interesting theoretical results in the study of forward processes (often outlooked in the context of diffusion modeling).

## Strengths
- The paper presents a theoretical foundation to the study of critically-damped Langevin diffusion as a candidate for forward processes, which will be of general interest to researchers of diffusion models. The $W_2$ convergence presents a new result.
- The paper is clearly written, with useful references to related work that draws parallel between KL convergence and Wasserstein convergence.
- While this paper does not provide an explanation about why CLD can outperform standard Ornstein-Uhlenbeck process, its suggestion of a modified forward process is nevertheless useful in itself, and provides further guidelines to further study of forward processes backed by convergence theory.
## Weaknesses
I think this paper's strengths outweigh its weaknesses, however, I find the theoretical insights provided by this paper somewhat limited in applicability because of the following reasons.
- Although this paper presents a novel result, it does not ultimately answer the question of the better numerical performance of kinetic Langevin sampler compared to the standard Ornstein-Uhlenbeck process: instead, the paper opts to provide an alternative forward process that better exploits the added flexibility introduced by auxiliary variables.
- Theorem 3.1 seems useful, but one can argue that convergence in the extended state space  $\pi\_{\text{data}} \otimes \pi\_v$ is not a point of interest: the moment variables are introduced as auxiliary variables for the forward process.
- The numerical illustration seems somewhat contrived. The setting is largely synthetic, and the paper admits that comparing across different choices of $\epsilon$s can be problematic.
- The absence of a practical dataset is usually acceptable for a theory paper, however, it still constitutes a weakness for this paper, especially when the paper proposes a brand new way of parametrizing forward processes. When comparing across different $\epsilon$s can be seen as subjective, it will be compelling to see the introduction of the regularization parameter make a different in a practical setting.

---

> ### Author Rebuttal · Authors · 2025-07-31
>
> We thank the reviewer for the positive and thoughtful review, as well as the opportunity to clarify our contributions. We address each point below and have already incorporated the corresponding changes into the revised manuscript. All references are cited consistently with those in the original version.
>
> **[W1] On the Theoretical Contribution and Practical Relevance of CLD.**
>
> Previous empirical work (Dockhorn et al., 2022) shows that CLD can outperform standard SGMs forwards on image benchmarks, yet no $\mathcal{W}\_{2}$ analysis existed; existing theory for CLD is restricted to KL (Chen et al., 2023; Conforti et al., 2025) and to a specific parametrization of the Langevin dynamics.  Our goal is to close this gap (Theorems 3.1 and 4.1) and place CLD on the same footing as VP/VE SDEs in Wasserstein distance, while maintaining the state‑of‑the‑art minimal regularity assumptions on the data distribution.
>
> By providing the first $\mathcal{W}\_{2}$ guarantees for CLD, we show that the method is not only empirically competitive but also theoretically sound and tunable, offering an alternative to classical SGMs. In this way, as training in the coupled phase space has been shown by Dockhorn et al. (2022) to be empirically more stable (in particular the Hybrid Score Matching loss), our results certify that the associated sampler remains $\mathcal{W}\_{2}$‑stable even when the score is learned only approximately (Assumption H3).
>
> Moreover, as the reverse SDE is Hamiltonian‑like, it is possible to design structure‑preserving integrators such as symplectic or leapfrog schemes, which are inaccessible to standard SGMs. On our toy example leapfrog integrator consistently outperforms Euler–Maruyama in generation quality, even in the $\varepsilon$‑regularized setting. This perspective paves the way for improved discretization error rates. While a full analysis of these structure‑preserving schemes is beyond the scope of this work, our results lay the groundwork for future improvements in discretization error rates.
>
> **[W2] Convergence rate for $\mathcal{W}\_2 ( \pi\_{\mathrm{data}}, \mathcal{L}(\bar X\_T^{\theta}) )$.**
>
> Since both training and sampling operate in the extended phase space, we established a convergence result for the joint distribution, even though the main quantity of interest is the marginal data distribution. The error on the data distribution is directly controlled by the joint bound via contraction under projection of the $\mathcal{W}_2$ distance,
> $$ \mathcal{W}\_2 ( \pi\_{\mathrm{data}}, \mathcal{L}(\bar X\_T^{\theta}) ) \leq  \mathcal{W}\_2 \left( \pi\_{\mathrm data}\otimes \pi\_v ,\mathcal{L}(  \mathbf{\bar U}^\theta_T) \right)  .$$
> We have added this clarification to the revised version of the paper.
>
> **[W3] Extended experiments and impact of $\varepsilon$.**
>
> We understand the concern regarding the use of real, high-dimensional datasets. However, training such models is very computationally intensive, and as our work is primarily focused on theoretical guarantees, we restricted our empirical validation to toy datasets that serve to illustrate our mathematical claims. In this way, the introduction of the $\varepsilon$ parameter was initially motivated theoretically by Lemma C.1, where we showed that larger $\varepsilon > 0$ loosens the stepsize condition on $h$—i.e., strengthens the effective contraction. However, we acknowledge that a complete global analysis of this trade-off remains an open question.
>
> To address this concerns within the limited rebuttal window, we extended our numerical experiments to a higher-dimensional toy setting. Specifically, we provide an analysis of the sensitivity of the $\varepsilon$ parameter on the 100-dimensional Funnel dataset across different choices of drift $a$. We report the mean and standard deviation over 5 runs in Table 1. In this extended empirical analysis it appears that the benefit of a small $\varepsilon$ regularization is confirmed even for different choices of $a$. Moreover, note that in our ablation study, we set $\sigma$ to $\sqrt{2}$ so that the case where $a=2$ and $\epsilon=0$ exactly matches the setting recommended by Dockhorn et al. (2022). We also adopted the same time renormalization as them to make the comparison as fair as possible (see Appendix C.1).
>
> On the one hand, we observe that this parametrization is highly effective on toy datasets. On the other hand, several $\varepsilon$-regularized variants perform even better. To further support this observation, we are extending our evaluation to additional challenging toy datasets: MG-25 (a 25-mode 100D Gaussian mixture) and Diamonds (a 2D Gaussian mixture with diamond geometry). Preliminary results suggest that the same conclusions hold (see Tables below).
> If of interest, we are happy to include standard deviations in the revised paper and to reproduce plots analogous to Figure 1 for these new datasets.
>
> **Table 1: Comparison of mean Wasserstein distance for different noise levels $\varepsilon$ on the Funnel‑100D**
> *(Mean ± standard deviation across 5 runs; lower is better)*
>
> | $\varepsilon$ |        | $a=0.25$             | $a=0.5$              | $a=1.0$                | $a=2.0$                |
> |:-------------:|--------|:--------------------:|:--------------------:|:----------------------:|:----------------------:|
> | 0             |        | 0.73 ± 0.0016        | 0.291 ± 0.0047       | 0.225 ± 0.056          | 0.223 ± 0.011          |
> | 0.1           |        | 0.632 ± 0.0016       | 0.278 ± 0.0012       | 0.158 ± 0.027          | 0.198 ± 0.004          |
> | 0.25          |        | 0.409 ± 0.0031       | 0.248 ± 0.012        | **0.137 ± 0.0046**     | **0.179 ± 0.0057**     |
> | 0.5           |        | **0.248 ± 0.0016**   | 0.228 ± 0.0051       | 0.157 ± 0.0022         | 0.203 ± 0.0028         |
> | 1             |        | 0.785 ± 0.011        | **0.191 ± 0.0079**   | 0.253 ± 0.006          | 0.233 ± 0.002          |
>
>
> **Table 2: Comparison of mean Wasserstein distance for different noise levels $\varepsilon$ on the MG25‑100D**
>
> | $\varepsilon$ |        | $a=0.1$        | $a=0.25$       | $a=0.5$         | $a=1.0$         | $a=2.0$         |
> |:-------------:|:------:|:--------------:|:--------------:|:---------------:|:---------------:|:---------------:|
> | 0             |        | 0.285          | 0.199          | 0.0346          | 0.0083          | 0.009           |
> | 0.1           |        | 0.191          | 0.158          | 0.0262          | **0.0063**      | 0.008           |
> | 0.25          |        | **0.013**      | 0.0654         | 0.0157          | 0.0076          | **0.007**       |
> | 0.5           |        | 0.188          | **0.005**      | **0.0093**      | 0.0071          | 0.0081          |
> | 1             |        | 0.381          | 0.0491         | 0.0121          | 0.0064          | **0.007**       |
>
>
> **Table 3: Comparison of mean Wasserstein distance for different noise levels $\varepsilon$ on the Diamond‑2D**
>
> | $\varepsilon$ |        | $a=0.1$        | $a=0.25$       | $a=0.5$         | $a=1.0$         | $a=2.0$         |
> |:-------------:|:------:|:--------------:|:--------------:|:---------------:|:---------------:|:---------------:|
> | 0             |        | 0.323          | 0.252          | 0.0325          | 0.0058          | 0.0077          |
> | 0.1           |        | 0.233          | 0.198          | 0.0237          | **0.0038**      | 0.0047          |
> | 0.25          |        | **0.0474**     | 0.0737         | 0.0214          | 0.0049          | **0.004**       |
> | 0.5           |        | 0.0722         | **0.0078**     | **0.0077**      | 0.0057          | 0.0056          |
> | 1             |        | 0.0958         | 0.0286         | 0.0143          | 0.0136          | 0.012           |
>
> We thank the reviewer once again for their insightful comments. The revisions directly address each point raised and, we believe, further strengthen the manuscript. We hope the changes align with the reviewer’s expectations and remain available to clarify any remaining questions.

---

> > ### Comment · Reviewer_8HZL · 2025-08-05
> >
> > Thank you for your detailed response and it is helpful in clearing up areas of confusion. I have no further questions at this stage.
> >
> > As I already recommend acceptance for this paper, I shall maintain my assessment.

---

### Official Review · Reviewer_bmjs · 2025-07-04

**Clarity:** 2
**Significance:** 2
**Originality:** 3
**Rating:** 4
**Confidence:** 2

**Summary:**

The paper investigates the convergence behavior of critically-damped Langevin diffusions and derives upper bounds on the sampling error in terms of the 2-Wasserstein distance between the generated and target data distributions. The main contribution is Theorem 3.1, which, under assumptions H1–H3, establishes a bound on the Wasserstein distance comprising three additive components: discretization error, approximation error, and a convergence term related to the initial distribution. This analysis contrasts with prior results that primarily focused on KL divergence. Furthermore, Theorem 4.1 strengthens the bound by introducing an additional assumption of strong log-concavity of the intermediate distributions $p_t$ for all  $0 \leq t \leq T$, leading to improved convergence guarantees.

**Questions:**

Please see my previous response.

**Ethical Concerns:**

["NO or VERY MINOR ethics concerns only"]

**Final Justification:**

I find the authors' responses satisfactory and therefore increased my score to 4.

**Limitations:**

Please see my previous response.

**Quality:**

3

**Strengths And Weaknesses:**

**Strengths:**

1. The paper addresses a relevant theoretical topic concerning the convergence of Langevin diffusions.

2. The convergence guarantee is provided in terms of the Wasserstein distance, which could be useful for other theoretical studies of Langevin-based generative modeling.

**Weaknesses:**

1. The paper is theoretically heavy and difficult to follow for a general machine learning audience. The authors do not provide simplified examples or intuitive remarks to help readers understand the main results. I highly recommend that the authors include intuitive examples or interpretations of their theorems to help readers better appreciate the significance and applicability of the results. In the current form, it is difficult to assess the non-triviality and practical value of the theoretical contributions.

2. While the authors offer a brief justification on page 5 regarding the mildness of their assumptions, it would be much clearer if they included concrete examples of data distributions that satisfy these assumptions. In particular, Assumption H3 appears non-standard and seems difficult to verify in practice. For instance, assuming the data distribution is Gaussian, how would one find a valid constant $M$ to apply the theoretical result?

3. The constant $h$ in Theorems 3.1 and 4.1 is vaguely defined. While looking into the proofs, I noticed that certain conditions on $h$ are required, but these are not mentioned explicitly in the theorem statements. Moreover, it appears that the constant
$c_1$ may depend on the choice of $h$, which raises concerns about whether the bound might hold trivially. The authors should provide a more detailed discussion of the role of $h$ in the bounds to clarify the implications and non-triviality of their results.

4. Following the above point, I would like to ask the authors to show what Theorem 3.1 or 4.1 mean when applied to a standard distribution such as the unit-variance Gaussian or uniform distribution. Can they compute the corresponding constants $h$ and $M$ in these cases to illustrate the implications of the bound in concrete settings?

5. Finally, it would be nice if the authors elaborated on the comparison between their Wasserstein-2 bounds and the existing KL divergence bounds. Is it possible to relate the two using standard inequalities between divergence measures? A discussion of such a relationship would enhance the understanding of the novelty and potential advantages of the authors' analysis.

---

> ### Author Rebuttal · Authors · 2025-07-31
>
> We thank the reviewer for their valuable feedback and have addressed each point below. The proposed revisions have been incorporated into the revised paper. Based on these comments mentioning that the paper is difficult to follow and lacks of intuitive remarks and comments or examples on the assumptions, we simplified and clarified the presentation of the main results in the revised version. Without loss of generality and without modifying our results we propose the two following revisions.
>
>
> - First, we provide a unified result that holds for all $\epsilon \geq 0$, rather than treating the cases $\epsilon=0$ and $\epsilon>0$ separately. This reformulation aims to make the theoretical contributions more accessible to a broader audience and clarifies the discussions (see below for our comments on the assumptions).
>
> - To clarify our assumptions and better compare our work with the existing literature (see, e.g., [1]), we have added a paragraph discussing our assumptions, in particular the one-sided Lipschitz assumption. Such assumptions on the score function provide a bound on the largest eigenvalue of its Jacobian and are the weakest assumptions in the score-based generative models literature, see [1,2].
> Our work aligns with the most recent results, and our assumptions cover in particular data distributions which are not covered by the strong logconcave assumptions used in the literature. In order to simplify the paper and to compare our framework with the existing literature we propose to replace H2 and H4 in the revised paper by the following assumption.
>
> **New Assumption H2'**.
> The data distribution is of the form  $\pi_{\rm data} (x) \propto \exp{(-u(x) + a(x))}$ and satisfies:
>
> - There exist $\alpha>0$ and $L_0>0$ such that $\alpha I_{d} \preceq \nabla^{2}u(x) \preceq L_0 I_{d}$ for all $x\in\mathbb{R}^{d}$.
> - There exists  $K>0$ such that $|a(x)-a(y)|\le K||x-y||$ for all $x,y\in\mathbb{R}^{d}$ with $||x-y||\le1$.
>
> This minor adaptation of our Lipschitz-type assumptions is consistent with the weakest assumptions found in the literature on classical score-based generative models (i.e. without the Langevin forward dynamics) and has no impact on our main results. This assumption models the data distribution as a strongly log-concave component $u(x)$ perturbed by a term $a(x)$.
> Intuitively, this assumption allows the distribution to deviate from strict log-concavity via the perturbation $a(x)$, while still maintaining sufficient regularity for analysis. When $a(x)=0$, the distribution reduces to a strongly log-concave case.
>
> **Discussion on Assumptions.**
>
> The novel assumptions proposed above are satisfied by standard distributions such as Gaussian and mixtures of Gaussian distributions. They are strictly weaker than classical assumptions like strong log-concavity (H4), which holds only for non-degenerate Gaussian distributions and excludes many practically relevant settings although it is still common in the literature. Regarding Assumption H3, as discussed in Section 3, it quantifies the score approximation error and is standard in recent works that establish non-asymptotic Wasserstein convergence guarantees for diffusion-based models (e.g., [2, 3]).
>
> In the particular case where the data distribution is Gaussian, the score function admits a closed form expression and does not require approximation via a neural network. Consequently, the score approximation error vanishes, and the constant $M$ in our bound can be as chosen as $M=0$.
>
>
> **Clarification on the step size $h$ and its impact.**
>
> We acknowledge that the definition of $h$ was not clearly specified earlier. The constant $h$ in Theorems 3.1 and 4.1 corresponds to the step size in the Euler--Maruyama discretization of the backward SDE. This is now clearly stated in the revised version of the paper.
>
> In Theorem 3.1, the result holds for all $h > 0$; the only condition we implicitly use in the proof is the elementary inequality $h \leq (1 - \mathrm{e}^{-h}) \mathrm{e}^h$, which is equivalent to $h + 1 \leq \mathrm{e}^{h}$, a bound satisfied for all $h > 0$ (as mentioned a few lines earlier in the proof).
> For Theorem 4.1, we already stated that the result holds for $h < h_\epsilon$, where $h_\epsilon$ is defined in Appendix B. We have now added the explicit form of $h_\epsilon$ to the main text for clarity.
>
> Regarding the dependence on $h$ in $c_2$, the discretization error introduces a term of order $\mathcal{O}(\sqrt{h})$. In fact, the discretization error takes the form $c_2 \sqrt{h} + o(h)$. Since $h$ is typically chosen to be sufficiently small in practice (see also [3, 4]), the higher-order term becomes negligible, and the $\sqrt{h}$ term dominates. In practice, discretization is performed using a uniform grid, with step size $h=T/N$, where $T$ is the total time horizon and $N$ is the number of discretization steps.
>
>
> **Comparison with KL divergence bounds and known results.**
>
> In the strongly log‑concave setting, our Wasserstein‑2 results cannot be recovered from existing KL‑divergence bounds.  Indeed, if the data distribution $\pi_{\mathrm{data}}$ is $c$-strongly log-concave, Talagrand’s inequality yields
> $$
> \mathcal{W}\_{2}^{2} \left(\pi\_{\mathrm{data}}\otimes \pi\_v, \mathcal{L} ( \bar{\mathbf{U}}^\theta\_T)\right) \leq
> \frac{2}{c} \mathrm{KL}\left(\mathcal{L}( \bar{\mathbf{U}}^\theta_T), \pi\_{\mathrm{ data}}\otimes \pi_v\right).
> $$
> By contrast, for distributions with compact support one may also bound the squared Wasserstein‑2 distance by the KL divergence ([6], Thm. 6.12, p. 82), but the corresponding constant depends on the diameter of the support, which can grow prohibitively large in high dimensions (e.g., image data).
>
> Since KL divergence is not symmetric, existing KL-based results only control
> $\mathrm{KL}\left(\pi_{\mathrm{data}}\otimes \pi_v,  \mathcal{L} (\bar{\mathbf{U}}^\theta_T)\right)$, so our convergence bound cannot be deduced as a corollary.
> Indeed, [3] also assumes strong log-concavity of the data distribution (among other conditions) in $\mathbb{R}^d$ and explicitly emphasizes that “our results are not implied by the existing convergence results for SDE-based samplers” ([3], p. 5).
> Finally, Theorem 3.1 in our work holds under far weaker assumptions (requiring neither strong log-concavity nor compact support). Moreover, existing KL-based analyses (e.g., [4, 5]) focus on a specific case of Langevin dynamics with $a = 1$ and $\sigma = 2$, which yields an isotropic Gaussian stationary distribution—this significantly simplifies the analysis. In contrast, our analysis covers the general CLD setting and extends beyond it to include an additional noise parameter $\varepsilon$.
>
> We hope that these additions fully address the reviewer’s suggestions and contribute to strengthening the paper. We are grateful for the valuable feedback and trust that the revised version meets the expectations raised.
>
> **References**
>
> [1] Stéphanovitch, A. (2025). Regularity of the score function in generative models. arXiv:2506.19559.
>
> [2] Gentiloni-Silveri, M. and Ocello, A. (2025). Beyond Log-Concavity and Score Regularity: Improved Convergence Bounds for Score-Based Generative Models in W2-distance. https://arxiv.org/abs/2501.02298.
>
> [3] Gao, X., Nguyen, H. M., and Zhu, L. (2025). Wasserstein convergence guarantees for a general class of score-based generative models. Journal of Machine Learning Research.
>
> [4] Chen, S., Chewi, S., Li, J., Li, Y., Salim, A., and Zhang, A. (2023). Sampling is as easy as learning the score: theory for diffusion models with minimal data assumptions. In International Conference362on Learning Representations.
>
> [5] Conforti, G., Durmus, A., and Silveri, M. G. (2025). Kl convergence guarantees for score diffusion models under minimal data assumptions. SIAM Journal on Mathematics of Data Science.
>
> [6] Villani, C. (2008). Optimal Transport: Old and Young. Springer.

---

> > ### Comment · Reviewer_bmjs · 2025-08-04
> > **Thanks for the response**
> >
> > I thank the authors for their thoughtful responses, which help with my comments on the work's presentation. I will update my score accordingly.

---

### Decision · Program_Chairs · 2025-09-17

**Decision:**

Accept (poster)

**Comment:**

Critically damped Langevin dynamics has been proposed in the literature as an alternative to the overdamped Ornstein Uhlenbeck forward process for diffusion models. There is some evidence that this process can be better in terms of training stability than the overdamped forward process. Novel integrators such as leap-frog etc are also possible in this setting. While prior works have provided convergence analysis of the diffusion models based on CLD in KL divergence, the current work derives convergence to target law in the $\mathcal{W}_2$ metric. The reviewers appreciated the rigor and depth of the mathematical analysis.

The main weaknesses of this paper pointed out by the reviewers were:
- This work does not demonstrate the advantage of CLD over the usual forward process in terms of the bounds.
- The assumption of strong-log-concavity, especially in Theorem 4.1, can be especially stringent. In this setting $\mathsf{KL}$ convergence from prior works can already imply $\mathcal{W}_2$ convergence using the $T_2$ inequality.

The authors gave a satisfactory response to most of these concerns. I am leaning towards accepting this paper.